# Is Behavior Cloning All You Need?
# Understanding Horizon in Imitation Learning

**Dylan J. Foster**
Microsoft Research
dylanfoster@microsoft.com

**Adam Block**
Department of Mathematics
MIT
ablock@mit.edu

**Dipendra Misra**
Microsoft Research
dipendrakumar.misra@databricks.com

## Abstract

Imitation learning (IL) aims to mimic the behavior of an expert in a sequential decision making task by learning from demonstrations, and has been widely applied to robotics, autonomous driving, and autoregressive text generation. The simplest approach to IL, *behavior cloning* (BC), is thought to incur sample complexity with unfavorable *quadratic* dependence on the problem horizon, motivating a variety of different *online* algorithms that attain improved *linear* horizon dependence under stronger assumptions on the data and the learner's access to the expert.

We revisit the apparent gap between offline and online IL from a learning-theoretic perspective, with a focus on the realizable/well-specified setting with general policy classes up to and including deep neural networks. Through a new analysis of behavior cloning with the *logarithmic loss*, we show that it is possible to achieve *horizon-independent* sample complexity in offline IL whenever (i) the range of the cumulative payoffs is controlled, and (ii) an appropriate notion of supervised learning complexity for the policy class is controlled. Specializing our results to deterministic, stationary policies, we show that the gap between offline and online IL is smaller than previously thought: (i) it is possible to achieve *linear* dependence on horizon in offline IL under dense rewards (matching what was previously only known to be achievable in online IL); and (ii) without further assumptions on the policy class, online IL cannot improve over offline IL with the logarithmic loss, even in benign MDPs. We complement our theoretical results with experiments on standard RL tasks and autoregressive language generation to validate the practical relevance of our findings.

## 1 Introduction

Imitation learning (IL) is the problem of emulating an expert policy for sequential decision making by learning from demonstrations. Compared to reinforcement learning (RL), the learner in IL does not observe reward-based feedback, and must imitate the expert's behavior based on demonstrations alone; their objective is to achieve performance close to that of the expert on an *unobserved* reward function. Imitation learning is motivated by the observation that in many domains, demonstrating the desired behavior for a task (e.g., robotic grasping) is simple, while designing a reward function to elicit the desired behavior can be challenging. IL is also often preferable to RL because it removes the need for exploration, leading to empirically reduced sample complexity and often much more stable training. Indeed, the relative ease of applying IL (over RL methods) has led to extensive adoption, ranging from classical applications in autonomous driving [63] and helicopter flight [1] to contemporary works that

38th Conference on Neural Information Processing Systems (NeurIPS 2024).

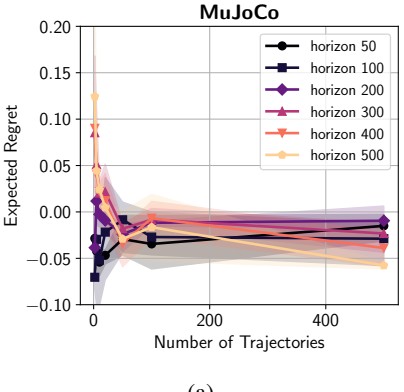
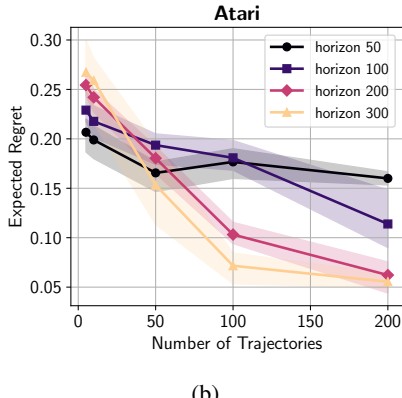

(a)                                                    (b)

Figure 1: Suboptimality of a policy learned with log-loss behavior cloning (`LogLossBC`) as a function of the number of expert trajectories, for varying values of horizon $H$. In each environment, an imitator is trained according to `LogLossBC` and the regret with respect to the expert is reported, with reward normalized to be horizon-independent. **(a)** Continuous control with MuJoCo environment `Walker2d-v4`. **(b)** Discrete control with Atari environment `BeamRiderNoFrameskip-v4`. For both environments, we find that the regret is *independent of horizon* (or in the case of Atari, slightly improving with horizon), as predicted by our theoretical results. Full experimental details are provided in Appendix C.

leverage deep learning to achieve state-of-the-art performance for self-driving vehicles [15, 6, 44], visuomotor control [30, 103], navigation [45], and game AI [46, 90]. Imitation learning also offers a conceptual framework through which to study autoregressive language modeling [21, 13], and a number of useful empirical insights have arisen as a result of this perspective. However, a central challenge limiting broader real-world deployment is to understand and improve the reliability and stability properties of algorithms that support general-purpose (deep/neural) function approximation.

In more detail, imitation learning algorithms can be loosely grouped into *offline* and *online* approaches. Offline imitation learning algorithms only require access to a dataset of logged trajectories from the expert, making them broadly applicable. The most widely used approach, *behavior cloning*, reduces imitation learning to a standard supervised learning problem in which the learner attempts to predict the expert's actions from observations given the collected trajectories. The simplicity of this approach allows the learner to leverage the considerable machinery developed for supervised learning and readily incorporate complex function approximation with deep models [10, 70]. On the other hand, BC seemingly ignores the problem of *distribution shift*, wherein small deviations from the expert policy early in rollout lead the learner off-distribution to regions where they are less able to accurately imitate. This apparent *error amplification* phenomenon has been widely observed empirically [70, 54, 13], and motivates *online* or *interactive* approaches to imitation learning [70, 72, 71, 78], which avoid error amplification by interactively querying the expert and learning to correct mistakes on-policy.

In theory, online imitation learning enables sample complexity guarantees with improved (linear, as opposed to quadratic) dependence on horizon for favorable MDPs. Yet, while online imitation learning has found empirical success [73, 52, 38, 6, 26, 51, 7, 108, 59], online access to the expert can be costly or infeasible in many applications, and offline imitation learning remains a dominant empirical paradigm. Motivated by this disconnect between theory and practice, we aim to understand to what extent the apparent gap between offline and online imitation learning is fundamental. We ask:

*Is online imitation learning truly more sample-efficient than offline imitation learning, or can existing algorithms or analyses be improved?*

## 1.1 Background: Offline and Online Imitation Learning

To motivate our results, we begin by formally introducing the offline and online imitation learning frameworks, highlighting gaps in current sample complexity guarantees concerning *horizon dependence*. We take a *learning-theoretic* perspective, with a focus on general policy classes.

**Markov decision processes.** We study imitation learning in episodic Markov decision processes. Formally, a Markov decision process $M = (\mathcal{X}, \mathcal{A}, P, r, H)$ consists of a (potentially large) state

space $\mathcal{X}$, action space $\mathcal{A}$, horizon $H$, probability transition function $P = \{P_h\}_{h=0}^H$, where $P_h : \mathcal{X} \times \mathcal{A} \to \Delta(\mathcal{X})$, and reward function $r = \{r_h\}_{h=1}^H$, where $r_h : \mathcal{X} \times \mathcal{A} \to \mathbb{R}$. A (randomized) policy is a sequence of per-timestep functions $\pi = \{\pi_h : \mathcal{X} \to \Delta(\mathcal{A})\}_{h=1}^H$. The policy induces a distribution over trajectories $(x_1, a_1, r_1), \ldots, (x_H, a_H, r_H)$ via the following process. The initial state is drawn via $x_1 \sim P_0(\varnothing)$,[1] then for $h = 1, \ldots, H$: $a_h \sim \pi(x_h)$, $r_h = r_h(x_h, a_h)$, and $x_{h+1} \sim P_h(x_h, a_h)$. For notational convenience, we use $x_{H+1}$ to denote a deterministic terminal state with zero reward. We let $\mathbb{E}^\pi[\cdot]$ and $\mathbb{P}^\pi[\cdot]$ denote expectation and probability law for $(x_1, a_1), \ldots, (x_H, a_H)$ under this process, respectively.[2]

The expected reward for policy $\pi$ is given by $J(\pi) := \mathbb{E}^\pi\left[\sum_{h=1}^H r_h\right]$, and the value functions for $\pi$ are given by $V_h^\pi(x) := \mathbb{E}^\pi\left[\sum_{h'=h}^H r_{h'} \mid x_h = x\right]$ and $Q_h^\pi(x, a) := \mathbb{E}^\pi\left[\sum_{h'=h}^H r_{h'} \mid x_h = x, a_h = a\right]$.

**Reward normalization.** To study the role of horizon in imitation learning in a way that disentangles the effects of reward scaling from other factors, we assume that rewards are normalized such that $\sum_{h=1}^H r_h \in [0, R]$ for a parameter $R > 0$ [47, 94, 104, 48]. We refer to the setting in which $r_h \in [0, 1]$ for all $h \in [H]$, which is the focus of most prior work [70, 72, 71, 67–69, 80], as the *dense reward setting*, which has $R \leq H$; we will frequently specialize our results to this setting.

### 1.1.1 Offline Imitation Learning: Behavior Cloning

Let $\pi^\star = \{\pi_h^\star : \mathcal{X} \to \Delta(\mathcal{A})\}_{h=1}^H$ denote the *expert policy*. In the offline imitation learning setting, we receive a dataset $\mathcal{D} = \{o^i\}_{i=1}^n$ of (reward-free) trajectories $o^i = (x_1^i, a_1^i), \ldots, (x_H^i, a_H^i)$ obtained by executing $\pi^\star$ in the underlying MDP $M^\star$. Using these trajectories, our goal is to learn a policy $\widehat{\pi}$ such that the rollout regret $J(\pi^\star) - J(\widehat{\pi})$ to $\pi^\star$ is as small as possible. *We emphasize that $\pi^\star$ is an arbitrary policy, and is not assumed to be optimal.*

**Behavior cloning.** *Behavior cloning*, which reduces the imitation learning problem to supervised prediction, is the dominant offline imitation learning paradigm. To describe the algorithm in its simplest form, consider the case where $\pi^\star := \{\pi_h^\star : \mathcal{X} \to \mathcal{A}\}_{h=1}^H$ is deterministic. For a user-specified policy class $\Pi \subset \{\pi_h : \mathcal{X} \to \Delta(\mathcal{A})\}_{h=1}^H$, the most basic version of behavior cloning [70] solves the supervised classification problem $\widehat{\pi} = \arg\min_{\pi \in \Pi} \sum_{i=1}^n \frac{1}{H} \sum_{h=1}^H \mathbb{I}\{\pi_h(x_h^i) \neq a_h^i\} =: \widehat{L}_{\mathsf{bc}}(\pi)$. Naturally, other classification losses (e.g., square loss, logistic loss, or log loss) may be used in place of the indicator loss.[3] To provide sample complexity bounds for this algorithm, we make a standard *realizability assumption* (e.g., Agarwal et al. [2], Foster and Rakhlin [33]).

**Assumption 1.1** (Realizability). *The policy class $\Pi$ contains the expert policy, i.e. $\pi^\star \in \Pi$.*

This assumption asserts that $\Pi$ is expressive enough to represent the expert policy. Depending on the application, $\Pi$ might be parameterized by simple linear models, or by flexible models such as convolutional neural networks or transformers. We primarily restrict our attention to the realizable setting throughout the paper, as it is meaningful and non-trivial, yet not fully understood. Our main results extend to provide guarantees for the misspecified case, but a thorough study of the role of misspecification is beyond the scope of this work. To simplify presentation, we adopt a standard convention in RL theory and focus on finite classes with $|\Pi| < \infty$ [2, 33].

To proceed with analyzing the behavior cloning algorithm, a standard uniform convergence argument implies that if we define $L_{\mathsf{bc}}(\pi) = \frac{1}{H} \sum_{h=1}^H \mathbb{P}^{\pi^\star}[\pi_h(x_h) \neq \pi_h^\star(x_h)]$, then with probability at least $1 - \delta$, behavior cloning has

$$L_{\mathsf{bc}}(\widehat{\pi}) \lesssim \frac{\log(|\Pi|\delta^{-1})}{n}.$$

Meanwhile, a standard error analysis for BC leads to the following bound on rollout performance:

$$J(\pi^\star) - J(\widehat{\pi}) \lesssim RH \cdot L_{\mathsf{bc}}(\widehat{\pi}). \tag{1}$$

---

[1] We use the convention that $P_0(\varnothing)$ denotes the initial state distribution.

[2] To simplify presentation, we assume that $\mathcal{X}$ and $\mathcal{A}$ are countable, but our results trivially extend to general spaces with an appropriate measure-theoretic treatment.

[3] Behavior cloning for stochastic expert policies has received limited attention in theory [67], but the logarithmic loss is widely used in practice. One contribution of our work is to fill this lacuna.

Combining these bounds, we conclude that

$$J(\pi^\star) - J(\widehat{\pi}) \lesssim RH \cdot \frac{\log(|\Pi|\delta^{-1})}{n}. \tag{2}$$

For the *dense reward setting* where $R = H$, this leads to *quadratic* dependence on horizon; that is, $\Omega(H^2)$ trajectories are required to achieve constant accuracy. Unfortunately, both steps in this argument are tight in general:

- The generalization bound $L_{\mathsf{bc}}(\widehat{\pi}) \lesssim \frac{\log(|\Pi|\delta^{-1})}{n}$ is tight even when $|\Pi| = 2$ (this is true not just for the indicator loss, but for other standard losses such as square loss, absolute loss, and hinge loss). Since the amount of information in a trajectory grows with $H$, one might hope a-priori that the generalization error would decrease with $H$; alas, this does not occur due to the *dependence* between samples in each trajectory.

- Ross and Bagnell [70] show that the inequality $J(\pi^\star) - J(\widehat{\pi}) \lesssim RH \cdot L_{\mathsf{bc}}(\widehat{\pi})$ is tight for MDPs with 3 states; the quadratic scaling in $H$ this induces under dense rewards is often attributed to *error amplification* or *distribution shift* incurred by passing from error under the state distribution of $\pi^\star$ to the state distribution of $\widehat{\pi}$.

Combining, these observations, it is natural to conclude that offline imitation learning is fundamentally harder than supervised classification, where linear dependence on horizon might be expected (e.g., with $H$ independent prediction tasks).

### 1.1.2 Online Imitation Learning and Recoverability

The aforementioned limitations of behavior cloning have motivated *online* approaches to IL [70, 72, 71, 78]. In the online framework, the learner can interactively choose policies to roll out and query the expert for the action at each state in the trajectory (see Appendix E.2 for a formal description), representing a substantially stronger (and in some cases unrealistic) assumption on the learner's access both to the MDP and the expert than in the offline setting. Online imitation learning can avoid error amplification and achieve improved dependence on horizon for MDPs that satisfy a *recoverability* condition [72, 68].

**Definition 1.1** (Recoverability parameter). *The* recoverability parameter *for an MDP $M^\star$ and expert $\pi^\star$ is given by*[4]

Under recoverability, the `Dagger` algorithm of Ross et al. [72] leverages online interaction by interactively querying the expert and learning to correct mistakes on-policy, leading to sample complexity

$$J(\pi^\star) - J(\widehat{\pi}) \lesssim \mu H \cdot \frac{\log|\Pi|}{n} \tag{3}$$

for any finite class $\Pi$ and deterministic expert policy $\pi^\star$ (for completeness, we include an analysis in Appendix E.2; see Propositions E.1 and E.2). For the dense reward setting where $R = H$, we can have $\mu = H$ in the worst case, in which case Eq. (3) matches the quadratic horizon dependence of behavior cloning, but when $\mu = O(1)$ (informally, this means it is possible to "recover" from a bad action that deviates from $\pi^\star$), the bound in Eq. (3) achieves linear dependence on horizon. Other online IL algorithms such as `Forward`, `Smile` [70], and `Aggrevate` [71] achieve similar guarantees (we are not aware of an approach that improves upon Eq. (3) for general finite classes).

The improvements of online IL notwithstanding, Eq. (2) is known to be tight for BC, but this is an *algorithm-dependent* (as opposed to information-theoretic) lower bound, and does not preclude the existence of more sample-efficient, purely offline algorithms. In this context, our central question can be restated as: *Can offline imitation learning algorithms achieve sub-quadratic horizon dependence for general policy classes $\Pi$?* While prior work has investigated this question for tabular and linear policies [67–69], we approach the problem from a new (learning-theoretic) perspective by considering general policy classes.

### 1.2 Contributions

We present several new results that clarify the role of horizon in offline and online imitation learning.

---

[4]For stochastic policies, we overload notation and write $f(\pi(x))$ as shorthand for $\mathbb{E}_{a \sim \pi(x)}[f(a)]$.

| | Parameter Sharing (Corollary 2.1) | No Parameter Sharing ($\Pi = \Pi_1 \times \cdots \Pi_H$) (e.g., [70]) |
|---|---|---|
| Sparse Rewards | $O\left(\frac{R\log(|\Pi|)}{n}\right)$ | $O\left(\frac{HR\log(\max_h |\Pi_h|)}{n}\right)$ |
| Dense Rewards ($R = H$) | $O\left(\frac{H\log(|\Pi|)}{n}\right)$ | $O\left(\frac{H^2\log(\max_h |\Pi_h|)}{n}\right)$ |

Table 1: Summary of upper bounds for deterministic experts; lower bounds are more nuanced, and discussed in Section 2.2. Each cell denotes the regret of a policy learned with log-loss behavior cloning (LogLossBC), which is optimal in each setting. Here, $\Pi$ is the policy class, $R$ is the reward range, $H$ is the horizon, and $n$ is the number of expert trajectories. In the dense-reward setting, we set $R = H$.

1. **Horizon-independent analysis of log-loss behavior cloning.** Through a new analysis of behavior cloning with the *logarithmic loss* (LogLossBC), we show that **it is possible to achieve *horizon-independent* sample complexity** [47, 94, 104, 105] in offline imitation learning whenever (i) the range of the cumulative payoffs is normalized, and (ii) an appropriate notion of supervised learning complexity for the policy class is controlled. Our result is facilitated by a novel information-theoretic analysis which controls policy behavior at the trajectory level, supporting both deterministic and stochastic expert policies.

2. **Deterministic policies: Closing the gap between offline and online IL.** Specializing LogLossBC to *deterministic stationary* policies (more generally, policies with parameter sharing) and cumulative rewards in the range $[0, H]$, we show that it is possible to achieve sample complexity with *linear* dependence on horizon in offline IL in arbitrary MDPs, matching was was previously only known of *online* IL. We complement this result with a lower bound showing that, without further structural assumptions on the policy class (e.g., no parameter sharing [67]), **online IL cannot improve over offline IL with LogLossBC**, even for benign MDPs. Our results are summarized in Table 1. Nonetheless, as observed in prior work [67], online imitation learning can still be beneficial for *non-stationary* policies.

3. **Stochastic policies: Tight understanding of optimal sample complexity.** For stochastic expert policies, our analysis of LogLossBC gives the first *variance-dependent* sample complexity bounds for imitation learning with general policy classes, which we prove to be tight in a problem-dependent and minimax sense. Using this result, we show that for stochastic stationary experts, (i) *quadratic dependence on the horizon is necessary* when cumulative rewards lie in the range $[0, H]$, in contrast to the deterministic setting, but (ii) LogLossBC—through our variance-dependent analysis—can sidestep this hardness and achieve linear dependence on horizon under a recoverability-like condition. Finally, we show that—as in the deterministic case—online IL cannot improve over offline IL with LogLossBC without further assumptions on the policy class. Our results are summarized in Table 2.

**Toward a learning-theoretic understanding of imitation learning.** Our findings highlight the need to develop a fine-grained, problem-dependent understanding of algorithms and complexity for IL. Instabilities of offline IL [60, 27, 13] and benefits of online IL [73, 52, 38, 6, 26, 51, 7, 108, 59] likely arise in practice, but existing assumptions in theoretical research are often too coarse to give insights into the true nature of these phenomena, leading to an important gap between theory and practice. As a first step in this direction, we highlight several under-explored mechanisms through which online IL can lead to improved sample complexity, including representational benefits and exploration (Appendix I). We also complement our theoretical results with empirical demonstrations of the phenomena we describe (Appendix C).

**Experiments.** In Appendix C (deferred to the appendix due to space constraints), we complement our theoretical results with an empirical demonstration of the horizon-independence of LogLossBC predicted by our theory (under parameter sharing and sparse rewards). We consider tasks where the horizon $H$ can be naturally scaled up and down—for example, an agent walking for a set number of timesteps—and use an expert trained according to RL to generate expert trajectories, before training a policy using LogLossBC. We consider both continuous action space (MuJoCo environment Walker2d) and discrete action space (Atari environment Beamrider) tasks to demonstrate the broad applicability of our theoretical results. As can be seen in Figure 1, the performance of the learned policy is independent or improving with horizon, consistent with our theoretical results. We also perform simplified experiments on autoregressive language generation with transformers. Here, we

find that the performance of the imitator is largely independent of $H$, as predicted by our results, though the results are more nuanced.

**Notation.** We use $\mathbb{I}_x \in \Delta(\mathcal{X})$ to denote the direct delta distribution, which places probability mass 1 on $x$. We adopt standard big-oh notation, and write $f = \widetilde{O}(g)$ to denote that $f = O(g \cdot \max\{1, \text{polylog}(g)\})$ and $a \lesssim b$ as shorthand for $a = O(b)$.

## 2 Horizon-Independent Analysis of Log-Loss Behavior Cloning

This section presents the first of our main results, a horizon-independent sample complexity analysis of log-loss behavior cloning for the case of deterministic experts. Our second main result, which handles the case of stochastic experts, builds on our results here and is presented in Section 3.

### 2.1 Log-Loss Behavior Cloning and Supervised Learning Guarantees

The workhorse for all of our results (both for deterministic and stochastic experts) is the following simple modification to behavior cloning. For a class of (potentially stochastic) policies $\Pi$, we minimize the *logarithmic loss*:

$$\widehat{\pi} = \arg\min_{\pi \in \Pi} \sum_{i=1}^{n} \sum_{h=1}^{H} \log\left(\frac{1}{\pi_h(a_h^i \mid x_h^i)}\right). \tag{4}$$

This scheme is ubiquitous in practice [45, 31], and forms the basis for autoregressive language modeling [64]; we refer to it as LogLossBC. We will show that this seemingly small change—moving from indicator loss to log loss—has significant benefits. Following the classical tradition of imitation learning [70, 72, 71], our analysis proceeds via *reduction* to supervised learning. We first show that LogLossBC satisfies an appropriate supervised learning guarantee, then translate this into rollout performance. Our starting point is to observe that LogLossBC, via Eq. (4), can be interpreted as performing maximum likelihood estimation over the set $\{\mathbb{P}^\pi\}_{\pi \in \Pi}$ in order to estimate the law $\mathbb{P}^{\pi^\star}$ over trajectories under $\pi^\star$ (see Appendix E.1 for details). As a result, standard guarantees for maximum likelihood estimation [89, 102] imply convergence in distribution whenever $\pi^\star \in \Pi$. To be precise, define the squared *Hellinger distance* for probability measures $\mathbb{P}$ and $\mathbb{Q}$ by $D_{\mathsf{H}}^2(\mathbb{P}, \mathbb{Q}) = \int \left(\sqrt{d\mathbb{P}} - \sqrt{d\mathbb{Q}}\right)^2$. Then for any finite policy class $\Pi$, we have the following guarantee.[5]

**Proposition 2.1** (Supervised learning guarantee for LogLossBC (special case of Theorem E.1))**.** *For any (potentially stochastic) expert $\pi^\star \in \Pi$, the* LogLossBC *algorithm ensures that with probability at least $1 - \delta, D_{\mathsf{H}}^2\left(\mathbb{P}^{\widehat{\pi}}, \mathbb{P}^{\pi^\star}\right) \leq 2\frac{\log(|\Pi|\delta^{-1})}{n}$.*

That is, by performing LogLossBC, we are implicitly estimating the law $\mathbb{P}^{\pi^\star}$; note that this result holds even if $\pi^\star$ is stochastic, as long as $\pi^\star \in \Pi$. We will focus on finite, realizable policy classes throughout this section to simplify presentation as much as possible, but guarantees for infinite classes under misspecification are given in Appendix E.1.

### 2.2 Horizon-Independent Analysis of LogLossBC for Deterministic Experts

We first consider the case where the expert $\pi^\star$ is deterministic. Our main result is the following theorem, which translates the supervised learning error $D_{\mathsf{H}}^2\left(\mathbb{P}^{\widehat{\pi}}, \mathbb{P}^{\pi^\star}\right)$ into a bound on rollout performance in a horizon-independent fashion.

**Theorem 2.1** (Horizon-independent regret decomposition (deterministic case))**.** *For any deterministic policy $\pi^\star$ and potentially stochastic policy $\widehat{\pi}$,*

$$J(\pi^\star) - J(\widehat{\pi}) \leq 4R \cdot D_{\mathsf{H}}^2\left(\mathbb{P}^{\widehat{\pi}}, \mathbb{P}^{\pi^\star}\right). \tag{5}$$

This result shows that horizon-independent bounds on rollout performance are possible whenever (i) rewards are appropriately normalized, and (ii) the supervised learning error $D_{\mathsf{H}}^2\left(\mathbb{P}^{\widehat{\pi}}, \mathbb{P}^{\pi^\star}\right)$ is appropriately controlled. It is proven using novel trajectory-level control over deviations between $\widehat{\pi}$ and $\pi^\star$; we will elaborate upon this in the sequel. We emphasize that this result would be trivial

---

[5]While unfamiliar readers might expect a bound on KL divergence, Hellinger distance turns out to be more natural due to a connection to the MGF of the log-loss [89, 102]. This facilitates scale-free generalization guarantees in spite of the potential unboundedness of the log-loss. .

if squared Hellinger distance were replaced by total variation distance in (5); that the bound scales with *squared* Hellinger distance is crucial for obtaining fast $1/n$-type rates and linear horizon dependence. We further remark that this reduction is not specific to LogLossBC, and can be applied to any IL algorithm for which we can bound the Hellinger distance. Combining Theorem 2.1 with Proposition 2.1, we obtain the following guarantee for finite policy classes.

**Corollary 2.1** (Regret of LogLossBC). *For any deterministic expert $\pi^\star \in \Pi$, the* LogLossBC *algorithm in Eq. (4) ensures that with probability at least $1 - \delta$, $J(\pi^\star) - J(\widehat{\pi}) \leq 8R \cdot \frac{\log(2|\Pi|\delta^{-1})}{n}$.*

To the best of our knowledge, this is the tightest available sample complexity guarantee for offline imitation learning with general policy classes. This bound improves upon the guarantee for indicator-loss behavior cloning in Eq. (2) by an $O(H)$ factor, and improves upon the guarantee for Dagger in Eq. (3) (replacing $H$ with $R \leq H$ under $r_h \in [0, 1]$) in the typical regime where $\mu = \Omega(1)$.

### 2.3 Interpreting the Sample Complexity of LogLossBC

To understand the behavior of the bound for LogLossBC in Corollary 2.1 in more detail, we consider two special cases (summarized in Table 1).

**Stationary policies and parameter sharing.** If $\log|\Pi| = O(1)$, the bound in Corollary 2.1 is *independent of horizon* in the case of sparse rewards ($R = O(1)$), and *linear in horizon* in the case of dense rewards ($R = O(H)$). In other words, our work establishes for the first time that:

$O(H)$ *sample complexity can be achieved in offline IL under dense rewards for general classes $\Pi$,*

as long as $\log|\Pi|$ is appropriately controlled and realizability holds. This runs somewhat counter to intuition expressed in prior work [70, 72, 71, 67–69, 80], but we will show in the sequel that there is no contradiction.

Generally speaking, we expect to have $\log|\Pi| = O(1)$ if $\Pi$ consists of stationary policies or more broadly, policies with parameter sharing across steps $h \in [H]$ (as is the case in transformers used for autoregressive text generation). As an example, for a tabular (finite state/action) MDP, if $\Pi$ consists of all stationary policies, we have $\log|\Pi| = |\mathcal{X}|\log|\mathcal{A}|$, so Corollary 2.1 gives $J(\pi^\star) - J(\widehat{\pi}) \lesssim \frac{R|\mathcal{X}|\log(|\mathcal{A}|\delta^{-1})}{n}$; that is, stationary policies can be learned with horizon-independent samples complexity under sparse rewards and linear dependence on horizon under dense rewards. Similar behavior holds for non-stationary policies with parameter sharing (e.g., log-linear policies of the form $\pi_h(a \mid x) \propto \exp(\langle \phi_h(x, a), \theta \rangle)$); see Appendix E.1 for details.

**Non-stationary policies or no parameter sharing.** For non-stationary policies or policies with no parameter sharing across steps $h$ (e.g., product classes where $\Pi = \Pi_1 \times \Pi_2 \cdots \times \Pi_H$), we expect $\log|\Pi| = O(H)$ (more generally, $D_{\mathsf{H}}^2(\mathbb{P}^{\widehat{\pi}}, \mathbb{P}^{\pi^\star}) = \widetilde{O}(H/n)$). For example, in a tabular MDP, if $\Pi$ consists of all non-stationary policies, we have $\log|\Pi| = H|\mathcal{X}|\log|\mathcal{A}|$. In this case, Corollary 2.1 gives linear dependence on horizon for sparse rewards ($J(\pi^\star) - J(\widehat{\pi}) \lesssim \frac{RH|\mathcal{X}|\log(|\mathcal{A}|\delta^{-1})}{n}$) and quadratic dependence on horizon for dense rewards ($J(\pi^\star) - J(\widehat{\pi}) \lesssim \frac{H^2|\mathcal{X}|\log(|\mathcal{A}|\delta^{-1})}{n}$). The latter bound is known to be optimal [67] for offline IL.

### 2.4 Optimality and Consequences for Online versus Offline Imitation Learning

We now investigate the optimality of Theorem 2.1 and discuss implications for online versus offline imitation learning, as well as connections to prior work. Our main result here shows that in the dense-reward regime where $r_h \in [0, 1]$ and $R = H$, Theorem 2.1 cannot be improved when $\log|\Pi| = O(1)$—even with online access, recoverability, and known dynamics.

**Theorem 2.2** (Lower bound for deterministic experts). *For any $n \in \mathbb{N}$ and $H \in \mathbb{N}$, there exists a (reward-free) MDP $M^\star$ with $|\mathcal{X}| = |\mathcal{A}| = 2$, a class of reward functions $\mathcal{R}$ with $|\mathcal{R}| = 2$, and a class of deterministic policies $\Pi$ with $|\Pi| = 2$ with the following property. For any (online or offline) imitation learning algorithm, there exists a deterministic reward function $r = \{r_h\}_{h=1}^H$ with $r_h \in [0, 1]$ (in particular, $R \leq H$) and (optimal) expert policy $\pi^\star \in \Pi$ with $\mu = 1$ such that $\mathbb{E}[J(\pi^\star) - J(\widehat{\pi})] \geq c \cdot \frac{H}{n}$ for an absolute constant $c > 0$. In addition, the dynamics, rewards, and expert policies are stationary.*

Together, Theorems 2.1 and 2.2 show that without further assumptions on $\Pi$, *online imitation learning cannot improve upon offline imitation learning* in the realizable setting. That is, even if recoverability

is satisfied, there is no online imitation learning algorithm that improves upon Theorem 2.1 uniformly for all policy classes. See Appendix H.1 for further lower bounds.

**Benefits of online IL for policies with no parameter sharing.** How can we reconcile our results with prior work showing that that online IL improves the horizon dependence of offline IL [70, 72, 71, 67–69, 80]? The important distinction here is that online IL can still improve on a *policy-class dependent* basis. In particular, methods like Dagger can still lead to improved sample complexity for policy classes with *no parameter sharing* across steps $h \in [H]$. Let $\Pi_h := \{\pi_h \mid \pi \in \Pi\}$ denote the projection of $\Pi$ onto step $h$. In Appendix E.2, we prove the following refined guarantee for a variant of Dagger based on the log-loss (LogLossDagger).

**Proposition 2.2** (Special case of Proposition E.2). *When $\pi^\star \in \Pi$ is deterministic,* LogLossDagger *ensures that with probability at least $1 - \delta$, $J(\pi^\star) - J(\widehat{\pi}) \lesssim \mu \cdot \sum_{h=1}^{H} \frac{\log(|\Pi_h|H\delta^{-1})}{n}$.*

For classes with no parameter sharing (i.e., product classes where $\Pi = \Pi_1 \times \Pi_2 \cdots \times \Pi_H$), we have $\sum_{h=1}^{H} \log|\Pi_h| = \log|\Pi|$. In this case, Proposition 2.2 scales as $J(\pi^\star) - J(\widehat{\pi}) \lesssim \mu \cdot \frac{\log(|\Pi|H\delta^{-1})}{n}$, improving on the bound for LogLossBC in Theorem 2.1 by replacing $R$ with $\mu \leq R$. Thus, online IL can indeed improve over offline IL for classes with no parameter sharing. This is consistent with Rajaraman et al. [67, 68], who proved a $\mu H$ vs. $H^2$ gap between online and offline IL for the special case of non-stationary tabular policies (where $\Pi$ is a product class with $\log|\Pi| \propto H$) under dense rewards. However, for classes with parameter sharing (i.e., where $\log|\Pi_h| \propto \log|\Pi|$), the bound in Proposition 2.2 scales as $\frac{\mu H \log|\Pi|}{n}$, which does not improve over Theorem 2.1 unless $\mu \ll 1$. Since virtually all empirical work on imitation learning uses parameter sharing across steps $h \in [H]$, we believe the finding that online IL does not improve over offline IL in this regime is quite salient. Nevertheless, it is important to emphasize that there are various practical considerations (e.g., misspecification or geometric structure) which this result may not account for.[6]

## 2.5 Proving Theorem 2.1: How Does LogLossBC Avoid Error Amplification?

The central object in the proof of Theorem 2.1 is the following *trajectory-level* distance function between policies. For a pair of potentially stochastic policies $\pi$ and $\pi'$, define

$$\rho(\pi \parallel \pi') := \mathbb{E}^\pi \mathbb{E}_{a'_{1:H} \sim \pi'(x_{1:H})}[\mathbb{I}\{\exists h : a_h \neq a'_h\}], \tag{6}$$

where we use the shorthand $a'_{1:H} \sim \pi'(x_{1:H})$ to indicate that $a'_1 \sim \pi'(x_1), \ldots, a'_H \sim \pi'(x_H)$. We begin by showing (Lemma F.2) that for all (potentially stochastic) policies $\pi^\star$ and $\widehat{\pi}$, $J(\pi^\star) - J(\widehat{\pi}) \leq R \cdot \rho(\pi^\star \parallel \widehat{\pi})$. We then show (Lemma F.3) that whenever $\pi^\star$ is deterministic, Hellinger distance satisfies[7] $D_{\mathsf{H}}^2\left(\mathbb{P}^{\widehat{\pi}}, \mathbb{P}^{\pi^\star}\right) \geq \frac{1}{4} \cdot \rho(\widehat{\pi} \parallel \pi^\star)$. Finally, we show (Lemma F.1) that the trajectory-level distance is symmetric, i.e. $\rho(\widehat{\pi} \parallel \pi^\star) = \rho(\pi^\star \parallel \widehat{\pi})$. This step is perhaps the most critical: by considering trajectory-level errors, we can switch from the state distribution induced by $\widehat{\pi}$ to that of $\pi^\star$ for free, without incurring error amplification or spurious horizon factors. Combining the preceding inequalities yields Theorem 2.1; see Appendix F for the full proof.

This analysis is closely related to a result in Rajaraman et al. [68]. For the special case of deterministic, linearly parameterized policies with parameter sharing, Rajaraman et al. [68] consider an algorithm that minimizes an empirical analogue of the trajectory-wise distance in Eq. (6), and show that it leads to a bound similar to Corollary 2.1 (i.e., linear-in-$H$ sample complexity under dense rewards). Relative to this work, our contributions are threefold: (i) we show that horizon-independent sample complexity can be achieved for *arbitrary* policy classes with parameter sharing, not just linear classes; (ii) we show that said guarantees can be achieved by a natural algorithm, LogLossBC, which is already widely used in practice; and (iii), by virtue of considering the log loss, our results readily generalize to encompass stochastic expert policies, as we will show in the sequel.

## 3 Horizon-Independent Analysis of LogLossBC for Stochastic Experts

In this section, we turn out attention to the general setting in which the expert policy $\pi^\star$ is stochastic. Stochastic policies are widely used in practice, where they are useful for modeling multimodal behavior [76, 25, 14], but have received relatively little exploration in theory beyond the work of Rajaraman

---

[6]Complementary to our results, various works show improved horizon dependence in the *inverse RL* setup where either (i) the MDP dynamics are known, or (ii) the learner can interact with the MDP online, but cannot interact with the expert itself [67, 79]; see Appendix B.

[7]In fact, the opposite direction of this inequality holds as well, up to an absolute constant.

et al. [67] for tabular policies.[8]   Our main result for this section is a regret decomposition based on the supervised learning error $D_{\mathsf{H}}^2\big(\mathbb{P}^{\widehat{\pi}}, \mathbb{P}^{\pi^\star}\big)$ that is horizon-independent and *variance-dependent* [107, 106, 93]. To state the result, define $\sigma_{\pi^\star}^2 := \sum_{h=1}^{H} \mathbb{E}^{\pi^\star}\big[(Q_h^{\pi^\star}(x_h, \pi_h^\star(x_h)) - Q_h^{\pi^\star}(x_h, a_h))^2\big]$ as the *variance* for the expert policy.

**Theorem 3.1** (Horizon-independent regret decomposition). *Assume $R \geq 1$. For any pair of (potentially stochastic) policies $\pi^\star$ and $\widehat{\pi}$ and any $\varepsilon \in (0, e^{-1})$,*

$$J(\pi^\star) - J(\widehat{\pi}) \leq \sqrt{6\sigma_{\pi^\star}^2 \cdot D_{\mathsf{H}}^2\big(\mathbb{P}^{\widehat{\pi}}, \mathbb{P}^{\pi^\star}\big)} + O\big(R \log(R\varepsilon^{-1})\big) \cdot D_{\mathsf{H}}^2\big(\mathbb{P}^{\widehat{\pi}}, \mathbb{P}^{\pi^\star}\big) + \varepsilon. \tag{7}$$

Applying this result with `LogLossBC` leads to the following guarantee.

**Corollary 3.1** (Regret of `LogLossBC`). *For any $\pi^\star \in \Pi$, the `LogLossBC` algorithm in Eq. (4) ensures that with prob. at least $1 - \delta$, $J(\pi^\star) - J(\widehat{\pi}) \leq O(1) \cdot \sqrt{\frac{\sigma_{\pi^\star}^2 \log(|\Pi|\delta^{-1})}{n}} + O(R \log(n)) \cdot \frac{\log(|\Pi|\delta^{-1})}{n}$.*

As we show, when the expert policy is stochastic, we can no longer hope for a "fast" $1/n$-type rate, and must instead settle for a "slow" $1/\sqrt{n}$-type rate. The slow term in Corollary 3.1 is controlled by the variance $\sigma_{\pi^\star}^2$ for the optimal policy. In particular, if $\pi^\star$ is deterministic, then $\sigma_{\pi^\star}^2 = 0$, and Corollary 3.1 recovers our bound for the deterministic setting in Corollary 2.1 up to a $\log(n)$ factor.

### 3.1 Horizon-Independence and Optimality for Stochastic Experts

To understand the dependence on horizon in Corollary 3.1, we restrict our attention to the "parameter sharing" case where $\log|\Pi| = O(1)$, and separately discuss the sparse and dense reward settings (results summarized in Table 2).

Consider the sparse reward setting where $R = O(1)$. Here, at first glance it would appear that the variance $\sigma_{\pi^\star}^2$ should scale with the horizon. Fortunately, this is not the case: The following result—via a law-of-total-variance-type argument [4]—implies that Corollary 3.1 is *fully horizon-independent*, with no explicit dependence on horizon when $R = O(1)$ and $\log|\Pi| = O(1)$. For a function $f(x_{1:H}, a_{1:H})$, let $\mathrm{Var}^\pi[f]$ denote the variance of $f$ under $(x_1, a_1), \ldots, (x_H, a_H) \sim \pi$.

**Proposition 3.1.** *We have that $\sigma_{\pi^\star}^2 \leq \mathrm{Var}^{\pi^\star}\big[\sum_{h=1}^{H} r_h\big] \leq R^2$.*

For the dense-reward regime where $R = H$, Proposition 3.1 gives $J(\pi^\star) - J(\widehat{\pi}) \lesssim H\sqrt{\log(|\Pi|)/n}$. This is somewhat disappointing, as we now require $\Omega(H^2)$ trajectories (quadratic sample complexity) to learn a non-trivial policy, even when $\log|\Pi| = O(1)$. We show now that this quadratic lower bound is qualitatively tight: the slow $H/\sqrt{n}$ rate for $\sigma_{\pi^\star}^2 = H^2$ is necessary in both offline and online IL. This reveals a fundamental difference between deterministic and stochastic experts, since $O(H)$ sample complexity is sufficient in the former case.

**Theorem 3.2** (informal). *For any $\sigma^2 \in [H, H^2]$, there exists $\Pi$ with $|\Pi| = 2$ such that $\sigma_{\pi^\star}^2 \leq \sigma^2$ and any (offline or online) IL algorithm must have $J(\pi^\star) - J(\widehat{\pi}) \gtrsim \sqrt{\frac{\sigma^2}{n}}$ with constant probability.*

Nonetheless, it is possible to obtain linear-in-$H$ sample complexity for dense rewards under a recoverability-like condition. Let us define the *signed recoverability constant* via $\widetilde{\mu} = \max_{x \in \mathcal{X}, a \in \mathcal{A}, h \in [H]} |(Q_h^{\pi^\star}(x, \pi_h^\star(x)) - Q_h^{\pi^\star}(x, a)|$. Note that $\widetilde{\mu} \in [0, R]$, and that $\widetilde{\mu} \geq \mu$, since this version counts actions $a$ that *outperform* $\pi^\star$, not just those that underperform. It is immediate to see that $\sigma_{\pi^\star}^2 \leq \widetilde{\mu}^2 H$. Hence, even if $R = H$, as long as $\widetilde{\mu} = O(1)$, Corollary 3.1 yields $J(\pi^\star) - J(\widehat{\pi}) \lesssim \sqrt{H \log(|\Pi|)/n}$, so that $O\big(\frac{H \log|\Pi|}{\varepsilon^2}\big)$ trajectories suffice to learn an $\varepsilon$-optimal policy.[9]

See Appendix H for further results concerning tightness of Theorem 3.1, including instance-dependent lower bounds.

---

[8]As discussed at length in Rajaraman et al. [67], many prior works [70, 72] state results in a level of generality that allows for stochastic experts, but the notions of supervised learning error found in these works (e.g., TV distance) do not lead to tight rates when instantiated for stochastic experts.

[9]An interesting question for future work is to understand if a similar conclusion holds if we replace $\widetilde{\mu}$ with $\mu$.

**Consequences for online versus offline IL.** The lower bound in Theorem G.1 holds even for online imitation learning algorithms. Thus, similar to the deterministic setting, there is no online IL algorithm that improves upon Theorem 3.1 uniformly for all policy classes. This means that even for stochastic experts, online imitation learning cannot improve upon offline imitation learning in the realizable setting without further assumptions (e.g., no parameter sharing) on the policy class under consideration.

**Proof sketch for Theorem 3.1.** When the expert is stochastic, the trajectory-wise distance in Eq. (6), is no longer useful (i.e., $\rho(\pi^\star \| \pi^\star) \neq 0$), which necessitates a more information-theoretic analysis. Our starting point is the following scale-sensitive change-of-measure lemma for Hellinger distance.

**Lemma 3.1** (Change-of-measure for Hellinger distance [34, 35]). *Let $\mathbb{P}$ and $\mathbb{Q}$ be probability distributions over a measurable space $(\mathcal{X}, \mathscr{F})$. Then for all functions $h : \mathcal{X} \to \mathbb{R}$,*

$$|\mathbb{E}_{\mathbb{P}}[h(X)] - \mathbb{E}_{\mathbb{Q}}[h(X)]| \leq \sqrt{\tfrac{1}{2}(\mathbb{E}_{\mathbb{P}}[h^2(X)] + \mathbb{E}_{\mathbb{Q}}[h^2(X)]) \cdot D_{\mathsf{H}}^2(\mathbb{P}, \mathbb{Q})}. \tag{8}$$

*In particular, if $h \in [0, R]$ almost surely, then $\mathbb{E}_{\mathbb{P}}[h(X)] \leq 2\,\mathbb{E}_{\mathbb{Q}}[h(X)] + R \cdot D_{\mathsf{H}}^2(\mathbb{P}, \mathbb{Q})$.*

We sketch how to use Lemma 3.1 to prove a weaker version of Theorem 3.1, and defer the full proof, which builds on this argument, to Appendix G.1. Define the *sum of advantages* for a trajectory $o = (x_1, a_1), \ldots, (x_H, a_H)$ via $\Delta(o) = \sum_{h=1}^{H} Q_h^{\pi^\star}(x_h, \pi_h^\star(x_h)) - Q_h^{\pi^\star}(x_h, a_h)$. By the performance difference lemma, we can write $J(\pi^\star) - J(\widehat{\pi}) = \mathbb{E}^{\widehat{\pi}}[\Delta(o)]$, so applying Eq. (8) yields

$$J(\pi^\star) - J(\widehat{\pi}) = \mathbb{E}^{\widehat{\pi}}[\Delta(o)] \lesssim \mathbb{E}^{\pi^\star}[\Delta(o)] + \sqrt{(\mathbb{E}^{\widehat{\pi}}[\Delta^2(o)] + \mathbb{E}^{\pi^\star}[\Delta^2(o)]) \cdot D_{\mathsf{H}}^2(\mathbb{P}^{\widehat{\pi}}, \mathbb{P}^{\pi^\star})}.$$

From here, we observe that $\mathbb{E}^{\pi^\star}[\Delta(o)] = 0$ and $\mathbb{E}^{\pi^\star}[\Delta^2(o)] = \sigma_{\pi^\star}^2$ (this follows because advantages are a martingale difference sequence under $\mathbb{P}^{\pi^\star}$), so all that remains is to bound the term $\mathbb{E}^{\widehat{\pi}}[\Delta^2(o)]$. A crude approach is to observe that $|\Delta(o)| \leq \widetilde{\mu}H$, so that applying Lemma 3.1 gives $\lesssim \mathbb{E}^{\pi^\star}[\Delta^2(o)] + (\widetilde{\mu}H)^2 \cdot D_{\mathsf{H}}^2(\mathbb{P}^{\widehat{\pi}}, \mathbb{P}^{\pi^\star})$, and consequently $J(\pi^\star) - J(\widehat{\pi}) \lesssim \sqrt{\sigma_{\pi^\star}^2 \cdot D_{\mathsf{H}}^2(\mathbb{P}^{\widehat{\pi}}, \mathbb{P}^{\pi^\star})} + \widetilde{\mu}H \cdot D_{\mathsf{H}}^2(\mathbb{P}^{\widehat{\pi}}, \mathbb{P}^{\pi^\star})$. This falls short of Eq. (9) due to the suboptimal lower-order term, which does not recover Theorem 2.1 when $\pi^\star$ is deterministic ($\sigma_{\pi^\star}^2 = 0$). The full proof in Appendix G.1 corrects this disparity using a subtle and significantly more involved argument based on stopping times and martingale concentration.

## 4 Discussion and Additional Results

Our results clarify the role of horizon in offline and online IL, and show that—at least under standard theoretical assumptions—the gap between online and offline IL is smaller than previously thought.

**Benefits of online interaction** Instabilities of offline IL [60, 27, 13] and benefits of online IL [73, 52, 38, 6, 26, 51, 7, 108, 59] likely arise in practice, but existing assumptions in theoretical research on imitation learning appear be too coarse to give insights into the true nature of these phenomena. Toward developing a fine-grained, problem-dependent understanding of algorithms and complexity for IL, in Appendix I, we highlight several special cases in which online interaction *does* lead to benefits over offline imitation learning, but in a policy class-dependent fashion not captured by existing theory. We identify three phenomena that can lead to improved sample complexity: (i) *representational benefits*; (ii) *value-based feedback*; and (iii) *exploration*.

**Further directions.** Additional directions for future research include (i) developing and analyzing imitation learning algorithms under control-theoretic assumptions that more directly capture practical notions of instability [61, 88, 13, 14], and (ii) developing a more refined theory in the context of language models, via the connection in Appendix B.3. For both settings, an important question is to understand whether the notion of supervised learning error $D_{\mathsf{H}}^2(\mathbb{P}^{\widehat{\pi}}, \mathbb{P}^{\pi^\star})$ we consider is a suitable proxy for real-world performance, or whether more refined notions are required.

**Additional results.** Secondary results deferred to the appendix for space include (i) examples and additional guarantees for `LogLossBC` and `LogLossDagger` (Appendix E); and (ii) additional lower bounds and results concerning the tightness of Theorems 2.2 and 3.1 (Appendix H).

**Acknowledgements**

We thank Jordan Ash, Audrey Huang, Akshay Krishnamurthy, Max Simchowitz, and Cyril Zhang for many helpful discussions. We thank Drew Bagnell for valuable comments and pointers to related work.

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

# Contents of Appendix

# A  Omitted Tables

| | Worst-case | Low-noise | $\widetilde{\mu}$-recoverable |
|---|---|---|---|
| Sparse Rewards | $\widetilde{O}\left(R\sqrt{\frac{\log(|\Pi|)}{n}}\right)$ | $\widetilde{O}\left(\sqrt{\frac{\sigma_{\pi^\star}^2 \log(|\Pi|)}{n}} + \frac{R\log(|\Pi|)}{n}\right)$ | N/A |
| Dense Rewards | $\widetilde{O}\left(H\sqrt{\frac{\log(|\Pi|)}{n}}\right)$ | $\widetilde{O}\left(\sqrt{\frac{\sigma_{\pi^\star}^2 \log(|\Pi|)}{n}} + \frac{H\log(|\Pi|)}{n}\right)$ | $\widetilde{O}\left(\widetilde{\mu}\sqrt{\frac{H\log(|\Pi|)}{n}} + \frac{H\log(|\Pi|)}{n}\right)$ |

Table 2: Summary of upper bounds for stochastic experts (Corollary 3.1). Each cell denotes the expected regret of a policy learned with LogLossBC; lower bounds are more nuanced and discussed in Section 3. Here $\Pi$ is the policy class, $R$ is the cumulative reward range, $H$ is the horizon, $n$ is the number of expert trajectories, $\sigma_{\pi^\star}^2$ is the variance of the expert policy , and $\widetilde{\mu}$ is the signed recoverability parameter ; see Section 3 for definitions.[10]

# B  Additional Related Work

## B.1  Theory of Imitation Learning and Reinforcement Learning

Classical theoretical works in imitation learning, beginning from the work of Ross and Bagnell [70] observes that behavior cloning (for the specific indicator loss in Section 1.1) can incur quadratic dependence on horizon, and shows that online interaction, via algorithms like Dagger and Aggrevate, can obtain improved sample complexity under recoverability-type conditions [70, 72, 71, 78]. Further works along this line include Cheng and Boots [22], Cheng et al. [24, 23], Yan et al. [99], Spencer et al. [77].

These papers can be thought of as *supervised learning reduction*, in the sense that—in the vein of Eq. (1)—they guarantee that the imitation learning performance is controlled by an appropriate notion of supervised learning performance. Notably, this holds for *any policy* $\widehat{\pi}$, which means that in practice, the rollout performance is good whenever supervised learning succeeds, even if we do not necessarily have a provable guarantee for the generalization of $\widehat{\pi}$ (e.g., for neural networks, where understanding generalization is an active area of research). However, as noted throughout this paper and elsewhere [67–69], these works typically state regret guarantees in terms of different, often incomparable notions of supervised learning performance, and avoid giving concrete, end-to-end guarantees for specific policy classes of interest. This can make it challenging to objectively evaluate optimality, and to understand whether limitations of specific algorithms are due to suboptimal design choices versus information-theoretic limitations. For example, Li and Zhang [56] show that in some cases, supervised learning oracles that satisfy assumptions required by prior work do not actually exist.

**Minimax sample complexity of imitation learning.** More recently, a line of work beginning with Rajaraman et al. [67] revisits the minimax sample complexity of imitation learning, aiming to provide end-to-end sample complexity guarantees and lower bounds, but primarily focused on tabular MDPs and policies [67–69, 80]. Notably, Rajaraman et al. [67] show that when $\Pi$ is the set of all non-stationary policies in a tabular MDP and $R = H$, online IL methods can achieve $O(\mu H)$ sample complexity, while offline IL methods must pay $\Omega(H^2)$; this is consistent with our findings in Section 2, as $\log|\Pi| = \Omega(H)$ for this setting. Other interesting findings from this line of work include the observation that when the MDP dynamics are *known*, the sample complexity for offline IL with non-stationary tabular policies can be brought down to $O(H^{3/2})$. As noted in Section 2, Rajaraman et al. [68] show that offline IL methods can obtain $O(H)$ sample complexity for *linearly parameterized* policies under parameter sharing; our analysis of LogLossBC for the special case of deterministic policies shows that it can be viewed as *implicitly* minimizing the objective they consider.

Xu et al. [98] also consider the problem of horizon independence in IL. Their work focuses on tabular MDPs and policies, and shows with knowledge of the dynamics, it is possible to achieve horizon dependence for a restricted class of MDPs termed *RBAS-MDPs*. In contrast, our work achieves horizon independence for general MDPs, without knowledge of dynamics.

Compared to the works above, we focus on general finite classes $\Pi$. Various works on theoretical reinforcement learning [2, 33] have observed that finite classes are a useful test case for general function approximation, because they are arguably the simplest type of policy class from a generaliza-

tion perspective, yet do not have any additional structure (e.g., linearity) that could lead to spurious conclusions that do not extend to rich function classes like neural networks.

Recent work of Tiapkin et al. [84] provides generalization guarantees for behavior cloning with the logarithmic loss, but their results scale linearly with the horizon, and thus cannot give tight guarantees for policy classes with parameter sharing. In addition, their results are stated in terms of KL-divergence and, as a consequence, require a lower bound on the action densities for the policy class under consideration. We expect that both of these limitations are inherent to KL divergence. Tiapkin et al. [84] also give variance-dependent bounds on rollout performance similar to Theorem 3.1, but their results require a bound on KL divergence (which is stronger than a bound on Hellinger distance), and thus are unlikely to meaningfully capture optimal horizon dependence. These bounds on rollout performance also do not recover the notion of variance in Theorem 3.1.

We also mention in passing Sekhari et al. [75], who consider *active* imitation learning algorithms, and focus on obtaining improved sample complexity with respect to dependence on the accuracy $\varepsilon$ (as opposed to $H$), under strong distributional assumptions in the vein of active learning [40].

**Inverse reinforcement learning.** A long line of research on *inverse reinforcement learning* and related techniques considers a setting in which either a) the dynamics of the MDP $M^\star$ are known, or b) it is possible to interact with $M^\star$ online (without expert feedback), with empirical [1, 109] and theoretical results [81, 83, 82, 16, 20]. This setting encompasses generative adversarial imitation learning and related moment matching methods [42, 57, 49, 79]. A detailed discussion is out of scope for the present work, but we believe this framework can improve over the sample complexity of offline IL in some but not all situations (e.g., Rajaraman et al. [67]).

**Benefits of logarithmic loss.** Our work draws inspiration from Foster and Krishnamurthy [32], who observed that the logarithmic loss can have benefits over square loss when outcomes are heteroskedastic, and used this observation to derive first-order regret bounds for contextual bandits. Subsequent works have extended their analysis technicals to derive first-order regret bounds in various reinforcement learning settings [92, 93, 3].[11] To the best of our knowledge, our work is the first to uncover a decision making setting in which switching to the logarithmic loss is beneficial even in a minimax sense. We emphasize that while our analysis uses the information-theoretic machinery introduced in Foster and Krishnamurthy [32] and related work [34, 35], our results are quite specialized to structure of the imitation learning setting, and cannot directly be derived from any of the results in Foster and Krishnamurthy [32], Wang et al. [92, 93], Ayoub et al. [3].

**Horizon-free reinforcement learning.** Our results also take inspiration from the line of research on *horizon-independent* sample complexity bounds for reinforcement learning [47, 101, 94, 104, 105], as well as a closely related line of research on variance-dependent regret bounds [107, 106, 93].[12] These papers provide sample complexity bounds for reinforcement learning that have little or no explicit dependence on horizon whenever rewards are normalized such that $\sum_{h=1}^{H} r_h \in [0, 1]$. We consider a simpler setting (imitation learning), but provide guarantees that hold under *general function approximation*, while the works above are restricted to either tabular MDPs or MDPs with linear/low-rank structure. Nonetheless, our proof of Theorem 3.1 makes use of concentration arguments inspired by Zhang et al. [104, 105].

### B.2 Empirical Research on Imitation Learning

Many empirical works have observed compounding error in behavior cloning. Outside of online imitation learning, mitigations include noise injection at data collection time [54, 50] or inverse RL methods that assume knowledge of system dynamics [109]. Other works take a control-theoretic perspective [88, 41, 61, 14], and augment behavior cloning with techniques designed to ensure incremental stability (or other control-theoretic notions of stability) of system.

**Online imitation learning.** Many empirical works have noted benefits of online imitation learning methods like Dagger over classical behavior cloning [73, 52, 38, 6, 26, 51, 7, 108, 59]. These results are not in contradiction to our findings, as they typically do not ablate the effect of the loss function

---

[11]We also mention in passing the work of Farebrother et al. [29], which observes that switching to the log-loss is beneficial empirically for approximate value iteration methods in offline reinforcement learning.

[12]Compared to variance-dependent bounds for RL in Zhou et al. [107], Zhao et al. [106], Wang et al. [93] an interesting feature of Theorem 3.1 is that it only depends on variance for $\pi^\star$, whereas these works typically depend on worst-case variance over all policies or similar quantities.

(e.g., [70] uses the squared hinge loss, Ross et al. [72] uses the hinge loss, and Ross et al. [73] uses the square loss). It is also possible that the perceived benefits arise from factors beyond horizon (e.g., representational benefits), as discussed in Appendix I.

### B.3 Autoregressive Language Modeling

Autoregressive language modeling with the standard next-token prediction objective [64] can be viewed as an instance of behavior cloning with the logarithmic loss. In this setting, $M^\star$ corresponds to a *token-level MDP*. Here $\mathcal{A}$ is a space or vocabulary of *tokens* The initial state is $x_1 = z \sim P_0$, where $z$ is a *prompt* or *context*. Given the prompt, for each $h = 1, \ldots, H$ the action $a_h \in \mathcal{A}$ is a new token, which is concatenated to the state via the deterministic dynamics $x_{h+1} \leftarrow (z, a_{1:h})$. Via Bayes' rule, an expert policy

$$\pi^\star(a_{1:H} \mid z) = \prod_{h=1}^{H} \pi_h^\star(a_h \mid z, a_{1:h-1}) = \prod_{h=1}^{H} \pi_h^\star(a_h \mid x_h)$$

can represent an arbitrary conditional distribution over sequences, from which a training set $\mathcal{D} = \{o^i\}$ with $o^i = (z^i, a_1^i, \ldots, a_H^i)$ is generated. With this setup, log-loss behavior cloning

$$\widehat{\pi} = \arg\max_{\pi \in \Pi} \sum_{i=1}^{n} \sum_{h=1}^{H} \log(\pi_h(a_h^i \mid z^i, a_{1:h-1}^i))$$

is equivalent to the standard next-token prediction objective for unsupervised language model pre-training [64], with the class $\Pi$ parameterized by a transformer or a similar neural net architecture. In this context, long-range error amplification arising from the next-token prediction objective (often referred to as *exposure bias*) has been widely observed by prior work [43, 17, 13], and in some cases speculated to be a fundamental limitation [55, 5].

**Applying our results.** To apply our results, consider a fixed reward function $r = \{r_h\}_{h=1}^{H}$, which might measure performance for a particular task of interest (e.g., question answering or commonsense reasoning). Then, for a model $\pi$, $J(\pi)$ corresponds to rollout performance at the task for an autoregressively generated sequence (i.e., given $z \sim P_0$, we sample $a_h \sim \pi_h(\cdot \mid z, a_{1:h-1})$ for all $h \in [H]$). For this setting, Theorem 3.1 states that

$$J(\pi^\star) - J(\widehat{\pi}) \leq \widetilde{O}\left( \sqrt{\sigma_{\pi^\star}^2 \cdot D_{\mathsf{H}}^2(\mathbb{P}^{\widehat{\pi}}, \mathbb{P}^{\pi^\star})} + R \cdot D_{\mathsf{H}}^2(\mathbb{P}^{\widehat{\pi}}, \mathbb{P}^{\pi^\star}) \right), \tag{9}$$

where $\sigma_{\pi^\star}^2 = \sum_{h=1}^{H} \mathbb{E}^{\pi^\star}\left[ (Q_h^{\pi^\star}(z, a_{1:h}) - V_h^{\pi^\star}(z, a_{1:h-1}))^2 \right]$. In particular, as long as the cumulative reward for the task is bounded by $R = O(1)$ (e.g., if we receive an episode-level reward $r_H = 1$ if a question is answered correctly, and receive zero reward otherwise), the rollout performance has *no explicit dependence on the sequence length*, except through the generalization error $D_{\mathsf{H}}^2(\mathbb{P}^{\widehat{\pi}}, \mathbb{P}^{\pi^\star})$. In light of this result, we expect that error amplification observed in practice may arise from challenges in minimizing the generalization error $D_{\mathsf{H}}^2(\mathbb{P}^{\widehat{\pi}}, \mathbb{P}^{\pi^\star})$ itself (e.g., architecture, data generation process, optimization [17, 13]), rather than fundamental limits of next-token prediction.

## C Experiments

In this section, we validate our theoretical results empirically. We first provide a detailed overview of our experimental setup, including the control and natural language tasks we consider, then present empirical results for each task individually. We ran all of our experiments on NVidia V100 GPUs. Training time for each experiment varied by environment, but all were less than 6 hours.

### C.1 Experimental Setup

We evaluate the effect of horizon on the performance of `LogLossBC` in three environments. We begin by describing our training and evaluation protocol (which is agnostic to the environment under consideration), then provide details for each environment.

In each experiment, we begin with an expert policy $\pi^\star$ (which is always a neural network; details below) and construct an offline dataset by rolling out with it $n$ times for $H$ timesteps per episode. To train the imitator policy $\widehat{\pi}$, we use the same architecture as the expert, but randomly initialize the weights and use stochastic gradient descent with the Adam optimizer to minimize the `LogLossBC`

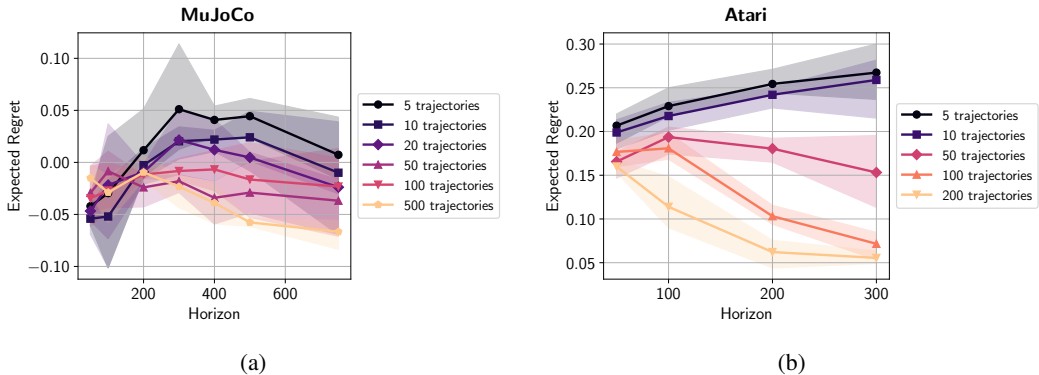

Figure 2: Dependence of expected regret on the horizon for multiple choices for the number of imitator trajectories $n$. **(a)** Continuous control environment Walker2d-v4. **(b)** Discrete Atari environment BeamriderNoFrameskip-v4. For both environments, increasing the horizon does not lead to a significant increase in regret, as predicted by our theory.

objective for the offline dataset; this setup ensures that the realizability assumption used by our main results (Assumption 1.1) is satisfied. We repeat this entire process for varying values of $H$.

To evaluate the regret $J(\pi^\star) - J(\widehat{\pi})$ after training, we approximate the average reward of the imitator policy $\widehat{\pi}$ by selecting new random seeds and collecting $n$ trajectories of length $H$ by rolling out with $\widehat{\pi}$; we approximate the average reward of the expert $\pi^\star$ in the same fashion, and we also compute several auxiliary performance measures (details below) that aim to capture the distance between $\widehat{\pi}$ and $\pi^\star$. In all environments, we normalize rewards so that the average reward of the expert is at most 1, in order to bring us to the sparse reward setting in Section 1.1 and keep the range of the possible rewards constant as a function of the (varying) horizon.

We consider four diverse environments, with the aim of evaluating LogLossBC in qualitatively different domains: (i) Walker2d, a classical continuous control task from MuJoCo [86, 85] where the learner attempts to make a stick figure-like agent walk to the right by controlling its joints; (ii) Beamrider, a standard discrete-action RL task from the Atari suite [9], where the learner attempts to play the game of Beamrider; (iii) Car, a top-down discrete car racing environment where the car has to avoid obstacles to reach a goal, and (iv) Dyck, an autoregressive language generation task where the agent is given a sequence of brackets in $\{\{,\},[,],(,)\}$ and has to close all open brackets in the correct order.

We emphasize diversity in task selection in order to demonstrate the generality of our results, covering discrete and continuous actions spaces, as well as both control and language generation. For some of the environment (Walker2d, Beamrider), the task is intended to be "stateless", in the sense that varying the horizon $H$ does not change the difficulty of the task itself (e.g., complexity of the expert policy $\pi^\star$), allowing for an honest evaluation of the difficulty of the *learning* problem as we vary the horizon $H$. For other domains, such as Dyck, horizon dependence is more nuanced, as here the capacity required to represent the expert grows as the horizon increases; this manifests itself in our theoretical results through the realizability condition (Assumption 1.1), which necessitates a more complex function class $\Pi$ as $H$ increases.

We now provide details for our experimental setup for each environment.

**Walker2d.** We use the Gymnasium [86] environment Walker2d-v4, which has continuous state and action spaces of dimensions 17 and 6 respectively. The agent is rewarded for moving to the right and staying alive as well as being penalized for excessively forceful actions; because we vary the horizon $H$, in order to make the comparison fair, we normalize the rewards so that our trained expert always has average reward 1. Our expert is a depth-2 MLP with width 64. We use the Stable-Baselines3 [66] implementation of the Proximal Policy Optimization (PPO) algorithm [74] with default settings to train the expert for 500K steps. The policy's action distribution is Gaussian, with the mean and covariance determined by the MLP; we use this for computation of the logarithmic loss. For data collection, we enforce a *deterministic* expert by always playing the mean of the Gaussian distribution produced by their policy. Our imitator policy uses the same architecture as the expert policy, with the

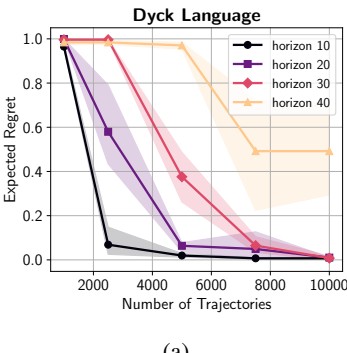
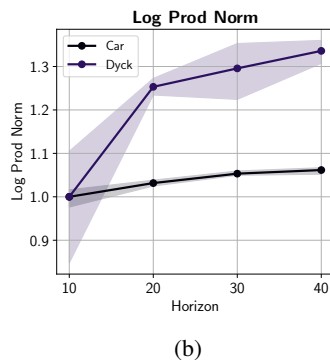

|     |     |
| :-: | :-: |
| (a) | (b) |

Figure 3: **(a)** Relationship between the number of expert trajectories and expected regret for the Dyck environment multiple choices of horizon $H$. The expert is trained to produce valid Dyck words of length $H$, and the imitator's ability to generate a valid word is evaluated. We find that regret increases as a function of $H$. **(b)** Logarithm of the product of weight matrix norms for the expert policy network as a function of $H$, for Dyck and Car environments. The log-product-norm acts as a proxy for complexity for the class $\Pi$; we rescale such that log-product-norm at $H = 10$ is 1.0 for both domains. For Dyck, we find that as $H$ increases, the complexity of $\Pi$ required to represent the expert policy (as measured by the log-product-norm) also increases, explaining the increasing regret in (a). However, the gain in log-product-norm for the Car domain is much lower, which is in line with the fact that the regret for the Car domain exhibits only mild scaling with horizon.

weights re-initialized randomly. We train the imitator using the logarithmic loss by default, but as an ablation, we also evaluate the effect of training with the mean-squared-error loss on the Euclidean norm over the actions. We train using the Adam optimizer [53] with a learning rate of $10^{-3}$ and a batch size of 128. We stop training early based on the validation loss on a held out set of expert trajectories. Note that the expert and imitator policies above are both *stationary policies*.

**Beamrider.** We use the Gymnasium environment BeamRiderNoFrameskip-v4, which has 9 discrete actions and a 210x160x3 image as the state; the rewards are computed as a function of how many enemies are destroyed. As in the case of the previous setup, we account for the varying of $H$ by normalizing expert rewards to be 1. Here we do not train our expert ourselves, but instead use the trained PPO agent provided by Raffin [65], which is a convolutional neural network. We use the same architecture for our imitator policy, with the weights re-initialized randomly. Here, the expert (and imitator) policies map the observation to a point on the probability simplex over actions, and so logarithmic loss computation is immediate. Similar to the case of Walker2d, we enforce a deterministic expert for collecting trajectories by taking the action with maximal probability. We then train our imitators using the same setup as in the Walker2d environment. As with Walker2d, the expert and imitator here are both stationary policies.

**Car.** We introduce a simple top-down navigation task where the agent is a "car" that always moves forward by one step, but can take actions to move left, right, or remain in its lane to avoid obstacles and reach the desired destination. There are $M$ possible lanes. At timestep $h \in [H + 1]$, if the agent is in lane $i \in [M]$, then the agent's state is $(i, h)$. We view the state space as a $M \times (H + 1)$ grid; a given point $(i, j)$ in the grid can be empty, or contain an obstacle, or contain the agent. The agent's action space consists of 3 possible actions: stay in the current lane ($(i, h) \mapsto (i, h + 1)$), move one step left ($(i, h) \mapsto (i - 1, h + 1)$), or move one step right ($(i, h) \mapsto (i + 1, h + 1)$). If the agent's action causes it to collide with an obstacle or the boundary of the grid, it is sent to an absorbing state. The agent gets a reward of 1 for reaching the goal state for the first time, and a reward of 0 otherwise. When the agent occupies a state $(i, h)$, it observes an image-based observation $x_h$ showing the state of all lanes for $V$ steps ahead where $V$ is the size of the viewing field. At the start of each episode, we randomly sample obstacles positions, the start position, and the goal position. The goal can be reached after $H$ actions, and it is always possible to reach the goal.

**Dyck.** In addition to the RL environments above, we evaluate LogLossBC for autoregressive language generation with transformers (cf. Appendix B.3), where the goal of the "agent" is to complete a valid word of a given length in a Dyck language; this has emerged as a popular sandbox for

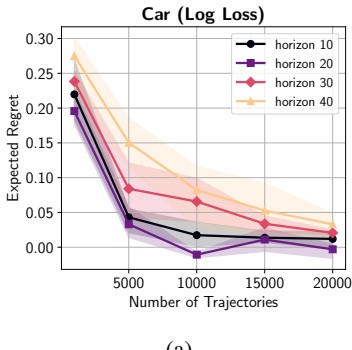
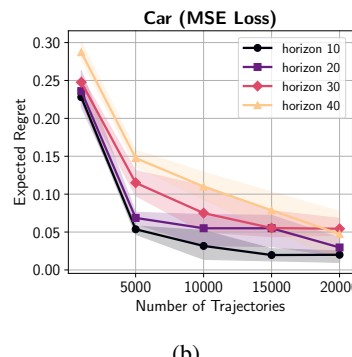

| (a) | (b) |

Figure 4: Dependence of expected regret on the number of expert trajectories for Car environment under varying values for horizon $H$ for log-loss (**a**) and mean-squared loss (**b**). The expert policy network is trained on a set of $2 \times 10^4$ episodes generated by an optimal policy via behavior cloning. We use LogLossBC to train imitator policy for varying values of the horizon $H$ and number of trajectories $n$. For both losses, we find that the expected regret goes down as the number of expert trajectories increases, but degrades slightly as a function of $H$.

understanding the nuances of autoregressive text generation in theory [100, 39, 12] and empirically [58, 95]. We recall that a Dyck language $\text{Dyck}_k$ consists of $2k$ matched symbols thought of as open and closed parentheses, with concatenations being valid words if the parentheses are closed in the correct order. For example, if we define the space of characters as '()', '[]', and '{}', then '([()]){}' is a valid word, whereas '([)]' and '((({}' are not.

Our experiments use the Dyck language $\text{Dyck}_3$. For our expert, we train an instance of GPT-2 small [64] with 6 layers, 3 heads, and 96 hidden dimensions from scratch to produce valid Dyck words. In particular, the training dataset consists of random Dyck prefixes that require exactly $H$ actions (symbols) to complete. To imitate this expert, we train a GPT-2 small model with the same architecture, but with randomly initialized weights on an offline dataset of sequences generated by the expert. We assign a reward 1 to each trajectory if the generated word is valid, and assign reward 0 otherwise. We use Adam optimization for training, with our experts trained for 40K iterations in order to ensure their quality. Note that in this environment, the expert and imitator policies are non-stationary, but use *parameter sharing* via the transformer architecture.

### C.2 Results

We summarize our main findings below.

**Effect of horizon on regret.** Figures 1 and 2 plot the relationship between expected regret and the number of expert trajectories for the Walker2d (MuJoCo), and BeamriderNoFrameskip (Atari) environments, as the horizon $H$ is varied from $50$ to $500$. For both environments, we find regret is largely independent of the horizon, consistent with our theoretical results. In fact, in the case of BeamriderNoFrameskip, we find that increasing the horizon leads to *better* regret. To understand this, note that our theory provides horizon-agnostic *upper bounds* independent of the environment. Our lower bounds are constructed for specific worst-case environments, and not rule out the possibility of improved performance with longer horizons environments with favorable structure. We conjecture that this phenomenon is related to the fact that longer horizons yield fundamentally more data, as the total number of state-action pairs in the expert dataset is equal to $nH$.[13]

Figure 3(a) plots our findings for the Dyck environment. Here, we see that with the number of trajectories $n$ fixed, regret does increase with $H$, which might appear to contradict our theory at first glance. However, we note that the policy class itself must become larger as $H$ increases, as the task itself becomes more difficult (equivalently, the supervised learning error $D_H^2\left(\mathbb{P}^{\pi^\star}, \mathbb{P}^{\widehat{\pi}}\right)$ must grow with $H$). As a result, the regret is not expected to be independent of $H$ for this environment, in spit of parameter sharing. To verify whether supervised learning error is indeed the cause for horizon

---

[13]For example, if we repeat a fixed contextual bandit instance $H$ times across the horizon and train a stationary policy, it is clear that regret should decrease with $H$ under sparse rewards. Less trivial instances where increasing horizon provably leads to better performance are known in some special cases [87].

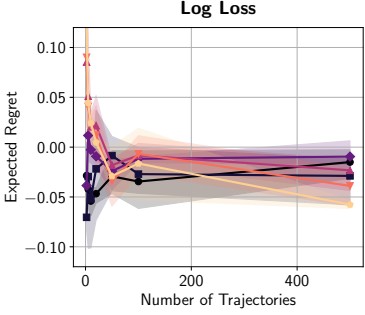 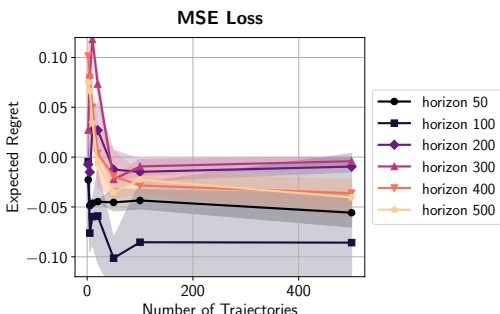

Figure 5: Dependence of expected regret on the number of expert trajectories for continuous control environment Walker2d-v4 under varying choices for horizon $H$. **(Left)** Behavior cloning with logarithmic loss (LogLossBC); **(Right)** Behavior cloning with mean squared error (MSE) Loss. Both losses lead to similar performance for this environment, possibly due to Gaussian policy parameterization.

dependence for Dyck, Figure 3(b) plots the logarithm of the product of the Frobenius norms of the weight matrices of the expert for varying values of $H$, as a proxy for supervised learning performance [8, 37].[14] We find that the log-product-norms do in fact grow with $H$, consistent with the fact that the regret grows with $H$ in this case.

For the Car environment, we observe similar behavior to the Dyck environment, visualized in Figure 4. We find that performance degrades slightly as a function of the horizon $H$, but that this increase in regret can be explained by an increase in the log-product-norm (Figure 3(b)). However, the effect is mild compared to Dyck.

**Comparison between log loss and square loss.** As an ablation, Figures 4 and 5 compare LogLossBC to the original behavior cloning objective of Pomerleau [63], which uses the mean squared error (MSE) to regress expert actions to observations in the offline dataset. Focusing on the Walker2d environment (Figure 5) and Car environment (Figure 4) (other environments presented difficulties in training[15]), we find that performance with the MSE loss is comparable to that of the logarithmic loss. For Walker2d, a possible explanation is that under the Gaussian policy parameterization we use, the MSE loss is the same as the logarithmic loss up to state-dependent heteroskedasticity.[16] Another possible explanation is that this is an instance of the phenomenon described in **??**.

**Relationship between regret and Hellinger distance to expert.** Finally, we directly evaluate the quality of (i) Hellinger distance $D_{\mathsf{H}}^2\big(\mathbb{P}^{\pi^\star}, \mathbb{P}^{\widehat{\pi}}\big)$, and (ii) validation loss as proxies for rollout performance. We estimate the Hellinger distance using sample trajectories. Figure 6 displays our findings for Walker2d with $H = n = 500$, where we observe that both metrics, particularly the Hellinger distance, are well correlated with rollout performance, as measured by average reward. In Figure 6a, we see that under LogLossBC, Hellinger distance and validation loss are highly correlated with each other, and negatively correlated with expected reward, thereby acting as excellent proxies for rollout performance. Meanwhile, in Figure 6b, we find that under behavior cloning with the MSE loss, validation error is less well correlated with the expected reward of the imitator policy, as evinced by the cluster in the upper left corner, where there are policies with roughly the same validation loss, but variable expected reward. On the other hand, the Hellinger distance $D_{\mathsf{H}}^2$ still appears to predict the performance of the policy well, as is consistent with our theoretical results.

---

[14]We only include log-product-norm plots for Dyck and Car because for the other environments (Walker2d and BeamriderNoFrameskip), we do not change the expert as a function of $H$.

[15]In particular, we attempted a similar result in the Atari environment, using MSE loss being between vectors on the probability simplex over $|\mathcal{A}|$ actions. For MSE loss, we found that the imitator did not train, in the sense that even with 500 expert trajectories, the performance of the cloner did not improve. We suspect this was due to numerical instability in optimization for the MSE loss in this setup or a failure of hyperparameter optimization.

[16]In theory, the MSE loss can still underperform the logarithmic loss when the heteroskedasticity is severe [32], but this may not manifest for this environment.

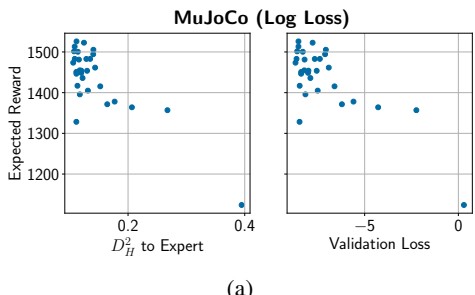
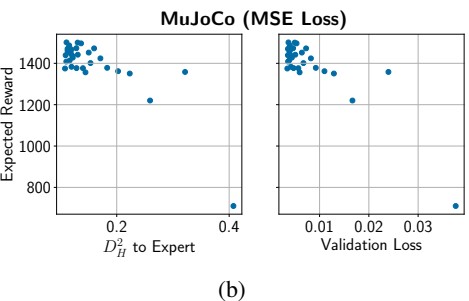

|  |  |
|:---:|:---:|
| (a) | (b) |

Figure 6: Evaluation of the quality of (i) Hellinger distance $D_{\mathsf{H}}^2\big(\mathbb{P}^{\pi^\star}, \mathbb{P}^{\widehat{\pi}}\big)$, and (ii) validation loss as a proxy for rollout reward. We plot Hellinger distance and validation loss against mean reward for a over a single training run for Walker2d environment with $H = 500$ and $n = 500$. **(a)** Results for LogLossBC, where the validation loss and Hellinger distance $D_{\mathsf{H}}^2$ are highly correlated, and serve as good proxies for the expected reward of the policy. **(b)** Results for MSE loss, where the validation loss is less well correlated with the expected reward (note the cluster in the upper left hand corner), but the Hellinger distance $D_{\mathsf{H}}^2$ remains a good proxy.

## D   Technical Tools

### D.1   Tail Bounds

**Lemma D.1** (e.g., Foster et al. [34])**.** *For any sequence of real-valued random variables $(X_t)_{t\leq T}$ adapted to a filtration $(\mathscr{F}_t)_{t\leq T}$, it holds that with probability at least $1 - \delta$, for all $T' \leq T$,*

$$\sum_{t=1}^{T'} -\log\big(\mathbb{E}_{t-1}\big[e^{-X_t}\big]\big) \leq \sum_{t=1}^{T'} X_t + \log(\delta^{-1}).$$

**Lemma D.2** (Time-uniform Freedman-type inequality)**.** *Let $(X_t)_{t\leq T}$ be a real-valued martingale difference sequence adapted to a filtration $(\mathscr{F}_t)_{t\leq T}$. If $|X_t| \leq R$ almost surely, then for any $\eta \in (0, 1/R)$, with probability at least $1 - \delta$, for all $T' \leq T$.*

$$\sum_{t=1}^{T'} X_t \leq \eta \sum_{t=1}^{T'} \mathbb{E}_{t-1}\big[X_t^2\big] + \frac{\log(\delta^{-1})}{\eta}.$$

**Proof of Lemma D.2.** Let $S_t = \sum_{s=1}^{t} X_s$ and $V_t = \sum_{s=1}^{t} \mathbb{E}_{t-1}\big[X_t^2\big]$. Let $Z_t = \exp(\eta S_t - \eta^2 V_t)$. As shown in Beygelzimer et al. [11] (see proof of Theorem 1), as long as $\eta \leq 1/R$,

$$\mathbb{E}_{t-1}[\exp(\eta X_t)] \leq \exp(\eta^2 \mathbb{E}_{t-1}\big[X_t^2\big]),$$

and so

$$\mathbb{E}_{t-1}[Z_t] = \mathbb{E}_{t-1}\big[\exp\big(\eta X_t - \eta^2 \mathbb{E}_{t-1}\big[X_t^2\big]\big)\big] \cdot Z_{t-1} \leq Z_{t-1}.$$

It follows that $(Z_t)$ is a non-negative supermartingale. Hence, by Ville's inequality, for any $\eta \in (0, 1/R)$, we have that for any $\tau > 0$,

$$\mathbb{P}[\exists t : S_t - \eta V_t \geq \tau] = \mathbb{P}[\exists t : Z_t \geq e^{\eta\tau}] \leq e^{-\eta\tau} \mathbb{E}[Z_T] \leq e^{-\eta\tau}.$$

We conclude by setting $\tau = \log(\delta^{-1})/\eta$.  $\square$

The following result is a standard consequence of Lemma D.2.

**Lemma D.3.** *Let $(X_t)_{t\leq T}$ be a sequence of random variables adapted to a filtration $(\mathscr{F}_t)_{t\leq T}$. If $0 \leq X_t \leq R$ almost surely, then with probability at least $1 - \delta$, for all $T' \leq T$,*

$$\sum_{t=1}^{T'} X_t \leq \frac{3}{2} \sum_{t=1}^{T'} \mathbb{E}_{t-1}[X_t] + 4R\log(2\delta^{-1}),$$

*and*

$$\sum_{t=1}^{T'} \mathbb{E}_{t-1}[X_t] \leq 2\sum_{t=1}^{T'} X_t + 8R\log(2\delta^{-1}).$$

## D.2 Information Theory

For a pair of probability measures $\mathbb{P}$ and $\mathbb{Q}$, we define the total variation distance as $D_{\mathsf{TV}}(\mathbb{P}, \mathbb{Q}) = \frac{1}{2}\int |\mathrm{d}\mathbb{P} - \mathrm{d}\mathbb{Q}|$, and define the $\chi^2$-divergence by $D_{\chi^2}(\mathbb{P} \| \mathbb{Q}) = \int \frac{(\mathrm{d}\mathbb{Q} - \mathrm{d}\mathbb{Q})^2}{\mathrm{d}\mathbb{Q}}$ if $\mathbb{P} \ll \mathbb{Q}$ and $D_{\chi^2}(\mathbb{P} \| \mathbb{Q}) = +\infty$ otherwise. We define KL divergence by $D_{\mathsf{KL}}(\mathbb{P} \| \mathbb{Q}) = \int \mathrm{d}\mathbb{P} \log\left(\frac{\mathrm{d}\mathbb{P}}{\mathrm{d}\mathbb{Q}}\right)$ if $\mathbb{P} \ll \mathbb{Q}$ and $D_{\mathsf{KL}}(\mathbb{P} \| \mathbb{Q}) = +\infty$ otherwise.

The following lemma states some basic inequalities between divergences.

**Lemma D.4** (e.g., [62]). *The following inequalities hold:*

- $D^2_{\mathsf{TV}}(\mathbb{P}, \mathbb{Q}) \le D^2_{\mathsf{H}}(\mathbb{P}, \mathbb{Q}) \le 2 D_{\mathsf{TV}}(\mathbb{P}, \mathbb{Q}).$

- $\frac{1}{6} D^2_{\mathsf{H}}(\mathbb{P}, \mathbb{Q}) \le D_{\chi^2}\left(\mathbb{P} \| \frac{1}{2}(\mathbb{P} + \mathbb{Q})\right) \le D^2_{\mathsf{H}}(\mathbb{P}, \mathbb{Q}).$

## D.3 Reinforcement Learning

The following lemma is a somewhat standard result; see, e.g., Lemma 15 in Zanette and Brunskill [101]. We include a proof for completeness.

**Lemma D.5** (Law of total variance). *For any (potentially stochastic) policy $\pi$, we have*

$$\mathrm{Var}^\pi\left[\sum_{h=1}^{H} r_h\right] = \mathbb{E}^\pi\left[\sum_{h=0}^{H} \mathrm{Var}^\pi\left[r_h + V_{h+1}^\pi(x_{h+1}) \mid x_h\right]\right],$$

*with the convention that $x_0$ is a deterministic dummy state (so that $P_0(x_1 = \cdot \mid x_0, a = \cdot)$ is the initial state distribution) and $r_0 = 0$.*

**Proof of Lemma D.5.** Let $h \in \{0, \ldots, H\}$ be fixed. We can expand

$$\mathrm{Var}^\pi\left[\sum_{\ell=h}^{H} r_\ell \mid x_h\right] = \mathbb{E}^\pi\left[\left(\sum_{\ell=h}^{H} r_\ell - V_h^\pi(x_h)\right)^2 \mid x_h\right]$$

$$= \mathbb{E}^\pi\left[\left(\sum_{\ell=h+1}^{H} r_\ell - V_{h+1}^\pi(x_{h+1}) + (r_h + V_{h+1}^\pi(x_{h+1}) - V_h^\pi(x_h))\right)^2 \mid x_h\right]$$

$$= \mathbb{E}^\pi\left[\left(\sum_{\ell=h+1}^{H} r_\ell - V_{h+1}^\pi(x_{h+1})\right)^2 \mid x_h\right] + \mathbb{E}^\pi\left[(r_h + V_{h+1}^\pi(x_{h+1}) - V_h^\pi(x_h))^2 \mid x_h\right]$$

$$+ 2\,\mathbb{E}^\pi\left[\left(\sum_{\ell=h+1}^{H} r_\ell - V_{h+1}^\pi(x_{h+1})\right)\left(r_h + V_{h+1}^\pi(x_{h+1}) - V_h^\pi(x_h)\right) \mid x_h\right]$$

$$= \mathbb{E}^\pi\left[\left(\sum_{\ell=h+1}^{H} r_\ell - V_{h+1}^\pi(x_{h+1})\right)^2 \mid x_h\right] + \mathbb{E}^\pi\left[(r_h + V_{h+1}^\pi(x_{h+1}) - V_h^\pi(x_h))^2 \mid x_h\right]$$

$$= \mathbb{E}^\pi\left[\mathrm{Var}^\pi\left[\sum_{\ell=h+1}^{H} r_\ell \mid x_{h+1}\right] \mid x_h\right] + \mathrm{Var}^\pi\left[(r_h + V_{h+1}^\pi(x_{h+1}) \mid x_h\right].$$

We conclude inductively that for all $h \in \{0, \ldots, H\}$,

$$\mathrm{Var}^\pi\left[\sum_{\ell=h}^{H} r_\ell \mid x_h\right] = \sum_{\ell=h}^{H} \mathbb{E}^\pi\left[\mathrm{Var}^\pi\left[(r_\ell + V_{\ell+1}^\pi(x_{\ell+1}) \mid x_\ell\right] \mid x_h\right].$$

To obtain the final expression, we note that

$$\mathrm{Var}^\pi\left[\sum_{h=1}^{H} r_h\right] = \mathrm{Var}^\pi\left[\sum_{h=0}^{H} r_h \mid x_0\right],$$

under the convention that $x_0$ is a deterministic dummy state (so that $P_1(x_1 = \cdot \mid x_0, a)$ is the initial state distribution) and $r_0 = 0$.

$\square$

### D.4 Maximum Likelihood Estimation

This section presents a self-contained analysis of the maximum likelihood estimator (MLE) for density estimation. The results are somewhat standard (e.g., Wong and Shen [97], van de Geer [89], Zhang [102]), but we include proofs for completeness.

Consider a setting where we receive $\{z^i\}_{i=1}^n$ i.i.d. from $z \sim g^\star$, where $g^\star \in \Delta(\mathcal{Z})$. We have a class $\mathcal{G} \subseteq \Delta(\mathcal{Z})$ that may or may not contain $g^\star$. We analyze the following maximum likelihood estimator:

$$\widehat{g} = \arg\max_{g \in \mathcal{G}} \sum_{i=1}^n \log(g(z^i)). \tag{10}$$

To provide sample complexity guarantees that support infinite classes, we appeal to the following notion of covering number (e.g., Wong and Shen [97]), which tailored to the log-loss.

**Definition D.1** (Covering number). *For a class $\mathcal{G} \subset \Delta(\mathcal{Z})$, we set that a class $\mathcal{G}' \subset \Delta(\mathcal{Z})$ is an $\varepsilon$-cover if for all $g \in \mathcal{G}$, there exists $g' \in \mathcal{G}'$ such that for all $z \in \mathcal{Z}$, $\log(g(z)/g'(z)) \leq \varepsilon$. We denote the size of the smallest such cover by $\mathcal{N}_{\log}(\mathcal{G}, \varepsilon)$.*

We also allow for optimization errors, and concretely assume that $\widehat{g}$ satisfies

$$\sum_{i=1}^n \log(\widehat{g}(z^i)) \geq \max_{g \in \mathcal{G}} \sum_{i=1}^n \log(g(z^i)) - \varepsilon_{\mathsf{opt}} \cdot n$$

for a parameter $\varepsilon_{\mathsf{opt}} \geq 0$; the case $\varepsilon_{\mathsf{opt}} = 0$ coincides with Eq. (10). Our main guarantee for MLE is as follows.

**Proposition D.1.** *The maximum likelihood estimator in Eq. (10) has that with probability at least $1 - \delta$,*

$$D_{\mathsf{H}}^2(\widehat{g}, g^\star) \leq \inf_{\varepsilon > 0} \left\{ \frac{6 \log(2\mathcal{N}_{\log}(\mathcal{G}, \varepsilon)/\delta^{-1})}{n} + 4\varepsilon \right\} + 2 \inf_{g \in \mathcal{G}} \log(1 + D_{\chi^2}(g^\star \parallel g)) + 2\varepsilon_{\mathsf{opt}}.$$

*In particular, if $\mathcal{G}$ is finite, the maximum likelihood estimator satisfies*

$$D_{\mathsf{H}}^2(\widehat{g}, g^\star) \leq \frac{6 \log(2|\mathcal{G}|/\delta^{-1})}{n} + 2 \inf_{g \in \mathcal{G}} \log(1 + D_{\chi^2}(g^\star \parallel g)) + 2\varepsilon_{\mathsf{opt}}.$$

Note that the term $\inf_{g \in \mathcal{G}} \log(1 + D_{\chi^2}(g^\star \parallel g))$ corresponds to misspecification error, and is zero if $g^\star \in \mathcal{G}$.

**Proof of Proposition D.1.** Let $\mathcal{G}_\varepsilon$ denote a minimal $\varepsilon$-cover for $\mathcal{G}$, and let $\widetilde{g} \in \mathcal{G}_\varepsilon$ denote any element that covers $\widehat{g}$ in the sense of Definition D.1. Going forward, we will use that $\widetilde{g}$ satisfies

$$D_{\mathsf{H}}^2(g^\star, \widetilde{g}) \leq D_{\mathsf{KL}}(g^\star \parallel \widetilde{g}) \leq \varepsilon. \tag{11}$$

Let $\ell^i(g) = -\log(g(z^i))$, and set $\widehat{L}(g) = -\sum_{i=1}^n \log(g(z^i))$. Set $X_i(g) = \frac{1}{2}(\ell^i(g) - \ell^i(g^\star))$. By applying Lemma D.1 with the sequence $(X_i(g))_{i=1}^n$ for each $g \in \mathcal{G}_\varepsilon$ and taking a union bound, we have that with probability at least $1 - \delta$, for all $g \in \mathcal{G}_\varepsilon$

$$-n \cdot \log\left(\mathbb{E}_{z \sim g^\star}\left[e^{\frac{1}{2}\log(g(z)/g^\star(z))}\right]\right) \leq \frac{1}{2}\left(\widehat{L}(g) - \widehat{L}(g^\star)\right) + \log(|\mathcal{G}_\varepsilon|\delta^{-1}).$$

Using a standard argument [102], we have that

$$-\log\left(\mathbb{E}_{z \sim g^\star}\left[e^{\frac{1}{2}\log(g(z)/g^\star(z))}\right]\right) = -\log\left(1 - \frac{1}{2}D_{\mathsf{H}}^2(g, g^\star)\right) \geq \frac{1}{2}D_{\mathsf{H}}^2(g, g^\star).$$

In particular, this implies that

$$D_{\mathsf{H}}^2(\widetilde{g}, g^\star) \leq \frac{2 \log(|\mathcal{G}|/\delta^{-1})}{n} + \frac{1}{n}\left(\widehat{L}(\widetilde{g}) - \widehat{L}(g^\star)\right),$$

and so

$$D_{\mathsf{H}}^2(\widehat{g}, g^\star) \leq 2D_{\mathsf{H}}^2(\widehat{g}, \widetilde{g}) + 2D_{\mathsf{H}}^2(\widetilde{g}, g^\star) \leq \frac{4 \log(|\mathcal{G}|/\delta^{-1})}{n} + \frac{2}{n}\left(\widehat{L}(\widetilde{g}) - \widehat{L}(g^\star)\right) + 2\varepsilon,$$

by the triangle inequality for Hellinger distance and Eq. (11).

It remains to bound the right-hand-side. Let $\bar{g} \in \mathcal{G}$ be arbitrary. We can bound

$$\widehat{L}(\widetilde{g}) - \widehat{L}(g^\star) \leq \widehat{L}(\widetilde{g}) - \widehat{L}(\widehat{g}) + \widehat{L}(\widehat{g}) - \widehat{L}(g^\star) \leq \widehat{L}(\widetilde{g}) - \widehat{L}(\widehat{g}) + \widehat{L}(\bar{g}) - \widehat{L}(g^\star) + \varepsilon_{\mathsf{opt}} n, \quad (12)$$

by the definition of the maximum likelihood estimator. For the first term in Eq. (12), we observe that

$$\widehat{L}(\widetilde{g}) - \widehat{L}(g^\star) = \sum_{i=1}^{n} \log(g^\star(z^i)/\widetilde{g}(z^i)) \leq \varepsilon n,$$

by Definition D.1.

To bound the second term in Eq. (12), set $Y_i = -(\ell^t(\bar{g}) - \ell^t(g^\star))$. Applying Lemma D.1 with the sequence $(Y_i)_{i=1}^{n}$, we have that with probability at least $1 - \delta$,

$$\widehat{L}(\bar{g}) - \widehat{L}(g^\star) \leq n \cdot \log\left(\mathbb{E}_{z \sim g^\star}\left[e^{\log(g^\star(z)/\bar{g}(z))}\right]\right) + \log(\delta^{-1}).$$

Finally, note that

$$\log\left(\mathbb{E}_{z \sim g^\star}\left[e^{\log(g^\star(z)/\bar{g}(z))}\right]\right) = \log\left(\mathbb{E}_{z \sim g^\star}\left[\frac{g^\star(z)}{\bar{g}(z)}\right]\right) = \log(1 + D_{\chi^2}(g^\star \parallel \bar{g})).$$

The result follows by choosing $\bar{g} \in \mathcal{G}$ to minimize this quantity. $\qquad \square$

# Part I

# Proofs and Supporting Results

## E   Examples and Supporting Results from Section 2 and Section 3

This section contains supporting results from Sections 2 and 3:

- Appendix E.1 presents general sample complexity guarantees for log-loss behavior cloning that support infinite policy classes and misspecification, as well as concrete examples.

- Appendix E.2 formally introduces the online imitation learning framework, and gives sample complexity guarantees for a log-loss variant of Dagger.

### E.1   General Guarantees and Examples for Log-Loss Behavior Cloning

In this section, we give bounds on the generalization error $D_{\mathsf{H}}^2\big(\mathbb{P}^{\widehat{\pi}}, \mathbb{P}^{\pi^\star}\big)$ for log-loss behavior cloning for concrete classes $\Pi$ of interest. To do so, we observe that the log-loss behavior cloning objective

$$\widehat{\pi} = \arg\max_{\pi \in \Pi} \sum_{i=1}^{n} \sum_{h=1}^{H} \log(\pi_h(a_h^i \mid x_h^i)).$$

is equivalent to performing maximum likelihood estimation over the *density class* $\mathcal{P} = \{\mathbb{P}^\pi\}_{\pi \in \Pi}$. Indeed, for any $\pi \in \Pi$, we have

$$\sum_{i=1}^{n} \log(\mathbb{P}^\pi(o^i)) = \sum_{i=1}^{n} \log\left( P_0(x_1^i) \prod_{h=1}^{H} P_h(x_{h+1}^i \mid x_h^i, a_h^i) \pi_h(a_h^i \mid x_h^i) \right)$$

$$= \sum_{i=1}^{n} \sum_{h=1}^{H} \log(\pi_h(a_h^i \mid x_h^i)) + C(\mathcal{D}),$$

where $C(\mathcal{D})$ is a constant that depends on the dataset $\mathcal{D}$ but not on $\pi$. It follows that both objectives have the same maximizer. Consequently, we can prove sample complexity bounds for log-loss behavior cloning by specializing sample complexity bounds for MLE given in Appendix D.4.

To give guarantees that support infinite policy classes, we appeal to the following notion of covering number.

**Definition E.1** (Policy covering number). *For a class $\Pi \subset \{\pi_h : \mathcal{X} \to \Delta(\mathcal{A})\}$, we set that $\Pi' \subset \{\pi_h : \mathcal{X} \to \Delta(\mathcal{A})\}$ is an $\varepsilon$-cover if for all $\pi \in \Pi$, there exists $\pi' \in \Pi'$ such that for all $x \in \mathcal{X}$, $a \in \mathcal{A}$, and $h \in [H]$, $\log(\pi_h(a \mid x)/\pi'_h(a \mid x)) \le \varepsilon$. We denote the size of the smallest such cover by $\mathcal{N}_{\mathrm{pol}}(\Pi, \varepsilon)$.*

In addition, to allow for optimization errors, we replace Eq. (4) with the assumption that $\widehat{\pi}$ satisfies

$$\sum_{i=1}^{n} \sum_{h=1}^{H} \log(\widehat{\pi}_h(a_h^i \mid x_h^i)) \ge \max_{\pi \in \Pi} \sum_{i=1}^{n} \sum_{h=1}^{H} \log(\pi_h(a_h^i \mid x_h^i)) - \varepsilon_{\mathsf{opt}} \cdot n \tag{13}$$

for a parameter $\varepsilon_{\mathsf{opt}} > 0$; Eq. (4) is the special case in which $\varepsilon_{\mathsf{opt}} = 0$. With these definitions, specializing Proposition D.1 leads to the following result.

**Theorem E.1** (Generalization bound for `LogLossBC`). *The* `LogLossBC` *policy in Eq. (13) has that with probability at least $1 - \delta$,*

$$D_{\mathsf{H}}^2\left(\mathbb{P}^{\widehat{\pi}}, \mathbb{P}^{\pi^\star}\right) \le \inf_{\varepsilon > 0} \left\{ \frac{6 \log(2\mathcal{N}_{\mathrm{pol}}(\Pi, \varepsilon/H)\delta^{-1})}{n} + 4\varepsilon \right\} + 2 \inf_{\pi \in \Pi} \log\left(1 + D_{\chi^2}\left(\mathbb{P}^{\pi^\star} \| \mathbb{P}^\pi\right)\right) + 2\varepsilon_{\mathsf{opt}}.$$

*In particular, if $\Pi$ is finite, the log-loss behavior cloning policy satisfies*

$$D_{\mathsf{H}}^2\left(\mathbb{P}^{\widehat{\pi}}, \mathbb{P}^{\pi^\star}\right) \le \frac{6 \log(2|\Pi|\delta^{-1})}{n} + 2 \inf_{\pi \in \Pi} \log\left(1 + D_{\chi^2}\left(\mathbb{P}^{\pi^\star} \| \mathbb{P}^\pi\right)\right) + 2\varepsilon_{\mathsf{opt}}.$$

Let us make two remarks.

- First, the only explicit dependence on the horizon $H$ is through the precision $\varepsilon/H$ through which we evaluate the covering number: $\mathcal{N}_{\mathrm{pol}}(\Pi, \varepsilon/H)$. As a result, for *parametric* classes where $\mathcal{N}_{\mathrm{pol}}(\Pi, \varepsilon) \asymp \log(\varepsilon^{-1})$ (we will give examples in the sequel), the result will scale at most logarithmically in $H$, but for nonparametric classes the dependence can be polynomial.

- Second, the remainder term $\inf_{\pi \in \Pi} \log(1 + D_{\chi^2}(\mathbb{P}^{\pi^\star} \| \mathbb{P}^\pi))$ corresponds to misspecification error, and is zero if $\pi^\star \in \Pi$. We remark that when $\pi^\star$ is deterministic, this expression can be simplified to $\inf_{\pi \in \Pi} \log\left(\mathbb{E}^{\pi^\star}\left[\frac{1}{\prod_{h=1}^{H} \pi_h(a_h|x_h)}\right]\right)$.

**Proof of Theorem E.1.** This follows by applying Proposition D.1 with the class $\{\mathbb{P}^\pi\}_{\pi \in \Pi}$, and noting that if $\pi'$ covers $\pi$ in the sense of Definition E.1, then for all $o \in (\mathcal{X} \times \mathcal{A})^H$, we have $\log(\mathbb{P}^\pi(o)/\mathbb{P}^{\pi'}(o)) \le \varepsilon H$, meaning that an $\varepsilon$-cover in the sense of Definition E.1 yields an $\varepsilon H$-cover in the sense of Definition D.1.

$\square$

### E.1.1 Example: Tabular Policies

We now instantiate Theorem E.1 to give generalization bounds for specific policy classes of interest.

Consider a tabular MDP in which $|\mathcal{X}|, |\mathcal{A}| < \infty$ are small and finite. Here, choosing $\Pi$ to be the set of all stationary policies leads to a bound independent of $H$.

**Corollary E.1** (Stationary tabular policies). *When $\Pi$ is the set of all deterministic stationary policies, the log-loss behavior cloning policy Eq. (4) has that with probability at least $1 - \delta$,*

$$D_{\mathsf{H}}^2\left(\mathbb{P}^{\widehat{\pi}}, \mathbb{P}^{\pi^\star}\right) \le O\left(\frac{|\mathcal{X}| \log(|\mathcal{A}|\delta^{-1})}{n}\right).$$

*Meanwhile, if $\Pi$ is the set of all* stochastic *stationary policies, the log-loss behavior cloning policy Eq. (4) has that with probability at least $1 - \delta$,*

$$D_{\mathsf{H}}^2\left(\mathbb{P}^{\widehat{\pi}}, \mathbb{P}^{\pi^\star}\right) \le \widetilde{O}\left(\frac{|\mathcal{X}||\mathcal{A}| \log(Hn\delta^{-1})}{n}\right).$$

**Proof of Corollary E.1.** This follows by noting that we have $\log|\Pi| \le |\mathcal{X}| \log|\mathcal{A}|$ in the deterministic case and $\log \mathcal{N}_{\mathrm{pol}}(\Pi, \varepsilon) \le \widetilde{O}(|\mathcal{X}||\mathcal{A}| \log(\varepsilon^{-1}))$ in the stochastic case (this follows

from a standard discretization argument, e.g., Wainwright [91]). □

Naturally, we can also give generalization guarantees for non-stationary tabular policies, though the sample complexity will scale with $H$ in this case.

**Corollary E.2** (Non-stationary tabular policies). *When $\Pi$ is the set of all deterministic non-stationary policies, the log-loss behavior cloning policy Eq. (4) has that with probability at least $1 - \delta$,*

$$D_{\mathsf{H}}^2\big(\mathbb{P}^{\widehat{\pi}}, \mathbb{P}^{\pi^\star}\big) \leq O\bigg(\frac{H|\mathcal{X}|\log(|\mathcal{A}|\delta^{-1})}{n}\bigg).$$

*Meanwhile, if $\Pi$ is the set of all* stochastic *non-stationary policies, the log-loss behavior cloning policy Eq. (4) has that with probability at least $1 - \delta$,*

$$D_{\mathsf{H}}^2\big(\mathbb{P}^{\widehat{\pi}}, \mathbb{P}^{\pi^\star}\big) \leq \widetilde{O}\bigg(\frac{H|\mathcal{X}||\mathcal{A}|\log(Hn\delta^{-1})}{n}\bigg).$$

**Proof of Corollary E.2.** This follows because we have $\log|\Pi| \leq H|\mathcal{X}|\log|\mathcal{A}|$ in the deterministic case and $\log\mathcal{N}_{\mathrm{pol}}(\Pi, \varepsilon) \leq \widetilde{O}\big(H|\mathcal{X}||\mathcal{A}|\log(\varepsilon^{-1})\big)$ in the stochastic case. □

### E.1.2 Example: Softmax Policies

Next, we give an example of a general family of policy classes based on function approximation for which the sample complexity is at most polylogarithmic in $H$.

For a vector $v \in \mathbb{R}^{\mathcal{A}}$, let $\sigma : \mathbb{R}^{\mathcal{A}} \to \Delta(\mathcal{A})$ be the softmax function, which is given by

$$\sigma_a(v) = \frac{\exp(v_a)}{\sum_{a' \in \mathcal{A}} \exp(v_{a'})}.$$

Let $\mathcal{F} \subset \{f_h : \mathcal{X} \times \mathcal{A} \to \mathbb{R}\}_{h=1}^H$ be a class of value functions, and define the induced class of *softmax policies* via

$$\Pi_{\mathcal{F}} = \{\pi_f \mid f \in \mathcal{F}\},$$

where

$$\pi_{f,h}(x) := \sigma_a(f_h(x, a)).$$

We give sample complexity guarantees based on covering numbers for the value function class $\mathcal{F}$.

**Definition E.2** (Value function covering number). *For a class $\mathcal{F} \subset \{f_h : \mathcal{X} \times \mathcal{A} \to \mathbb{R}\}$, we set that $\mathcal{F}' \subset \{f_h : \mathcal{X} \times \mathcal{A} \to \mathbb{R}\}$ is an $\varepsilon$-cover if for all $f \in \mathcal{F}$, there exists $f' \in \mathcal{F}'$ such that for all $x \in \mathcal{X}$, $a \in \mathcal{A}$, and $h \in [H]$, $|f_h(x,a) - f_h'(x,a)| \leq \varepsilon$. We denote the size of the smallest such cover by $\mathcal{N}_{\mathrm{val}}(\Pi, \varepsilon)$.*

**Corollary E.3** (Softmax policies). *When $\Pi = \Pi_{\mathcal{F}}$ is the softmax policy class for a value function class $\mathcal{F}$, the log-loss behavior cloning policy Eq. (4) has that with probability at least $1 - \delta$,*

$$D_{\mathsf{H}}^2\Big(\mathbb{P}^{\widehat{\pi}}, \mathbb{P}^{\pi^\star}\Big) \leq O(1) \cdot \inf_{\varepsilon > 0}\bigg\{\frac{\log(\mathcal{N}_{\mathrm{val}}(\mathcal{F}, \varepsilon/H)\delta^{-1})}{n} + \varepsilon\bigg\} + 2\inf_{\pi \in \Pi_{\mathcal{F}}} \log\Big(1 + D_{\chi^2}\big(\mathbb{P}^{\pi^\star} \| \mathbb{P}^\pi\big)\Big).$$

**Proof of Corollary E.3.** Consider a pair of functions $f, f'$ with $|f_h(x,a) - f_h'(x,a)| \leq \varepsilon$ for all $x \in \mathcal{X}$, $a \in \mathcal{A}$, and $h \in [H]$. The induced softmax policies satisfy

$$\log(\pi_{f,h}(a \mid x)/\pi_{f',h}(a \mid x)) = f_h(x,a) - f_h'(x,a) + \log\bigg(\frac{\sum_{a' \in \mathcal{A}} \exp(f_h'(x,a'))}{\sum_{a \in \mathcal{A}} \exp(f_h'(x,a'))}\bigg).$$

Clearly we have $f_h(x,a) - f_h'(x,a) \leq \varepsilon$, and we can bound

$$
\begin{aligned}
\log\bigg(\frac{\sum_{a' \in \mathcal{A}} \exp(f_h'(x,a'))}{\sum_{a \in \mathcal{A}} \exp(f_h(x,a'))}\bigg) &= \log\bigg(\frac{\sum_{a' \in \mathcal{A}} \exp(f_h(x,a')) \cdot \exp(f_h'(x,a') - f_h(x,a'))}{\sum_{a \in \mathcal{A}} \exp(f_h(x,a'))}\bigg) \\
&\leq \log\bigg(\frac{\sum_{a' \in \mathcal{A}} \exp(f_h(x,a')) \cdot \max_{a'' \in \mathcal{A}} \exp(f_h'(x,a'') - f_h(x,a''))}{\sum_{a \in \mathcal{A}} \exp(f_h(x,a'))}\bigg) \\
&\leq \max_{a'' \in \mathcal{A}}\{f_h'(x,a'') - f_h(x,a'')\} \leq \varepsilon.
\end{aligned}
$$

Hence, an $\varepsilon$-cover in the sense of Definition E.2 implies a $2\varepsilon$-cover in the sense of Definition E.1. $\quad\square$

Whenever $\mathcal{F}$ is parametric in the sense that $\log \mathcal{N}_{\mathrm{val}}(\mathcal{F}, \varepsilon) \propto \log(\varepsilon^{-1})$, Corollary E.3 leads to polylogarithmic dependence on $H$. The following result gives such an example.

**Linear softmax policies.** Consider the set of stationary linear softmax policies induced by the value function class

$$\mathcal{F} = \{(x, a, h) \mapsto \langle \phi_h(x, a), \theta \rangle \mid \|\theta\|_2 \le B\},$$

where $\phi_h(x, a) \in \mathbb{R}^d$ is a known feature map with $\|\phi_h(x, a)\| \le B$. Here, we have $\log \mathcal{N}_{\mathrm{val}}(\mathcal{F}, \varepsilon) \propto d \log(B\varepsilon^{-1})$ (e.g., Wainwright [91]), which yields the following generalization guarantee.

**Corollary E.4.** *When $\Pi$ is the set of stationary linear softmax policies and $\pi^\star \in \Pi$, the log-loss behavior cloning policy Eq. (4) has that with probability at least $1 - \delta$,*

$$D_{\mathsf{H}}^2(\mathbb{P}^{\widehat{\pi}}, \mathbb{P}^{\pi^\star}) \le O\left(\frac{d \log(BHn\delta^{-1})}{n}\right).$$

### E.2 Online IL Framework and Sample Complexity Bounds for Log-Loss Dagger

In this section, we give sample complexity bounds for a variant of the Dagger algorithm for online IL [72] that uses the logarithmic loss. The main purpose of including this result is to give end-to-end sample complexity guarantees for general policy classes, which we use in Sections 2 and 3 to compare the optimal rates for online and offline IL. For this comparison, we are be mainly interested in the case of deterministic expert policies, but our analysis supports stochastic policies, which may be of independent interest.

**Online imitation learning framework.** In the online imitation learning framework, learning proceeds in $n$ episodes in which the learner can directly interact with the underlying MDP $M^\star$ and query the expert advice. Concretely, for each episode $i \in [n]$, the learner executes a policy $\pi^i = \{\pi_h^i : \mathcal{X} \to \Delta(\mathcal{A})\}_{h=1}^H$ and receives a trajectory $o^t = (x_1^i, a_1^i, a_1^{\star,i}), \ldots, (x_H^i, a_H^i, a_H^{\star,i})$, in which $a_h^i \sim \pi_h^i(x_h^i)$, $a_h^{\star,i} \sim \pi^\star(x_h^t)$, and $x_{h+1}^i \sim P_h(x_h^i, a_h^i)$; in other words, the trajectory induced by the learner's policy is annotated by the expert's action $a_h^\star \sim \pi_h^\star(x_h)$ at each state $x_h$ encountered. After all $n$ episodes conclude, they can use all of the data collected to produce a policy $\widehat{\pi}$ such that $J(\pi^\star) - J(\widehat{\pi})$ is small.

**Dagger algorithm.** We consider a general version of the Dagger algorithm. The algorithm is parameterized by an online learning algorithm $\mathbf{Alg}_{\mathsf{Est}}$, which attempts to estimate the expert policy in a sequential fashion based on trajectories.

Set $\mathcal{D}^1 = \varnothing$. For $i = 1, \ldots, n$:

- Query online learning algorithm $\mathbf{Alg}_{\mathsf{Est}}$ with $\mathcal{D}^i$ and receive policy $\widehat{\pi}$.

- Execute $\widehat{\pi}$ and observe $o^i = (x_1^i, a_1^i, a_1^{\star,i}), \ldots, (x_H^i, a_H^i, a_H^{\star,i})$.

- Update $\mathcal{D}^{i+1} \leftarrow \mathcal{D}^i \cup \{o^i\}$.

At the end, we output $\widehat{\pi} = \mathrm{unif}(\pi^1, \ldots, \pi^n)$ as the final policy.

To measure the performance of the estimation oracle, we define the online estimation error as:

$$\mathbf{Est}_{\mathsf{H}}^{\mathsf{on}}(n) = \frac{1}{n} \sum_{i=1}^{n} \sum_{h=1}^{H} \mathbb{E}^{\widehat{\pi}^i} \left[ D_{\mathsf{H}}^2(\widehat{\pi}_h^i(x_h), \pi^\star(x_h)) \right].$$

As we will show in a moment, this notion of estimation error is well-suited for online learning algorithms that estimate $\pi^\star$ using the logarithmic loss.

Our following result gives a general guarantee for Dagger that holds for any choice of online learning algorithm. To state the result, let $\mathbb{P}^{\pi^\star|\pi}$ denote the law of $o = (x_1, a_1, a_1^\star), \ldots, (x_H, a_H, a_H^\star)$ when $\pi^\star$ is the expert policy and we execute $\pi$. Let

$$\sigma_{\pi^\star|\pi}^2 = \sum_{h=1}^{H} \mathbb{E}^{\pi \circ_h \pi^\star} \left[ (Q_h^{\pi^\star}(x_h, a_h) - V_h^{\pi^\star}(x_h))^2 \right],$$

so that $\sigma_{\pi^\star}^2 = \sigma_{\pi^\star | \pi^\star}^2$ and define $\bar{\sigma}_{\pi^\star}^2 = \sup_\pi \sigma_{\pi^\star | \pi}^2$. Note that $\bar{\sigma}_{\pi^\star}^2 = 0$ whenever $\pi^\star$ is deterministic, but in general, $\bar{\sigma}_{\pi^\star}^2 \geq \sigma_{\pi^\star}^2$.

**Proposition E.1** (Regret for `Dagger`). *For any MDP $M^\star$ with signed recoverability parameter $\widetilde{\mu}$ and any online learning algorithm $\mathbf{Alg}_{\mathsf{Est}}$, `Dagger` ensures that*

$$J(\pi^\star) - J(\widehat{\pi}) \lesssim \sqrt{\bar{\sigma}_{\pi^\star}^2 \cdot \mathbf{Est}_{\mathsf{H}}^{\mathsf{on}}(n)} + \widetilde{\mu} \cdot \mathbf{Est}_{\mathsf{H}}^{\mathsf{on}}(n).$$

*Furthermore, whenever $\pi^\star$ is deterministic, `Dagger` ensures that*

$$J(\pi^\star) - J(\widehat{\pi}) \lesssim \mu \cdot \mathbf{Est}_{\mathsf{H}}^{\mathsf{on}}(n). \tag{14}$$

To instantiate the bound above, we choose $\mathbf{Alg}_{\mathsf{Est}}$ by applying the exponential weights algorithm (e.g., Cesa-Bianchi and Lugosi [19]) with the logarithmic loss. Let $\Pi_h := \{\pi_h \mid \pi \in \Pi\}$ denote the projection of $\Pi$ onto step $h$. The algorithm proceeds as follows. At step $i \in [n]$, given the dataset $\mathcal{D}^i$, for each layer $h \in [H]$ we define a distribution $\mu_h^i \in \Delta(\Pi_h)$ via

$$\mu_h^i(\pi) \propto \exp\left(\sum_{j<j} \log(\pi_h(a_h^{\star,j} \mid x_h^j))\right) = \prod_{j<j} \pi_h(a_h^{\star,j} \mid x_h^j).$$

We then set

$$\widehat{\pi}_h^i(a \mid x) = \mathbb{E}_{\pi_h \sim \mu_h^i}[\pi_h(a \mid x)].$$

We refer to the resulting algorithm as `LogLossDagger`. This leads to the following guarantee for finite classes.

**Proposition E.2** (Regret for `LogLossDagger`). *When $\pi^\star \in \Pi$, the log-loss exponential weights algorithm ensures that with probability at least $1 - \delta$,*

$$\mathbf{Est}_{\mathsf{H}}^{\mathsf{on}}(n) \leq \frac{2}{n} \sum_{h=1}^H \log(|\Pi_h| H \delta^{-1}).$$

*Consequently, `LogLossDagger` ensures that with probability at least $1 - \delta$,*

$$J(\pi^\star) - J(\widehat{\pi}) \lesssim \sqrt{\bar{\sigma}_{\pi^\star}^2 \cdot \sum_{h=1}^H \frac{\log(|\Pi_h| H \delta^{-1})}{n}} + \widetilde{\mu} \cdot \sum_{h=1}^H \frac{\log(|\Pi_h| H \delta^{-1})}{n},$$

*and when $\pi^\star$ is deterministic,*

$$J(\pi^\star) - J(\widehat{\pi}) \lesssim \mu \cdot \sum_{h=1}^H \frac{\log(|\Pi_h| H \delta^{-1})}{n}.$$

We note that for many parameter regimes, the sample complexity bound in Proposition E.1 can be worse than that of `LogLossBC` in Theorem 3.1 (for stationary policies, Proposition E.1 has spurious dependence on $H$, and the variance-like quantity in the leading order term is weaker). It would be interesting to get the best of both worlds, though this may require changing the algorithm.

**Proof of Proposition E.1.** Consider an arbitrary policy $\widehat{\pi}$. Begin by writing

$$J(\pi^\star) - J(\widehat{\pi}) = \sum_{h=1}^H \mathbb{E}^{\widehat{\pi}|\widehat{\pi}}\left[Q_h^{\pi^\star}(x_h, \pi_h^\star(x_h)) - Q_h^{\pi^\star}(x_h, a_h)\right].$$

Fix a layer $h$. By Lemma 3.1, we have

$$\mathbb{E}^{\widehat{\pi}|\widehat{\pi}}\left[Q_h^{\pi^\star}(x_h, \pi_h^\star(x_h)) - Q_h^{\pi^\star}(x_h, a_h)\right]$$

$$\leq \mathbb{E}^{\pi^\star|\widehat{\pi}}\left[Q_h^{\pi^\star}(x_h, \pi_h^\star(x_h)) - Q_h^{\pi^\star}(x_h, a_h)\right]$$

$$+ \sqrt{\left(\mathbb{E}^{\widehat{\pi}|\widehat{\pi}}[(Q_h^{\pi^\star}(x_h, \pi_h^\star(x_h)) - Q_h^{\pi^\star}(x_h, a_h))^2] + \mathbb{E}^{\pi^\star|\widehat{\pi}}[(Q_h^{\pi^\star}(x_h, \pi_h^\star(x_h)) - Q_h^{\pi^\star}(x_h, a_h))^2]\right) \mathbb{E}^{\widehat{\pi}}[D_{\mathsf{H}}^2(\widehat{\pi}_h(x_h), \pi_h^\star(x_h))]}$$

$$= \sqrt{\left(\mathbb{E}^{\widehat{\pi}|\widehat{\pi}}[(Q_h^{\pi^\star}(x_h, \pi_h^\star(x_h)) - Q_h^{\pi^\star}(x_h, a_h))^2] + \mathbb{E}^{\pi^\star|\widehat{\pi}}[(Q_h^{\pi^\star}(x_h, \pi_h^\star(x_h)) - Q_h^{\pi^\star}(x_h, a_h))^2]\right) \mathbb{E}^{\widehat{\pi}}[D_{\mathsf{H}}^2(\widehat{\pi}_h(x_h), \pi_h^\star(x_h))]}.$$

Furthermore, using Lemma 3.1, we have

$$\mathbb{E}^{\widehat{\pi}|\widehat{\pi}}\Big[(Q_h^{\pi^\star}(x_h, \pi_h^\star(x_h)) - Q_h^{\pi^\star}(x_h, a_h))^2\Big]$$

$$\lesssim \sum_{h=1}^{H} \mathbb{E}^{\pi^\star|\widehat{\pi}}\Big[(Q_h^{\pi^\star}(x_h, \pi_h^\star(x_h)) - Q_h^{\pi^\star}(x_h, a_h))^2\Big] + \widetilde{\mu}^2 \sum_{h=1}^{H} \mathbb{E}^{\widehat{\pi}}\big[D_{\mathsf{H}}^2(\widehat{\pi}_h(x_h), \pi_h^\star(x_h))\big],$$

so that

$$\mathbb{E}^{\widehat{\pi}|\widehat{\pi}}\Big[Q_h^{\pi^\star}(x_h, \pi_h^\star(x_h)) - Q_h^{\pi^\star}(x_h, a_h)\Big] \tag{15}$$

$$\lesssim \sqrt{\mathbb{E}^{\pi^\star|\widehat{\pi}}[(Q_h^{\pi^\star}(x_h, \pi_h^\star(x_h)) - Q_h^{\pi^\star}(x_h, a_h))^2] \cdot \mathbb{E}^{\widehat{\pi}}[D_{\mathsf{H}}^2(\widehat{\pi}_h(x_h), \pi_h^\star(x_h))]} + \widetilde{\mu} \cdot \mathbb{E}^{\widehat{\pi}}\big[D_{\mathsf{H}}^2(\widehat{\pi}_h(x_h), \pi_h^\star(x_h))\big].$$

Recall that the `Dagger` policy satisfies

$$J(\pi^\star) - J(\widehat{\pi}) = \frac{1}{n}\sum_{i=1}^{n} J(\pi^\star) - J(\widehat{\pi}^i).$$

Applying Eq. (15) to each policy $\widehat{\pi}^i$, summing over all layer $h$, and applying Cauchy-Schwarz yields

$$J(\pi^\star) - J(\widehat{\pi}) \lesssim \sqrt{\frac{1}{n}\sum_{i=1}^{n} \sigma_{\pi^\star|\pi^i}^2 \cdot \mathbf{Est}_{\mathsf{H}}^{\mathsf{on}}(n)} + \widetilde{\mu} \cdot \mathbf{Est}_{\mathsf{H}}^{\mathsf{on}}(n)$$

$$\lesssim \sqrt{\overline{\sigma}_{\pi^\star}^2 \cdot \mathbf{Est}_{\mathsf{H}}^{\mathsf{on}}(n)} + \widetilde{\mu} \cdot \mathbf{Est}_{\mathsf{H}}^{\mathsf{on}}(n).$$

In the deterministic case, we tighten the argument above by applying the following improved change-of-measure argument based on Lemma 3.1:

$$\mathbb{E}^{\pi^\star|\widehat{\pi}}\Big[Q_h^{\pi^\star}(x_h, \pi_h^\star(x_h)) - Q_h^{\pi^\star}(x_h, a_h)\Big]$$

$$\leq \mathbb{E}^{\pi^\star|\widehat{\pi}}\Big[(Q_h^{\pi^\star}(x_h, \pi_h^\star(x_h)) - Q_h^{\pi^\star}(x_h, a_h))_+\Big]$$

$$\leq 2\,\mathbb{E}^{\pi^\star|\widehat{\pi}}\Big[(Q_h^{\pi^\star}(x_h, \pi_h^\star(x_h)) - Q_h^{\pi^\star}(x_h, a_h))_+\Big] + \mu \cdot \mathbb{E}^{\widehat{\pi}}\big[D_{\mathsf{H}}^2(\widehat{\pi}_h(x_h), \pi_h^\star(x_h))\big]$$

$$= \mu \cdot \mathbb{E}^{\widehat{\pi}}\big[D_{\mathsf{H}}^2(\widehat{\pi}_h(x_h), \pi_h^\star(x_h))\big],$$

This leads to Eq. (14).

$\square$

**Proof of Proposition E.2.** Since $\pi^\star \in \Pi$, a standard guarantee for exponential weights with the log-loss (e.g., Cesa-Bianchi and Lugosi [19]) ensures that for all $h \in [H]$, the following bound holds almost surely:

$$\sum_{i=1}^{n} \log(1/\widehat{\pi}_h^i(a_h^{\star,i} \mid x_h^i)) \leq \sum_{i=1}^{n} \log(1/\pi_h^\star(a_h^{\star,i} \mid x_h^i)) + \log|\Pi_h|.$$

From here, for each $h \in [H]$, Lemma A.14 of Foster et al. [34] implies that with probability at least $1 - \delta$,

$$\sum_{i=1}^{n} \mathbb{E}^{\widehat{\pi}^i}\big[D_{\mathsf{H}}^2(\widehat{\pi}_h^i(x_h), \pi^\star(x_h))\big] \leq \log|\Pi_h| + 2\log(\delta^{-1}).$$

The result now follows by taking a union bound.

$\square$

# F Proofs from Section 2

## F.1 Proof of Theorem 2.1

**Proof of Theorem 2.1.** We begin by defining the following *trajectory-wise* semi-metric between policies. For a pair of potentially stochastic policies $\pi$ and $\pi'$, define

$$\rho(\pi \parallel \pi') := \mathbb{E}^\pi \, \mathbb{E}_{a'_{1:H} \sim \pi'(x_{1:H})}[\mathbb{I}\{\exists h : a_h \neq a'_h\}],$$

where we use the shorthand $a'_{1:H} \sim \pi'(x_{1:H})$ to indicate that $a'_1 \sim \pi'_1(x_1), \ldots, a'_H \sim \pi'_H(x_H)$. Despite being defined in an asymmetric fashion, the following lemma shows that the trajectory-wise distance $\rho(\cdot \parallel \cdot)$ is symmetric, from which it follows that it is indeed a semi-metric.

**Lemma F.1.** *For all (potentially stochastic) policies $\pi$ and $\pi'$, it holds that*

$$\rho(\pi \parallel \pi') = \rho(\pi' \parallel \pi).$$

Next, we show that it is possible to bound the difference in reward for any pair of policies in terms of the trajectory-wise distance $\rho(\cdot \parallel \cdot)$.

**Lemma F.2.** *For all (potentially stochastic) policies $\pi$ and $\pi'$, it holds that*

$$J(\pi) - J(\pi') \leq R \cdot \rho(\pi \parallel \pi').$$

Finally, using Lemma F.1, we show that when one of the policies is deterministic, the trajectory-wise distance is equivalent to Hellinger distance up to an absolute constant.

**Lemma F.3.** *Let $\pi^\star$ be a deterministic policy and $\pi$ be an arbitrary stochastic policy. Then we have that*

$$\frac{1}{4} \cdot \rho(\pi^\star \parallel \pi) \leq D^2_{\mathsf{H}}\left(\mathbb{P}^\pi, \mathbb{P}^{\pi^\star}\right) \leq 2 \cdot \rho(\pi^\star \parallel \pi).$$

Combining Lemmas F.2 and F.3, we conclude that for any deterministic policy $\pi^\star$ and stochastic policy $\widehat{\pi}$,

$$J(\pi^\star) - J(\widehat{\pi}) \leq 4R \cdot D^2_{\mathsf{H}}\left(\mathbb{P}^{\widehat{\pi}}, \mathbb{P}^{\pi^\star}\right).$$

$\square$

**Proof of Lemma F.1.** This follows by noting that we can write

$$\rho(\pi \parallel \pi') = 1 - \mathbb{E}^\pi \, \mathbb{E}_{a'_{1:H} \sim \pi'(x_{1:H})}[\mathbb{I}\{a_h = a'_h \; \forall h\}]$$

$$= 1 - \sum_{x_{1:H}, a_{1:H}, a'_{1:H}} P_0(x_1) \prod_{h=1}^{H} P_h(x_{h+1} \mid x_h, a_h) \pi_h(a_h \mid x_h) \pi'_h(a'_h \mid x_h) \mathbb{I}\{a_h = a'_h\}$$

$$= 1 - \sum_{x_{1:H}, a_{1:H}, a'_{1:H}} P_0(x_1) \prod_{h=1}^{H} P_h(x_{h+1} \mid x_h, a'_h) \pi_h(a_h \mid x_h) \pi'_h(a'_h \mid x_h) \mathbb{I}\{a_h = a'_h\}$$

$$= 1 - \mathbb{E}^{\pi'} \, \mathbb{E}_{a'_{1:H} \sim \pi(x_{1:H})}[\mathbb{I}\{a_h = a'_h \; \forall h\}] = \rho(\pi' \parallel \pi).$$

$\square$

**Proof of Lemma F.2.** Observe that since $\sum_{h=1}^{H} r_h \in [0, R]$, we can bound the reward for $\pi$ as

$$J(\pi) \leq \mathbb{E}^\pi\left[\left(\sum_{h=1}^{H} r_h\right) \mathbb{E}_{a'_{1:H} \sim \pi'(x_{1:H})}[\mathbb{I}\{a'_h = a_h \; \forall h\}]\right] + R \cdot \mathbb{E}^\pi \, \mathbb{E}_{a'_{1:H} \sim \pi'(x_{1:H})}[\mathbb{I}\{\exists h : a'_h \neq a_h\}]$$

$$= \mathbb{E}^\pi\left[\left(\sum_{h=1}^{H} r_h\right) \mathbb{E}_{a'_{1:H} \sim \pi'(x_{1:H})}[\mathbb{I}\{a'_h = a_h \; \forall h\}]\right] + R \cdot \rho(\pi \parallel \pi').$$

We can bound the first term as

$$\mathbb{E}^{\pi}\left[\left(\sum_{h=1}^{H} r_h\right) \mathbb{E}_{a'_{1:H} \sim \pi'(x_{1:H})}[\mathbb{I}\{a'_h = a_h \;\forall h\}]\right]$$

$$= \mathbb{E}^{\pi}\left[f(x_{1:H}, a_{1:H}) \,\mathbb{E}_{a'_{1:H} \sim \pi'(x_{1:H})}[\mathbb{I}\{a'_h = a_h \;\forall h\}]\right],$$

where $f(x_{1:H}, a_{1:H}) := \sum_{h=1}^{H} \mathbb{E}[r_h \mid x_h, a_h]$. We now observe that for any function $f$,

$$\mathbb{E}^{\pi}\left[f(x_{1:H}, a_{1:H}) \,\mathbb{E}_{a'_{1:H} \sim \pi'(x_{1:H})}[\mathbb{I}\{a'_h = a_h \;\forall h\}]\right]$$

$$= \sum_{x_{1:H}, a_{1:H}, a'_{1:H}} f(x_{1:H}, a_{1:H}) \cdot P_0(x_1) \prod_{h=1}^{H} P_h(x_{h+1} \mid x_h, a_h)\pi_h(a_h \mid x_h)\pi'_h(a'_h \mid x_h)\mathbb{I}\{a_h = a'_h\}$$

$$= \sum_{x_{1:H}, a_{1:H}, a'_{1:H}} f(x_{1:H}, a'_{1:H}) \cdot P_0(x_1) \prod_{h=1}^{H} P_h(x_{h+1} \mid x_h, a'_h)\pi_h(a_h \mid x_h)\pi'_h(a'_h \mid x_h)\mathbb{I}\{a_h = a'_h\}$$

$$\leq \sum_{x_{1:H}, a'_{1:H}} f(x_{1:H}, a'_{1:H}) \cdot P_0(x_1) \prod_{h=1}^{H} P_h(x_{h+1} \mid x_h, a'_h)\pi'_h(a'_h \mid x_h)$$

$$= \mathbb{E}^{\pi'}[f(x_{1:H}, a_{1:H})].$$

We conclude that

$$\mathbb{E}^{\pi}\left[\left(\sum_{h=1}^{H} r_h\right) \mathbb{E}_{a'_{1:H} \sim \pi'(x_{1:H})}[\mathbb{I}\{a'_h = a_h \;\forall h\}]\right] \leq J(\pi'),$$

so that

$$J(\pi) - J(\pi') \leq R \cdot \rho(\pi \parallel \pi').$$

$\square$

**Proof of Lemma F.3.** Define the *triangular discrimination* via $D_{\Delta}(\mathbb{P}, \mathbb{Q}) := \int \frac{(d\mathbb{P} - d\mathbb{Q})^2}{d\mathbb{P} + d\mathbb{Q}}$, and recall that $\frac{1}{2}D_{\Delta}(\mathbb{P}, \mathbb{Q}) \leq D_{\mathsf{H}}^2(\mathbb{P}, \mathbb{Q}) \leq D_{\Delta}(\mathbb{P}, \mathbb{Q})$ (e.g., Foster and Krishnamurthy [32]). Next, define the shorthand $P(x_{1:H} \mid a_{1:H}) := \prod_{h=0}^{H-1} P(x_{h+1} \mid x_h, a_h)$ and $P^{\pi}(a_{1:H} \mid x_{1:H}) := \prod_{h=1}^{H} \pi_h(a_h \mid x_h)$ (these quantities do not have an interpretation as conditional probability measures in the way the notation might suggest, but this will not be relevant to the proof). For any deterministic policy $\pi^{\star}$, we can write

$$D_{\Delta}\left(\mathbb{P}^{\pi}, \mathbb{P}^{\pi^{\star}}\right)$$

$$= \sum_{x_{1:H}} \sum_{a_{1:H}} P(x_{1:H} \mid a_{1:H-1}) \cdot \frac{(P^{\pi}(a_{1:H} \mid x_{1:H}) - P^{\pi^{\star}}(a_{1:H} \mid x_{1:H}))^2}{P^{\pi}(a_{1:H} \mid x_{1:H}) + P^{\pi^{\star}}(a_{1:H} \mid x_{1:H})}$$

$$= \sum_{x_{1:H}} \sum_{a_{1:H} = \pi^{\star}(x_{1:H})} P(x_{1:H} \mid a_{1:H-1}) \cdot \frac{(P^{\pi}(a_{1:H} \mid x_{1:H}) - P^{\pi^{\star}}(a_{1:H} \mid x_{1:H}))^2}{P^{\pi}(a_{1:H} \mid x_{1:H}) + P^{\pi^{\star}}(a_{1:H} \mid x_{1:H})}$$

$$+ \sum_{x_{1:H}} \sum_{a_{1:H} \neq \pi^{\star}(x_{1:H})} P(x_{1:H} \mid a_{1:H-1}) \cdot \frac{(P^{\pi}(a_{1:H} \mid x_{1:H}) - P^{\pi^{\star}}(a_{1:H} \mid x_{1:H}))^2}{P^{\pi}(a_{1:H} \mid x_{1:H}) + P^{\pi^{\star}}(a_{1:H} \mid x_{1:H})}.$$

Since $\pi^{\star}$ is deterministic, $P^{\pi^{\star}}(a_{1:H} \mid x_{1:H}) = 1$ if $a_{1:H} = \pi^{\star}(x_{1:H})$, and is $P^{\pi^{\star}}(a_{1:H} \mid x_{1:H}) = 0$ otherwise. Using this, we can write the second term above as

$$\sum_{x_{1:H}} \sum_{a_{1:H} \neq \pi^{\star}(x_{1:H})} P(x_{1:H} \mid a_{1:H-1}) \cdot \frac{(P^{\pi}(a_{1:H} \mid x_{1:H}) - 0)^2}{P^{\pi}(a_{1:H} \mid x_{1:H}) + 0}$$

$$= \sum_{x_{1:H}} \sum_{a_{1:H} \neq \pi^{\star}(x_{1:H})} P(x_{1:H} \mid a_{1:H-1})P^{\pi}(a_{1:H} \mid x_{1:H})$$

$$= \mathbb{P}^{\pi}[\exists h : a_h \neq \pi^{\star}(x_H)] = \rho(\pi \parallel \pi^{\star}).$$

This proves that $D_\Delta\left(\mathbb{P}^\pi, \mathbb{P}^{\pi^\star}\right) \geq \rho(\pi \parallel \pi^\star)$. For the upper bound, we use that $\pi^\star$ is deterministic once more to write the first term above as

$$\sum_{x_{1:H}} \sum_{a_{1:H} = \pi^\star(x_{1:H})} P(x_{1:H} \mid a_{1:H-1}) \cdot \frac{(P^\pi(a_{1:H} \mid x_{1:H}) - 1)^2}{P^\pi(a_{1:H} \mid x_{1:H}) + 1}$$

$$= \mathbb{E}^{\pi^\star}\left[\frac{(P^\pi(a_{1:H} \mid x_{1:H}) - 1)^2}{P^\pi(a_{1:H} \mid x_{1:H}) + 1}\right] \leq \mathbb{E}^{\pi^\star}\left[(P^\pi(a_{1:H} \mid x_{1:H}) - 1)^2\right].$$

We further note that

$$\mathbb{E}^{\pi^\star}\left[(P^\pi(a_{1:H} \mid x_{1:H}) - 1)^2\right]$$

$$= \mathbb{E}^{\pi^\star} \mathbb{E}_{a'_{1:H} \sim \pi(x_{1:H})}[1 + (P^\pi(a'_{1:H} \mid x_{1:H}) - 2)\mathbb{I}\{a'_{1:H} = a_{1:H}\}]$$

$$\leq \mathbb{E}^{\pi^\star} \mathbb{E}_{a'_{1:H} \sim \pi(x_{1:H})}[1 - \mathbb{I}\{a'_{1:H} = a_{1:H}\}]$$

$$= \mathbb{E}^{\pi^\star} \mathbb{E}_{a'_{1:H} \sim \pi(x_{1:H})}[\mathbb{I}\{\exists h : a'_{1:H} \neq a_{1:H}\}] = \rho(\pi^\star \parallel \pi).$$

By Lemma F.1, we conclude that $D_\Delta\left(\mathbb{P}^\pi, \mathbb{P}^{\pi^\star}\right) \leq \rho(\pi \parallel \pi^\star) + \rho(\pi^\star \parallel \pi) = 2\rho(\pi^\star \parallel \pi)$. $\qquad\square$

### F.2  Proof of Theorem 2.2

**Proof of Theorem 2.2.**    For this proof, we consider a slightly more general online imitation learning model in which the learner is allowed to select $a_h^i$ based on the sequence $(x_1^i, a_1^i, a_1^{\star,i}), \ldots, (x_{h-1}^i, a_{h-1}^i, a_{h-1}^{\star,i}), (x_h^i, a_h^{\star,i})$ at training time; this subsumes the offline imitation learning model. Let $n \in \mathbb{N}$ and $H \in \mathbb{N}$ be fixed. Let $\Delta \in (0, 1/3)$ be a parameter whose value will be chosen later.

We first specify the dynamics for the reward-free MDP $M^\star$ and the policy class $\Pi$. Set $\mathcal{X} = \{\mathbf{x}, \mathbf{y}\}$ and $\mathcal{A} = \{\mathfrak{a}, \mathfrak{b}\}$. The initial state distribution sets $P_0(\mathbf{x}) = 1 - \Delta$ and $P_0(\mathbf{y}) = \Delta$. The transition dynamics are $P_h(x' \mid x, a) = \mathbb{I}\{x' = x\}$ for all $h$; that is, $\mathbf{x}, \mathbf{y}$ are self-looping terminal states. We set $\Pi = \{\pi^\mathfrak{a}, \pi^\mathfrak{b}\}$, where the expert policies are $\pi^\mathfrak{a}$, which sets $\pi_h^\mathfrak{a}(x) = \mathfrak{a}$ for all $h$ and $x$, and $\pi^\mathfrak{b}$, which sets $\pi_h^\mathfrak{b}(\mathbf{x}) = \mathfrak{a}$ and sets $\pi_h^\mathfrak{b}(\mathbf{y}) = \mathfrak{b}$.

Let a *problem instance* $\mathcal{I} = (M^\star, r, \pi^\star)$ refer to a tuple consisting of the reward-free MDP $M^\star$, a reward function $r = \{r_h\}_{h=1}^H$, and an expert policy $\pi^\star$. We consider two problem instances, $\mathcal{I}^\mathfrak{a} = (M^\star, r^\mathfrak{a}, \pi^\mathfrak{a})$ and $\mathcal{I}^\mathfrak{b} = (M^\star, r^\mathfrak{b}, \pi^\mathfrak{b})$:

- For problem instance $\mathcal{I}^\mathfrak{a}$, the expert policy is $\pi^\mathfrak{a}$. We set $r_h^\mathfrak{a}(\mathbf{x}, \cdot) = 0$, $r_h^\mathfrak{a}(\mathbf{y}, a) = \mathbb{I}\{a = \mathfrak{a}\}$ for all $h$.

- For problem instance $\mathcal{I}^\mathfrak{b}$, the expert policy is $\pi^\mathfrak{b}$. We set $r_h^\mathfrak{b}(\mathbf{x}, \cdot) = 0$, $r_h^\mathfrak{b}(\mathbf{y}, a) = \mathbb{I}\{a = \mathfrak{b}\}$ for all $h$.

Note that both of these instances satisfy $\mu = 1$, and that $\pi^\mathfrak{a}$ and $\pi^\mathfrak{b}$ are optimal policies for their respective instances. Let $J^\mathfrak{a}$ denote the expected reward function for instance $\mathcal{I}^\mathfrak{a}$, and likewise for $\mathcal{I}^\mathfrak{b}$.

Going forward, we fix the online imitation learning algorithm under consideration and let $\mathbb{P}^\mathfrak{a}$ denote the law of $o^1, \ldots, o^n$ when we execute the algorithm on instance $\mathfrak{a}$, and likewise for $\mathfrak{b}$; let $\mathbb{E}^\mathfrak{a}[\cdot]$ and $\mathbb{E}^\mathfrak{b}[\cdot]$ denote the corresponding expectations. In addition, for any policy $\pi$, let $\mathbb{P}^{\pi^\mathfrak{a}|\pi}$ denote the law of $o = (x_1, a_1, a_1^\star), \ldots, (x_H, a_H, a_H^\star)$ when we execute $\pi$ in the online imitation learning framework and the expert policy is $\pi^\star = \pi^\mathfrak{a}$, and define $\mathbb{P}^{\pi^\mathfrak{b}|\pi}$ analogously.

We first observe that for any policy $\widehat{\pi}$,

$$J^\mathfrak{a}(\pi^\mathfrak{a}) - J^\mathfrak{a}(\widehat{\pi}) = \Delta \cdot \sum_{h=1}^H \mathbb{E}_{a_h \sim \widehat{\pi}_h(\mathbf{y})}[\mathbb{I}\{a_h \neq \pi_h^\mathfrak{a}(\mathbf{y})\}],$$

and that $J^\mathfrak{b}(\pi^\mathfrak{b}) - J^\mathfrak{b}(\widehat{\pi}) = \Delta \cdot \sum_{h=1}^H \mathbb{E}_{a_h \sim \widehat{\pi}_h(\mathbf{y})}[\mathbb{I}\{a_h \neq \pi_h^\mathfrak{b}(\mathbf{y})\}]$. Defining $\rho(\pi, \pi') = \sum_{h=1}^H \mathbb{E}_{a_h \sim \pi_h(\mathbf{y}), a'_h \sim \pi'_h(\mathbf{y})} \mathbb{I}\{a_h \neq a'_h\}$ as a metric, we note that $\rho(\pi^\mathfrak{a}, \pi^\mathfrak{b}) = H$, and hence by the standard Le Cam two-point argument (e.g.,. Wainwright [91]), the algorithm must have

$$\max\{\mathbb{E}^\mathfrak{a}[J^\mathfrak{a}(\pi^\mathfrak{a}) - J^\mathfrak{a}(\widehat{\pi})], \mathbb{E}^\mathfrak{b}[J^\mathfrak{b}(\pi^\mathfrak{b}) - J^\mathfrak{b}(\widehat{\pi})]\} \geq \frac{\Delta H}{4}(1 - D_{\mathsf{TV}}(\mathbb{P}^\mathfrak{a}, \mathbb{P}^\mathfrak{b})),$$

where $D_{\mathsf{TV}}(\cdot, \cdot)$ denotes total variation distance. Next, using Lemma D.2 of Foster et al. [36], we can bound

$$D_{\mathsf{TV}}^2(\mathbb{P}^{\mathtt{a}}, \mathbb{P}^{\mathtt{b}}) \leq D_{\mathsf{H}}^2(\mathbb{P}^{\mathtt{a}}, \mathbb{P}^{\mathtt{b}}) \leq 7\,\mathbb{E}^{\mathtt{a}}\left[\sum_{i=1}^{n} D_{\mathsf{H}}^2\left(\mathbb{P}^{\pi^{\mathtt{a}}|\pi^i}, \mathbb{P}^{\pi^{\mathtt{b}}|\pi^i}\right)\right].$$

Since, the feedback the learner receives for a given episode $i$ is identical under instances $\mathcal{I}^{\mathtt{a}}$ and $\mathcal{I}^{\mathtt{b}}$ unless $x_1 = \mathfrak{y}$ (regardless of how $\pi^i$ is chosen), we can bound

$$D_{\mathsf{H}}^2\left(\mathbb{P}^{\pi^{\mathtt{a}}|\pi^i}, \mathbb{P}^{\pi^{\mathtt{b}}|\pi^i}\right) \leq 2\Delta,$$

and hence

$$D_{\mathsf{TV}}^2(\mathbb{P}^{\mathtt{a}}, \mathbb{P}^{\mathtt{b}}) \leq 14\Delta n.$$

We set $\Delta = 1/56n$, and conclude that any algorithm must have

$$\max\{\mathbb{E}^{\mathtt{a}}[J^{\mathtt{a}}(\pi^{\mathtt{a}}) - J^{\mathtt{a}}(\widehat{\pi})], \mathbb{E}^{\mathtt{b}}[J^{\mathtt{b}}(\pi^{\mathtt{b}}) - J^{\mathtt{b}}(\widehat{\pi})]\} \geq \frac{\Delta H}{8} = c \cdot \frac{H}{n}$$

for an absolute constant $c > 0$.

$\square$

# G   Proofs from Section 3

## G.1   Proof of Theorem 3.1

**Proof of Theorem 3.1.** Assume without loss of generality that $R = 1$. Let $o = (x_1, a_1), \ldots, (x_H, a_H)$, and for each $h \in [H]$, define the sum of advantages up to step $h$ via

$$\Delta_h(o) = \sum_{\ell=1}^{h}\left(Q_\ell^{\pi^\star}(x_\ell, \pi_\ell^\star(x_\ell)) - Q_\ell^{\pi^\star}(x_\ell, a_\ell)\right),$$

which has $|\Delta(o)| \leq H$ almost surely. Consider the filtration $\mathscr{F}_h := \sigma(x_1, a_1, \ldots, x_h, a_h)$. Fix a parameter $L \geq 1$ whose value will be chosen later, and define a random variable

$$H^\star := \min\{h \mid |\Delta_h(o)| > L\},$$

with $H^\star := H + 1$ if there is no $h$ such that $|\Delta_h(o)| > L$; we will adopt the convention that $Q_{H+1}^{\pi^\star} = V_{H+1}^{\pi^\star} = 0$.

**Lemma G.1.** $H^\star$ *is a stopping time with respect* $(\mathscr{F}_h)_{h\geq 1},$[17] *and has* $|\Delta_{H^\star}(o)| \leq L + 1$ *almost surely.*

The following lemma, which is one of the central technical components of this proof, gives a bound on regret in terms of the expected advantage at the stopping time $H^\star$. We use the stopping time to keep the sum of advantages $\Delta_{H^\star}$ bounded, which facilitates a strong change-of-measure argument in the sequel.

**Lemma G.2** (Regret decomposition for stopped advantages). *If* $r_h \geq 0$ *and* $\sum_{h=1}^{H} r_h \in [0, R]$*, then for all policies* $\widehat{\pi}$*, we have that*

$$J(\pi^\star) - J(\widehat{\pi}) \leq \mathbb{E}^{\widehat{\pi}}[\Delta_{H^\star}(o)] + R \cdot \mathbb{P}^{\widehat{\pi}}[H^\star \leq H]. \tag{16}$$

Note that even though we assume $R = 1$ throughout this proof, we state this lemma for general $R$ for the sake of keeping it self-contained.

We proceed to bound the right-hand-side of Eq. (16) using change-of-measure based on Hellinger distance (Lemma 3.1). For the second term in Eq. (16), Lemma 3.1 gives

$$\mathbb{P}^{\widehat{\pi}}[H^\star \leq H] \leq 2\mathbb{P}^{\pi^\star}[H^\star \leq H] + D_{\mathsf{H}}^2\left(\mathbb{P}^{\widehat{\pi}}, \mathbb{P}^{\pi^\star}\right)$$

$$= 2\mathbb{P}^{\pi^\star}[\exists h : |\Delta_h(o)| > L] + D_{\mathsf{H}}^2\left(\mathbb{P}^{\widehat{\pi}}, \mathbb{P}^{\pi^\star}\right).$$

---

[17]That is, for all $h$, $\mathbb{I}\{h = H^\star\}$ is a measurable function of $(x_1, a_1), \ldots, (x_h, a_h)$.

For the first term in Eq. (16), Lemma 3.1, gives that

$$\mathbb{E}^{\widehat{\pi}}[\Delta_{H^\star}(o)] \leq \mathbb{E}^{\pi^\star}[\Delta_{H^\star}(o)] + \sqrt{\tfrac{1}{2}\Big(\mathbb{E}^{\widehat{\pi}}[\Delta^2_{H^\star}(o)] + \mathbb{E}^{\pi^\star}[\Delta^2_{H^\star}(o)]\Big) \cdot D^2_{\mathsf{H}}(\mathbb{P}^{\widehat{\pi}}, \mathbb{P}^{\pi^\star})}.$$

To bound the first moment and second moment of $\Delta_{H^\star}(o)$ under $\pi^\star$, we use the following lemma, which follows from elementary properties of stopped martingale difference sequences.

**Lemma G.3.** *We have that*

$$\mathbb{E}^{\pi^\star}[\Delta_{H^\star}(o)] \leq 0, \quad and \quad \mathbb{E}^{\pi^\star}\big[\Delta^2_{H^\star}(o)\big] \leq 4\sigma^2_{\pi^\star}.$$

It remains to bound the second moment under $\widehat{\pi}$. Here, since $|\Delta_{H^\star}(o)| \leq L + 1$ almost surely by Lemma G.1, we note that Lemma 3.1 gives

$$\mathbb{E}^{\widehat{\pi}}\big[\Delta^2_{H^\star}(o)\big] \leq 2\,\mathbb{E}^{\pi^\star}\big[\Delta^2_{H^\star}(o)\big] + (L+1)^2 D^2_{\mathsf{H}}\Big(\mathbb{P}^{\widehat{\pi}}, \mathbb{P}^{\pi^\star}\Big).$$

Combining these developments, we have that

$$\mathbb{E}^{\widehat{\pi}}[\Delta_{H^\star}(o)] \leq \sqrt{\tfrac{3}{2}\,\mathbb{E}^{\pi^\star}[\Delta^2_{H^\star}(o)] \cdot D^2_{\mathsf{H}}(\mathbb{P}^{\widehat{\pi}}, \mathbb{P}^{\pi^\star})} + (L+1)D^2_{\mathsf{H}}\Big(\mathbb{P}^{\widehat{\pi}}, \mathbb{P}^{\pi^\star}\Big)$$

$$\leq \sqrt{6\sigma^2_{\pi^\star} \cdot D^2_{\mathsf{H}}(\mathbb{P}^{\widehat{\pi}}, \mathbb{P}^{\pi^\star})} + (L+1)D^2_{\mathsf{H}}\Big(\mathbb{P}^{\widehat{\pi}}, \mathbb{P}^{\pi^\star}\Big),$$

and thus

$$J(\pi^\star) - J(\widehat{\pi}) \leq \sqrt{6\sigma^2_{\pi^\star} \cdot D^2_{\mathsf{H}}(\mathbb{P}^{\widehat{\pi}}, \mathbb{P}^{\pi^\star})} + (L+2)D^2_{\mathsf{H}}\Big(\mathbb{P}^{\widehat{\pi}}, \mathbb{P}^{\pi^\star}\Big) + 2\mathbb{P}^{\pi^\star}[\exists h : |\Delta_h(o)| > L].$$

To wrap up, we appeal to the second of our main technical lemmas, Lemma G.4.

**Lemma G.4** (Concentration for advantages). *Assume that $r_h \geq 0$ and $\sum_{h=1}^{H} r_h \in [0, R]$ almost surely for some $R > 0$. Then for any (potentially stochastic) policy $\pi$, it holds that for all $\delta \in (0, e^{-1})$,*

$$\mathbb{P}^{\pi}\left[\exists H' : \left|\sum_{h=1}^{H'} Q^\pi_h(x_h, a_h) - V^\pi_h(x_h)\right| \geq c \cdot R\log(\delta^{-1})\right] \leq \delta,$$

*for an absolute constant $c > 0$.*

Let $\varepsilon \in (0, e^{-1})$ be fixed. If we define

$$L = c \cdot \log(\varepsilon^{-1}),$$

where $c > 1$ is a sufficiently large absolute constant, then by Lemma G.4, we have that

$$\mathbb{P}^{\pi^\star}[\exists h : |\Delta_h(o)| > L] \leq \varepsilon.$$

This proves the result.

$\square$

**Proof of Lemma G.1.** To prove that $H^\star$ is a stopping time, we observe that for all $h \leq H$, we have

$$\mathbb{I}\{h = H^\star\} = \mathbb{I}\{|\Delta_h(o)| > L, |\Delta_{h'}(o)| \leq L \;\forall h' < h\},$$

and $\Delta_h(o)$ is a measurable function of $(x_1, a_1), \ldots, (x_h, a_h)$. Likewise, we have

$$\mathbb{I}\{h = H^\star + 1\} = \mathbb{I}\{|\Delta_h(o)| \leq L \;\forall h \leq H\},$$

which is a measurable function of $(x_1, a_1), \ldots, (x_H, a_H)$.

For the second claim, we observe that

$$|\Delta_{H^\star}(o)| \leq |\Delta_{H^\star - 1}(o)| + \left|Q^{\pi^\star}_{H^\star}(x_{H^\star}, \pi^\star_{H^\star}(x_{H^\star})) - Q^{\pi^\star}_{H^\star}(x_{H^\star}, a_{H^\star})\right|$$

$$\leq L + 1$$

almost surely.

$\square$

**Proof of Lemma G.3.** Define $X_h := Q_h^{\pi^\star}(x_h, \pi_h^\star(x_h)) - Q_h^{\pi^\star}(x_h, a_h)$, and $\mathscr{F}_h = \sigma(x_1, a_1, \ldots, x_h, a_h)$, with $X_{H+1} := 0$. Since $H^\star$ is a stopping time with respect to $(\mathscr{F}_h)$ and $X_h$ is a martingale difference sequence (under $\pi^\star$), the optional stopping theorem (e.g., [96]) implies that[18]

$$\mathbb{E}^{\pi^\star}[\Delta_{H^\star}(o)] = \mathbb{E}^{\pi^\star}\left[\sum_{h=1}^{H^\star} X_h\right] = 0.$$

We now bound the second moment. Recall Doob's maximal inequality (e.g., Williams [96]).

**Lemma G.5.** *If $(S_h)_{h\in[H]}$ is a non-negative submartingale, then*

$$\mathbb{E}\left[\max_{h\in[H]} S_h^2\right] \leq 4\,\mathbb{E}\left[S_H^2\right].$$

We claim that $|\Delta_h(o)|$ is a submartingale, since a convex function of a martingale is a submartingale.[19] As a result, Lemma G.5 gives that

$$\mathbb{E}\left[\Delta_{H^\star}^2(o)\right] \leq \mathbb{E}\left[\max_{h\in[H]} \Delta_h^2(o)\right] \leq 4\,\mathbb{E}\left[\Delta_H^2(o)\right].$$

Finally, we note that

$$\mathbb{E}^{\pi^\star}\left[\Delta_H^2(o)\right] = \mathbb{E}^{\pi^\star}\left[\left(\sum_{h=1}^{H}\left(Q_h^{\pi^\star}(x_h, \pi_h^\star(x_h)) - Q_h^{\pi^\star}(x_h, a_h)\right)\right)^2\right]$$

$$= \sum_{h=1}^{H} \mathbb{E}^{\pi^\star}\left[(Q_h^{\pi^\star}(x_h, \pi_h^\star(x_h)) - Q_h^{\pi^\star}(x_h, a_h))^2\right] = \sigma_{\pi^\star}^2,$$

where we have once more used that $X_h = Q_h^{\pi^\star}(x_h, \pi_h^\star(x_h)) - Q_h^{\pi^\star}(x_h, a_h)$ is a martingale difference sequence.

$\square$

### G.1.1 Proof of Lemma G.2 (Regret Decomposition for Stopped Advantages)

**Proof of Lemma G.2.** Consider the following non-Markovian policy:

$$\widetilde{\pi}_h(\cdot \mid x_{1:h}, a_{1:h-1}) = \begin{cases} \widehat{\pi}_h(\cdot \mid x_h) & h \leq H^\star, \\ \pi_h^\star(\cdot \mid x_h) & h > H^\star. \end{cases}$$

This is a well-defined policy, since we can write $\mathbb{I}\{h > H^\star\} = \max_{h' < h}\mathbb{I}\{h' = H^\star\}$, and $\mathbb{I}\{h' = H^\star\}$ is a measurable function of $(x_1, a_1), \ldots, (x_{h'}, a_{h'}) \subset (x_1, a_1), \ldots, (x_{h-1}, a_{h-1})$ for $h' < h$.

We begin by writing

$$J(\pi^\star) - J(\widehat{\pi}) = J(\pi^\star) - J(\widetilde{\pi}) + J(\widetilde{\pi}) - J(\widehat{\pi}). \tag{17}$$

For the second pair of terms in Eq. (17), we use the following lemma.

**Lemma G.6.** *Under the same assumptions as Lemma G.2, it holds that*

$$J(\widetilde{\pi}) - J(\widehat{\pi}) \leq R \cdot \mathbb{P}^{\widehat{\pi}}[H^\star \leq H].$$

---

[18]To give self-contained proof, note that we can write $\mathbb{E}^{\pi^\star}\left[\sum_{h=1}^{H^\star} X_h\right] = \mathbb{E}^{\pi^\star}\left[\sum_{h=1}^{H} X_h\mathbb{I}\{H^\star \geq h\}\right]$ We claim that $\mathbb{I}\{H^\star \geq h\}$ is a measurable function of $\mathscr{F}_{h-1}$, since $\mathbb{I}\{H^\star \geq h\} = 1 - \mathbb{I}\{H^\star < h\}$, and $\mathbb{I}\{H^\star = h'\}$ is a measurable function of $(x_1, a_1), \ldots, (x_{h'}, a_{h'}) \subset (x_1, a_1), \ldots, (x_{h-1}, a_{h-1})$ for $h' < h$. We conclude that $\mathbb{E}^{\pi^\star}[X_h\mathbb{I}\{H^\star \geq h\} \mid \mathscr{F}_{h-1}] = \mathbb{E}^{\pi^\star}[X_h \mid \mathscr{F}_{h-1}]\mathbb{I}\{H^\star \geq h\} = 0.$

[19]For completeness, note that $\mathbb{E}[|\Delta_h(o)| \mid \mathscr{F}_{h-1}] = \mathbb{E}[|\Delta_{h-1}(o) + X_h| \mid \mathscr{F}_{h-1}] \geq |\Delta_{h-1}(o) + \mathbb{E}[X_h \mid \mathscr{F}_{h-1}]| = |\Delta_{h-1}(o)|.$

For the first pair of terms in Eq. (17), using the performance difference lemma, we can write[20]

$$
\begin{aligned}
J(\pi^\star) - J(\widetilde{\pi}) &= \mathbb{E}^{\widetilde{\pi}}\left[\sum_{h=1}^{H} Q_h^{\pi^\star}(x_h, \pi_h^\star(x_h)) - Q_h^{\pi^\star}(x_h, a_h)\right] \\
&= \mathbb{E}^{\widetilde{\pi}}\left[\sum_{h=1}^{H} \mathbb{E}_{h-1}\left[Q_h^{\pi^\star}(x_h, \pi_h^\star(x_h)) - Q_h^{\pi^\star}(x_h, a_h)\right]\right] \\
&= \mathbb{E}^{\widetilde{\pi}}\left[\sum_{h=1}^{H} \mathbb{E}_{h-1}\left[Q_h^{\pi^\star}(x_h, \pi_h^\star(x_h)) - Q_h^{\pi^\star}(x_h, a_h)\right]\mathbb{I}\{h \le H^\star\}\right] \\
&= \mathbb{E}^{\widetilde{\pi}}\left[\sum_{h=1}^{H} \mathbb{E}_{h-1}\left[\left(Q_h^{\pi^\star}(x_h, \pi_h^\star(x_h)) - Q_h^{\pi^\star}(x_h, a_h)\right)\mathbb{I}\{h \le H^\star\}\right]\right] \\
&= \mathbb{E}^{\widetilde{\pi}}\left[\sum_{h=1}^{H^\star} Q_h^{\pi^\star}(x_h, \pi_h^\star(x_h)) - Q_h^{\pi^\star}(x_h, a_h)\right] = \mathbb{E}^{\widetilde{\pi}}[\Delta_{H^\star}(o)],
\end{aligned}
$$

where the third equality uses that $\widetilde{\pi}_h(\cdot \mid x_{1:h}, a_{1:h-1}) = \pi_h^\star(\cdot \mid x_h)$ for $h > H^\star$, and the fourth equality uses that $\mathbb{I}\{h \le H^\star\}$ is $\mathscr{F}_{h-1}$-measurable. We now appeal to the following lemma, proven in the sequel.

**Lemma G.7.** *Under the same assumptions as Lemma G.2, it holds that*

$$
\mathbb{E}^{\widetilde{\pi}}[\Delta_{H^\star}(o)] = \mathbb{E}^{\widehat{\pi}}[\Delta_{H^\star}(o)].
$$

Altogether, we conclude that

$$
J(\pi^\star) - J(\widehat{\pi}) \le \mathbb{E}^{\widehat{\pi}}[\Delta_{H^\star}(o)] + R \cdot \mathbb{P}^{\widehat{\pi}}[H^\star \le H].
$$

$\square$

**Proof of Lemma G.6.** Let us define $f(o) = \sum_{h=1}^{H} \mathbb{E}[r_h \mid x_h, a_h]$ and $g(o) = \mathbb{I}\{H^\star > H\}$; note that $g(o)$ is indeed a measurable function of $o = (x_1, a_1), \ldots, (x_H, a_H)$, since $\mathbb{I}\{H^\star > H\} = 1 - \mathbb{I}\{H^\star \le H\}$, $\{H^\star \le H\} = \cup_{h \le H}\{H^\star = h\}$, and $\{H^\star = h\}$ is a measurable function of $(x_1, a_1), \ldots, (x_h, a_h)$. We can write

$$
J(\widetilde{\pi}) \le \mathbb{E}^{\widetilde{\pi}}\left[\left(\sum_{h=1}^{H} r_h\right)\mathbb{I}\{H^\star > H\}\right] + R \cdot \mathbb{P}^{\widetilde{\pi}}[H^\star \le H]. \tag{18}
$$

Let us adopt the shorthand $P(x_{1:H} \mid a_{1:H-1}) := \prod_{h=0}^{H-1} P_h(x_{h+1} \mid x_h, a_h)$. We can bound the first term in Eq. (18) via

$$
\begin{aligned}
\mathbb{E}^{\widetilde{\pi}}\left[\left(\sum_{h=1}^{H} r_h\right)\mathbb{I}\{H^\star > H\}\right] &= \sum_{o=x_{1:H}, a_{1:H}} f(o)g(o)P(x_{1:H} \mid a_{1:H-1})\prod_{h=1}^{H} \widetilde{\pi}_h(a_h \mid x_{1:h}, a_{1:h-1}) \\
&= \sum_{o=x_{1:H}, a_{1:H}} f(o)g(o)P(x_{1:H} \mid a_{1:H-1})\prod_{h=1}^{H} \widehat{\pi}_h(a_h \mid x_h) \\
&\le \sum_{o=x_{1:H}, a_{1:H}} f(o)P(x_{1:H} \mid a_{1:H-1})\prod_{h=1}^{H} \widehat{\pi}_h(a_h \mid x_h) \\
&= \mathbb{E}^{\widehat{\pi}}\left[\sum_{h=1}^{H} r_h\right] = J(\widehat{\pi}),
\end{aligned}
$$

---

[20]Since $\widetilde{\pi}$ is non-Markovian, we need to expand the state space to $x_h' = x_{1:h}, a_{1:h-1}$ to apply the performance difference lemma, but since $\pi^\star$ itself is Markovian, this results in the claimed expression.

where the second equality uses that $\widetilde{\pi}(\cdot \mid x_{1:h}, a_{1:h-1}) = \widehat{\pi}(\cdot \mid x_h)$ for all $h \in [H]$ whenever $g(o) = 1$.

To bound the second term in Eq. (18), we can write

$$\mathbb{P}^{\widetilde{\pi}}[H^\star \leq H] = \sum_{h=1}^{H} \mathbb{P}^{\widetilde{\pi}}[H^\star = h].$$

For each $h$, let $o_h := (x_1, a_1), \ldots, (x_h, a_h)$ and $g_h(o_h) := \mathbb{I}\{H^\star = h\}$ (recall that $\mathbb{I}\{H^\star = h\}$ is a measurable function of $(x_1, a_1), \ldots, (x_h, a_h)$). Note that for each $h$, if we define $P(x_{1:h} \mid a_{1:h-1}) := \prod_{h=0}^{h-1} P_\ell(x_{\ell+1} \mid x_\ell, a_\ell)$, then

$$\mathbb{P}^{\widetilde{\pi}}[H^\star = h] = \sum_{o_h = x_{1:h}, a_{1:h}} g_h(o_h) P(x_{1:h} \mid a_{1:h-1}) \prod_{\ell=1}^{h} \widetilde{\pi}_\ell(a_\ell \mid x_{1:\ell}, a_{1:\ell-1})$$

$$= \sum_{o_h = x_{1:h}, a_{1:h}} g_h(o_h) P(x_{1:h} \mid a_{1:h-1}) \prod_{\ell=1}^{h} \widehat{\pi}_\ell(a_\ell \mid x_\ell)$$

$$= \mathbb{P}^{\widehat{\pi}}[H^\star = h],$$

where the second inequality uses that $\widetilde{\pi}(\cdot \mid x_{1:\ell}, a_{1:\ell-1}) = \widehat{\pi}(\cdot \mid x_\ell)$ whenever $\ell \leq H^\star$. $\qquad\square$

**Proof of Lemma G.7.** We start by writing

$$\mathbb{E}^{\widetilde{\pi}}[\Delta_{H^\star}(o)] = \sum_{h=1}^{H+1} \mathbb{E}^{\widetilde{\pi}}[\mathbb{I}\{H^\star = h\}\Delta_h(o)].$$

For each $h \leq H + 1$, let $o_h := (x_1, a_1), \ldots, (x_h, a_h)$ and $g_h(o_h) := \mathbb{I}\{H^\star = h\}$ (recall that $\mathbb{I}\{H^\star = h\}$ is a measurable function of $(x_1, a_1), \ldots, (x_h, a_h)$). For each $h \leq H + 1$, if we define $P(x_{1:h} \mid a_{1:h-1}) := \prod_{h=0}^{h-1} P_\ell(x_{\ell+1} \mid x_\ell, a_\ell)$, then

$$\mathbb{E}^{\widetilde{\pi}}[\mathbb{I}\{H^\star = h\}\Delta_h(o)] = \sum_{o_h = x_{1:h}, a_{1:h}} g_h(o_h) \Delta_h(o_h) P(x_{1:h} \mid a_{1:h-1}) \prod_{\ell=1}^{h} \widetilde{\pi}_\ell(a_\ell \mid x_{1:\ell}, a_{1:\ell-1})$$

$$= \sum_{o_h = x_{1:h}, a_{1:h}} g_h(o_h) \Delta_h(o_h) P(x_{1:h} \mid a_{1:h-1}) \prod_{\ell=1}^{h} \widehat{\pi}_\ell(a_\ell \mid x_\ell)$$

$$= \mathbb{E}^{\widehat{\pi}}[\mathbb{I}\{H^\star = h\}\Delta_h(o)],$$

where the second inequality uses that $\widetilde{\pi}(\cdot \mid x_{1:\ell}, a_{1:\ell-1}) = \widehat{\pi}(\cdot \mid x_\ell)$ whenever $\ell \leq H^\star$.

$\qquad\square$

### G.1.2 Proof of Lemma G.4 (Concentration for Advantages)

Lemma G.4 is proven using arguments similar to those in Zhang et al. [104, 105], but requires non-trivial modifications to accommodate the fact that $\pi$ is an arbitrary, potentially suboptimal policy.

**Proof of Lemma G.4.** Let us abbreviate $Q = Q^\pi$ and $V = V^\pi$. Assume without loss of generality that $R = 1$, and note that this implies that $r_h \in [0, 1]$ and $Q_h, V_h \in [0, 1]$, which we will use throughout the proof.

Define a filtration $\mathscr{F}_{h-1} := \sigma((x_1, a_1, r_1), \ldots, (x_{h-1}, a_{h-1}, r_{h-1}), x_h)$. Since

$$\mathbb{E}_{h-1}[Q_h(x_h, a_h) - V_h(x_h)] = 0,$$

two applications of Lemma D.2 and a union bound imply that with probability at least $1 - \delta$, for all $H' \in [H]$

$$\left| \sum_{h=1}^{H'} Q_h(x_h, a_h) - V_h(x_h) \right| \leq \sum_{h=1}^{H'} \mathbb{E}^\pi\left[(Q_h(x_h, a_h) - V_h(x_h))^2 \mid x_h\right] + \log(2\delta^{-1}).$$

Since $\mathbb{E}^\pi[Q_h(x_h, a_h) \mid x_h] = V_h(x_h)$, we can write

$$\sum_{h=1}^{H'} \mathbb{E}^\pi\big[(Q_h(x_h, a_h) - V_h(x_h))^2 \mid x_h\big] = \sum_{h=1}^{H'} \mathbb{E}^\pi\big[(Q_h^2(x_h, a_h) \mid x_h\big] - V_h^2(x_h)$$

$$= \sum_{h=1}^{H'} \big(\mathbb{E}^\pi\big[(Q_h^2(x_h, a_h) \mid x_h\big] - V_{h+1}^2(x_{h+1})\big) + V_{H'+1}^2(x_{H'+1}) - V_1^2(x_1)$$

$$\leq \sum_{h=1}^{H'} \big(\mathbb{E}^\pi\big[(Q_h^2(x_h, a_h) \mid x_h\big] - V_{h+1}^2(x_{h+1})\big) + 1.$$

Observe that by Jensen's inequality, we have

$$\mathbb{E}^\pi\big[(Q_h^2(x_h, a_h) \mid x_h\big] \leq \mathbb{E}^\pi\big[(r_h + V_{h+1}(x_{h+1}))^2 \mid x_h\big]$$

$$= \mathbb{E}^\pi\big[V_{h+1}^2(x_{h+1}) \mid x_h\big] + \mathbb{E}^\pi\big[r_h^2 \mid x_h\big] + 2\,\mathbb{E}^\pi[r_h V_{h+1}(x_{h+1}) \mid x_h]$$

$$\leq \mathbb{E}^\pi\big[V_{h+1}^2(x_{h+1}) \mid x_h\big] + 3\,\mathbb{E}^\pi[r_h \mid x_h],$$

so that

$$\sum_{h=1}^{H'} \mathbb{E}^\pi\big[(Q_h(x_h, a_h) - V_h(x_h))^2 \mid x_h\big] \leq \sum_{h=1}^{H'} \mathbb{E}^\pi\big[V_{h+1}^2(x_{h+1}) \mid x_h\big] - V_{h+1}^2(x_{h+1}) + 3\sum_{h=1}^{H'} \mathbb{E}^\pi[r_h \mid x_h] + 1.$$

By [Lemma D.3](), we have that with probability at least $1 - \delta$, for all $H' \in [H]$,

$$\sum_{h=1}^{H'} \mathbb{E}^\pi[r_h \mid x_h] \leq \frac{3}{2}\sum_{h=1}^{H'} r_h + 4\log(2\delta^{-1})$$

$$\leq \frac{3}{2} + 4\log(2\delta^{-1}).$$

Likewise, by [Lemma D.2](), we have that with probability at least $1 - \delta$, for all $H' \in [H]$,

$$\sum_{h=1}^{H'} \mathbb{E}^\pi\big[V_{h+1}^2(x_{h+1}) \mid x_h\big] - V_{h+1}^2(x_{h+1}) \leq \sum_{h=1}^{H'} \mathbb{E}^\pi\Big[\big(V_{h+1}^2(x_{h+1}) - \mathbb{E}^\pi\big[V_{h+1}^2(x_{h+1}) \mid x_h\big]\big)^2 \mid x_h\Big] + \log(\delta^{-1})$$

$$= \sum_{h=1}^{H'} \mathrm{Var}^\pi\big[V_{h+1}^2(x_{h+1}) \mid x_h\big] + \log(\delta^{-1})$$

$$\leq 4\sum_{h=1}^{H'} \mathrm{Var}^\pi[V_{h+1}(x_{h+1}) \mid x_h] + \log(\delta^{-1}),$$

where the last line uses the following lemma, proven in the sequel.

**Lemma G.8.** *If $X$ is a random variable with $|X| \leq 1$, then*

$$\mathrm{Var}(X^2) \leq 4\mathrm{Var}(X).$$

We now appeal to the following lemma, also proven in the sequel.

**Lemma G.9.** *Under the same setting as [Lemma G.4](), we have that for any $\delta \in (0, 1)$, with probability at least $1 - 2\delta$, for all $H' \in [H]$,*

$$\sum_{h=1}^{H'} \mathrm{Var}^\pi\big[V_{h+1}^\pi(x_{h+1}) \mid x_h\big] \leq 8 + 32\log(2\delta^{-1}).$$

Putting together all of the developments so far, we have that with probability at least $1 - 5\delta$, for all $H' \in [H]$,

$$\left|\sum_{h=1}^{H'} Q_h(x_h, a_h) - V_h(x_h)\right| \leq 4\sum_{h=1}^{H'} \mathrm{Var}^\pi[V_{h+1}(x_{h+1}) \mid x_h] + 6 + 14\log(2\delta^{-1})$$

$$\leq 38 + 142\log(2\delta^{-1}).$$

$\square$

**Proof of Lemma G.8.** Note that we have

$$\text{Var}(X^2) = \mathbb{E}\big[(X^2 - \mathbb{E}[X^2])^2\big] \le \mathbb{E}\Big[(X^2 - \mathbb{E}[X]^2)^2\Big] \le 4\,\mathbb{E}\big[(X - \mathbb{E}[X])^2\big],$$

where the last line uses that $\big|a^2 - b^2\big| \le 2|a - b|$ for $a, b \in [-1, 1]$. $\qquad\square$

**Proof of Lemma G.9.** Abbreviate $V \equiv V^\pi$. By telescoping, we can write

$$
\begin{aligned}
Z_{H'} &:= \sum_{h=1}^{H'} \text{Var}^\pi[V_{h+1}(x_{h+1}) \mid x_h] \\
&= \sum_{h=1}^{H'} \mathbb{E}^\pi\big[V_{h+1}^2(x_{h+1}) \mid x_h\big] - (\mathbb{E}^\pi[V_{h+1}(x_{h+1}) \mid x_h])^2 \\
&= \sum_{h=1}^{H'} \mathbb{E}^\pi\big[V_{h+1}^2(x_{h+1}) \mid x_h\big] - V_{h+1}^2(x_{h+1}) + \sum_{h=1}^{H'} V_h^2(x_h) - (\mathbb{E}^\pi[V_{h+1}(x_{h+1}) \mid x_h])^2 + V_{H'+1}^2(x_{H'+1}) - V_1^2(x_1) \\
&\le \sum_{h=1}^{H'} \mathbb{E}^\pi\big[V_{h+1}^2(x_{h+1}) \mid x_h\big] - V_{h+1}^2(x_{h+1}) + \sum_{h=1}^{H'} V_h^2(x_h) - (\mathbb{E}^\pi[V_{h+1}(x_{h+1}) \mid x_h])^2 + 1.
\end{aligned}
$$

For the latter term, since $\big|a^2 - b^2\big| \le 2|a - b|$ for $a, b \in [0, 1]$, we have that

$$
\begin{aligned}
\sum_{h=1}^{H'} V_h^2(x_h) - (\mathbb{E}^\pi[V_{h+1}(x_{h+1}) \mid x_h])^2 &\le 2\sum_{h=1}^{H'}|V_h(x_h) - \mathbb{E}^\pi[V_{h+1}(x_{h+1}) \mid x_h]| \\
&= 2\sum_{h=1}^{H'}|\mathbb{E}^\pi[r_h \mid x_h]| \le 2\sum_{h=1}^{H'} \mathbb{E}^\pi[r_h \mid x_h],
\end{aligned}
$$

By Lemma D.3, we have that with probability at least $1 - \delta$, for all $H' \in [H]$,

$$\sum_{h=1}^{H'}\mathbb{E}^\pi[r_h \mid x_h] \le \frac{3}{2}\sum_{h=1}^{H'} r_h + 4\log(2\delta^{-1}) \le \frac{3}{2} + 4\log(2\delta^{-1}).$$

For the first term, by Lemma D.2, we have that for all $\eta \in (0, 1)$, with probability at least $1 - \delta$, for all $H' \in [H]$,

$$
\begin{aligned}
\sum_{h=1}^{H'}\mathbb{E}^\pi\big[V_{h+1}^2(x_{h+1}) \mid x_h\big] - V_{h+1}(x_{h+1}) &\le \eta\sum_{h=1}^{H'}\mathbb{E}^\pi\Big[\big(V_{h+1}^2(x_{h+1}) - \mathbb{E}^\pi\big[V_{h+1}^2(x_{h+1}) \mid x_h\big]\big)^2 \mid x_h\Big] + \eta^{-1}\log(\delta^{-1}) \\
&= \eta\sum_{h=1}^{H'}\text{Var}^\pi\big[V_{h+1}^2(x_{h+1}) \mid x_h\big] + \eta^{-1}\log(\delta^{-1}) \\
&\le 4\eta\sum_{h=1}^{H'}\text{Var}^\pi[V_{h+1}(x_{h+1}) \mid x_h] + \eta^{-1}\log(\delta^{-1}) \\
&= 4\eta Z_{H'} + \eta^{-1}\log(\delta^{-1}),
\end{aligned}
$$

where the last inequality uses Lemma G.8. Putting everything together and setting $\eta = 1/8$, we conclude that with probability at least $1 - 2\delta$, for all $H' \in [H]$

$$Z_{H'} \le \frac{1}{2}Z_{H'} + 16\log(2\delta^{-1}) + 4,$$

which yields the result after rearranging.

$\qquad\square$

## G.2 Formal Statement and Proof of Theorem G.1

The following result shows that the dependence on the variance in Corollary 2.1 cannot be improved in general, which implies that the horizon-dependence in this regime is tight.

**Theorem G.1** (Lower bound for stochastic experts)**.** *Consider the dense reward setting where $r_h \in [0,1]$ and $R = H$. For any $n \in \mathbb{N}$, $H \in \mathbb{N}$ and $\sigma^2 \in [H, H^2]$, there exists a reward-free MDP $M^\star$ with $|\mathcal{X}| = 3$ and $|\mathcal{A}| = 2$, a class of reward functions $\mathcal{R}$ with $|\mathcal{R}| = 2$, and a class of policies $\Pi$ with $|\Pi| = 2$ with the following property. For any (online or offline) imitation learning algorithm, there exists a deterministic reward function $r = \{r_h\}_{h=1}^H$ and expert policy $\pi^\star \in \Pi$ such that $\sigma^2_{\pi^\star} \leq \sigma^2$ and $\widetilde{\mu} \leq \sigma^2/H$, and for which*

$$\mathbb{P}\left( J(\pi^\star) - J(\widehat{\pi}) \geq c \cdot \sqrt{\frac{\sigma^2}{n}} \right) \geq \frac{1}{8}$$

*for an absolute constant $c \geq 1$.*

Beyond showing that a slow $1/\sqrt{n}$ rate is required for stochastic policies,[21]

**Proof of Theorem G.1.**     For this proof, we consider a slightly more general online imitation learning model in which the learner is allowed to select $a_h^i$ based on the sequence $(x_1^i, a_1^i, a_1^{\star,i}), \ldots, (x_{h-1}^i, a_{h-1}^i, a_{h-1}^{\star,i}), (x_h^i, a_h^{\star,i})$ at training time; this subsumes the offline imitation learning model. Let $H \in \mathbb{N}$, $n \in \mathbb{N}$, and $\sigma^2 \in [H, H^2]$ be given. Fix a parameter $K \in \mathbb{N}$ such that $H/K$ is an integer and a parameter $\Delta \in (0, 1/2)$ be fixed; both parameters will be chosen at the end of the proof.

We first specify the dynamics for the reward-free MDP $M^\star$ and the policy class $\Pi$. Let $\mathcal{A} = \{\mathfrak{a}, \mathfrak{b}\}$, and let $\mathcal{X} = \{\mathfrak{s}, \mathfrak{a}, \mathfrak{b}\}$. We consider the following (deterministic) dynamics. For $h \in \mathcal{H} := [1, K+1, 2K+1, \ldots]$, always the state is always $x_h = \mathfrak{s}$. For such a step $h \in \mathcal{H}$, choosing $a_h = \mathfrak{a}$ sets $x_h = \mathfrak{a}$ for the next $K-1$ steps until returning to $\mathfrak{s}$ at time $h+K$, and choosing $a_h = \mathfrak{b}$ sets $x_h = \mathfrak{b}$ until returning to $\mathfrak{s}$ at time $h+K$ (that is, the action has no effect for $h \notin \mathcal{H}$).

We consider a class $\Pi = \{\pi^{\mathfrak{a}}, \pi^{\mathfrak{b}}\}$ consisting of two experts $\pi^{\mathfrak{a}}$ and $\pi^{\mathfrak{b}}$. $\pi^{\mathfrak{a}}$ sets $\pi_h^{\mathfrak{a}}(\mathfrak{a} \mid \mathfrak{s}) = \frac{1}{2} + \Delta$ for $h \in \mathcal{H}$ and sets $\pi_h(x) = \mathfrak{a}$ for all $h \notin \mathcal{H}$ and $x \in \mathcal{X}$. Meanwhile, $\pi^{\mathfrak{b}}$ sets $\pi^{\mathfrak{b}}(\mathfrak{b} \mid \mathfrak{s}) = \frac{1}{2} + \Delta$ for $h \in \mathcal{H}$ and sets $\pi_h(x) = \mathfrak{a}$ for all $h \notin \mathcal{H}$ and $x \in \mathcal{X}$.

We consider two choices of reward function, $r^{\mathfrak{a}}$ and $r^{\mathfrak{b}}$. $r^{\mathfrak{a}}$ sets $r_h^{\mathfrak{a}}(\mathfrak{s}, \mathfrak{a}) = 1$ and $r_h^{\mathfrak{a}}(\mathfrak{s}, \mathfrak{b}) = 0$ for $h \in \mathcal{H}$, and sets $r_h^{\mathfrak{a}}(\mathfrak{a}, \cdot) = 1$ and $r_h^{\mathfrak{a}}(\mathfrak{b}, \cdot) = 0$ for $h \notin \mathcal{H}$. Meanwhile, $r^{\mathfrak{b}}$ sets $r_h^{\mathfrak{b}}(\mathfrak{s}, \mathfrak{b}) = 1$ and $r_h^{\mathfrak{b}}(\mathfrak{s}, \mathfrak{a}) = 0$ for $h \in \mathcal{H}$ and sets $r_h^{\mathfrak{b}}(\mathfrak{a}, \cdot) = 0$ and $r_h^{\mathfrak{b}}(\mathfrak{b}, \cdot) = 1$ for $h \notin \mathcal{H}$.

Let a *problem instance* $\mathcal{I} = (M^\star, r, \pi^\star)$ refer to a tuple consisting of the reward-free MDP $M^\star$, a reward function $r = \{r_h\}_{h=1}^H$, and an expert policy $\pi^\star$. We consider four problem instances altogether: $(M^\star, r^{\mathfrak{a}}, \pi^{\mathfrak{a}})$, $(M^\star, r^{\mathfrak{b}}, \pi^{\mathfrak{a}})$, $(M^\star, r^{\mathfrak{a}}, \pi^{\mathfrak{b}})$, and $(M^\star, r^{\mathfrak{b}}, \pi^{\mathfrak{b}})$.

Let $\mathbb{P}^{\mathfrak{a}}$ denote the law of $o^1, \ldots, o^n$ when $\mathfrak{a}$ when we execute the algorithm on the underlying instance, and likewise for $\mathfrak{b}$ (recall that the law does not depend on the choice of reward function, since this is not observed); let $\mathbb{E}^{\mathfrak{a}}[\cdot]$ and $\mathbb{E}^{\mathfrak{b}}[\cdot]$ denote the corresponding expectations. In addition, for any policy $\pi$, let $\mathbb{P}^{\pi^{\mathfrak{a}}|\pi}$ denote the law of $o = (x_1, a_1, a_1^\star), \ldots, (x_H, a_H, a_H^\star)$ when we execute $\pi$ in the online imitation learning framework and the expert policy is $\pi^\star = \pi^{\mathfrak{a}}$, and define $\mathbb{P}^{\pi^{\mathfrak{b}}|\pi}$ analogously.

We begin by lower bounding the regret. Consider a fixed policy $\widehat{\pi} = \{\widehat{\pi}_h : \mathcal{X} \to \Delta(\mathcal{X})\}$, and let $\overline{\pi}(a) := \frac{1}{|\mathcal{H}|} \sum_{h \in \mathcal{H}} \widehat{\pi}_h(a \mid \mathfrak{s})$. Observe that for instance $(M^\star, r^{\mathfrak{a}}, \pi^{\mathfrak{a}})$, we have

$$J_{r^{\mathfrak{a}}}(\pi^{\mathfrak{a}}) - J_{r^{\mathfrak{a}}}(\widehat{\pi}) = \left(\frac{1}{2} + \Delta\right)H - K \sum_{h \in \mathcal{H}} \widehat{\pi}_h(\mathfrak{a} \mid \mathfrak{s}) = \left(\frac{1}{2} + \Delta\right)H - H\overline{\pi}(a)$$

---

[21]Rajaraman et al. [67] show that for the tabular setting, it is possible to achieve a $1/n$-type rate *in-expectation* for stochastic policies. Their result critically exploits the assumption that $|\mathcal{X}|$ and $|\mathcal{A}|$ are small and finite to argue that it is possible to build an unbiased estimator for $\pi^\star$. Theorem G.1 shows that such a result cannot hold with even *constant probability* for the same setting. We believe the fact that a $1/n$-type rate is even possible in expectation to be an artifact of the tabular setting, and unlikely to hold for general policy classes.

and for instance $(M^\star, r^\flat, \pi^\mathfrak{a})$,

$$J_{r^\flat}(\pi^\mathfrak{a}) - J_{r^\flat}(\widehat{\pi}) = \left(\frac{1}{2} - \Delta\right)H - K\sum_{h\in\mathcal{H}}\widehat{\pi}_h(\flat \mid \mathfrak{s}) = H\overline{\pi}(\mathfrak{a}) - \left(\frac{1}{2} + \Delta\right)H.$$

Likewise, for instance $(M^\star, r^\flat, \pi^\flat)$, we have

$$J_{r^\flat}(\pi^\flat) - J_{r^\flat}(\widehat{\pi}) = \left(\frac{1}{2} + \Delta\right)H - K\sum_{h\in\mathcal{H}}\widehat{\pi}_h(\flat \mid \mathfrak{s}) = \overline{\pi}(\mathfrak{a})H - \left(\frac{1}{2} - \Delta\right)H$$

and for instance $(M^\star, r^\mathfrak{a}, \pi^\flat)$,

$$J_{r^\mathfrak{a}}(\pi^\flat) - J_{r^\mathfrak{a}}(\widehat{\pi}) = \left(\frac{1}{2} - \Delta\right)H - K\sum_{h\in\mathcal{H}}\widehat{\pi}_h(\mathfrak{a} \mid \mathfrak{s}) = \left(\frac{1}{2} - \Delta\right)H - \overline{\pi}(\mathfrak{a})H.$$

We conclude that for any $\varepsilon > 0$, since the law of the dataset is independent of the choice of the reward function,

$$\max\{\mathbb{P}^\mathfrak{a}[J_{r^\mathfrak{a}}(\pi^\mathfrak{a}) - J_{r^\mathfrak{a}}(\widehat{\pi}) \geq \varepsilon H], \mathbb{P}^\mathfrak{a}[J_{r^\flat}(\pi^\mathfrak{a}) - J_{r^\flat}(\widehat{\pi}) \geq \varepsilon H], \mathbb{P}^\flat[J_{r^\flat}(\pi^\flat) - J_{r^\flat}(\widehat{\pi}) \geq \varepsilon H], \mathbb{P}^\flat[J_{r^\mathfrak{a}}(\pi^\flat) - J_{r^\mathfrak{a}}(\widehat{\pi}) \geq \varepsilon H]\}$$

$$\geq \max\left\{\begin{array}{l}\mathbb{P}^\mathfrak{a}\left[\left(\frac{1}{2} + \Delta\right)H - \overline{\pi}(\mathfrak{a})H \geq \varepsilon H\right], \mathbb{P}^\mathfrak{a}\left[\overline{\pi}(\mathfrak{a})H - \left(\frac{1}{2} + \Delta\right)H \geq \varepsilon H\right], \\ \mathbb{P}^\flat\left[\overline{\pi}(\mathfrak{a})H - \left(\frac{1}{2} - \Delta\right)H \geq \varepsilon H\right], \mathbb{P}^\flat\left[\left(\frac{1}{2} - \Delta\right)H - \overline{\pi}(\mathfrak{a})H \geq \varepsilon H\right]\end{array}\right\}$$

$$\geq \frac{1}{2}\max\left\{\mathbb{P}^\mathfrak{a}\left[\left|\left(\frac{1}{2} + \Delta\right) - \overline{\pi}(\mathfrak{a})\right|H \geq \varepsilon H\right], \mathbb{P}^\flat\left[\left|\overline{\pi}(\mathfrak{a}) - \left(\frac{1}{2} - \Delta\right)\right|H \geq \varepsilon H\right]\right\}$$

$$= \frac{1}{2}\max\left\{\mathbb{P}^\mathfrak{a}\left[\left|\left(\frac{1}{2} + \Delta\right) - \overline{\pi}(\mathfrak{a})\right| \geq \varepsilon\right], \mathbb{P}^\flat\left[\left|\overline{\pi}(\mathfrak{a}) - \left(\frac{1}{2} - \Delta\right)\right| \geq \varepsilon\right]\right\}$$

$$\geq \frac{1}{4}\left(\mathbb{P}^\mathfrak{a}\left[\left|\left(\frac{1}{2} + \Delta\right) - \overline{\pi}(\mathfrak{a})\right| \geq \varepsilon\right] + \mathbb{P}^\flat\left[\left|\overline{\pi}(\mathfrak{a}) - \left(\frac{1}{2} - \Delta\right)\right| \geq \varepsilon\right]\right)$$

$$\geq \frac{1}{4}\left(1 - \mathbb{P}^\mathfrak{a}\left[\left|\left(\frac{1}{2} + \Delta\right) - \overline{\pi}(\mathfrak{a})\right| \leq \varepsilon\right] + \mathbb{P}^\flat\left[\left|\overline{\pi}(\mathfrak{a}) - \left(\frac{1}{2} - \Delta\right)\right| \geq \varepsilon\right]\right)$$

$$\geq \frac{1}{4}\left(1 - \mathbb{P}^\mathfrak{a}\left[\left|\left(\frac{1}{2} - \Delta\right) - \overline{\pi}(\mathfrak{a})\right| \geq \varepsilon\right] + \mathbb{P}^\flat\left[\left|\overline{\pi}(\mathfrak{a}) - \left(\frac{1}{2} - \Delta\right)\right| \geq \varepsilon\right]\right)$$

$$\geq \frac{1}{4}(1 - D_{\mathsf{TV}}(\mathbb{P}^\mathfrak{a}, \mathbb{P}^\flat)),$$

where the second inequality uses the union bound (i.e. $\mathbb{P}[|x| \geq \varepsilon] = \mathbb{P}[x \geq \varepsilon \cup -x \geq \varepsilon] \leq \mathbb{P}[x \geq \varepsilon] + \mathbb{P}[-x \geq \varepsilon]$), and the second-to-last inequality holds as long as $\varepsilon < \Delta$. In particular, this implies that

$$\max\left\{\begin{array}{l}\mathbb{P}^\mathfrak{a}\left[J_{r^\mathfrak{a}}(\pi^\mathfrak{a}) - J_{r^\mathfrak{a}}(\widehat{\pi}) \geq \frac{\Delta H}{2}\right], \mathbb{P}^\mathfrak{a}\left[J_{r^\flat}(\pi^\mathfrak{a}) - J_{r^\flat}(\widehat{\pi}) \geq \frac{\Delta H}{2}\right], \\ \mathbb{P}^\flat\left[J_{r^\flat}(\pi^\flat) - J_{r^\flat}(\widehat{\pi}) \geq \frac{\Delta H}{2}\right], \mathbb{P}^\flat\left[J_{r^\mathfrak{a}}(\pi^\flat) - J_{r^\mathfrak{a}}(\widehat{\pi}) \geq \frac{\Delta H}{2}\right]\end{array}\right\} \geq \frac{1}{4}(1 - D_{\mathsf{TV}}(\mathbb{P}^\mathfrak{a}, \mathbb{P}^\flat)).$$

Next, using Lemma D.2 of Foster et al. [36], we can bound

$$D_{\mathsf{TV}}^2(\mathbb{P}^\mathfrak{a}, \mathbb{P}^\flat) \leq D_{\mathsf{H}}^2(\mathbb{P}^\mathfrak{a}, \mathbb{P}^\flat) \leq 7\,\mathbb{E}^\mathfrak{a}\left[\sum_{i=1}^n D_{\mathsf{H}}^2\left(\mathbb{P}^{\pi^\mathfrak{a}|\pi^i}, \mathbb{P}^{\pi^\flat|\pi^i}\right)\right].$$

Observe that for a given episode $i$, regardless of how the policy $\pi^i$ is selected:

- The feedback for steps $h \notin \mathcal{H}$ is identical under $\mathbb{P}^\mathfrak{a}$ and $\mathbb{P}^\flat$.
- The feedback at step $h \in \mathcal{H}$ differs only in the distribution of $a_h^\star \sim \pi^\mathfrak{a}(\mathfrak{s})$ versus $a_h^\star \sim \pi^\flat(\mathfrak{s})$. This is equivalently to $\mathrm{Ber}(1/2 + \Delta)$ feedback versus $\mathrm{Ber}(1/2 - \Delta)$ feedback.

As a result, using Lemma D.2 of Foster et al. [36] once more, we have

$$D_{\mathsf{H}}^2\left(\mathbb{P}^{\pi^{\mathfrak{a}}|\pi^i}, \mathbb{P}^{\pi^{\mathfrak{b}}|\pi^i}\right) \le 7 \sum_{h \in \mathcal{H}} D_{\mathsf{H}}^2(\mathrm{Ber}(1/2 + \Delta), \mathrm{Ber}(1/2 - \Delta))$$

Since $\Delta \in (0, 1/2)$, we have $D_{\mathsf{H}}^2(\mathrm{Ber}(1/2 + \Delta), \mathrm{Ber}(1/2 - \Delta)) \le O(\Delta^2)$ (e.g., Foster et al. [34, Lemma A.7]). We conclude that

$$D_{\mathsf{TV}}^2(\mathbb{P}^{\mathfrak{a}}, \mathbb{P}^{\mathfrak{b}}) \le O(n \cdot |\mathcal{H}| \cdot \Delta^2) = O\left(n \cdot \frac{H}{K} \cdot \Delta^2\right)$$

We set $\Delta^2 = c \cdot \frac{K}{Hn}$ for $c > 0$ sufficiently small so that $D_{\mathsf{TV}}^2(\mathbb{P}^{\mathfrak{a}}, \mathbb{P}^{\mathfrak{b}}) \le 1/2$, and conclude that on at least one of the four problem instances, the algorithm must have

$$J(\pi^\star) - J(\widehat{\pi}) \ge \Omega(\Delta H) = \Omega\left(\sqrt{\frac{HK}{n}}\right)$$

with probability at least $1/8$.

Finally, we compute the variance and choose the parameter $K$. Observe that for all of the choices of expert policy and reward function described above, we have $Q_h^{\pi^\star}(x_h, \pi^\star(x_h)) - Q_h^{\pi^\star}(x_h, a) = 0$ for $h \notin \mathcal{H}$, while

$$\left|Q_h^{\pi^\star}(x_h, \pi^\star(x_h)) - Q_h^{\pi^\star}(x_h, a)\right| \le K$$

for $h \in \mathcal{H}$, so we can take $\widetilde{\mu} \le K$. Consequently, we have

$$\sigma_{\pi^\star}^2 = \sum_{h=1}^{H} \mathbb{E}^{\pi^\star}\left[(Q_h^{\pi^\star}(x_h, \pi^\star(x_h)) - Q_h^{\pi^\star}(x_h, a_h))^2\right] \le \sum_{h \in \mathcal{H}} \mathbb{E}^{\pi^\star}\left[(Q_h^{\pi^\star}(\mathfrak{s}, \pi^\star(\mathfrak{s})) - Q_h^{\pi^\star}(\mathfrak{s}, a_h))^2\right]$$

$$\le \frac{H}{K} \cdot K^2 = HK.$$

We conclude by setting $K = \sigma^2/H$, which is admissible for $\sigma^2 \in [H, H^2]$ (up to a loss in absolute constants, we can assume that $\sigma^2/H$ is an integer without loss of generality). $\qquad\square$

### G.3 Additional Proofs

**Proof of Proposition 3.1.** We have

$$\sigma_{\pi^\star}^2 = \sum_{h=1}^{H} \mathbb{E}^{\pi^\star}\left[(Q_h^{\pi^\star}(x_h, a_h) - V_h^{\pi^\star}(x_h))^2\right].$$

Note that $Q_h^{\pi^\star}(x_h, a_h) = \mathbb{E}\left[r_h + V_h^{\pi^\star}(x_{h+1}) \mid x_h, a_h\right]$. Hence, by Jensen's inequality we can bound

$$\mathbb{E}^{\pi^\star}\left[(Q_h^{\pi^\star}(x_h, a_h) - V_h^{\pi^\star}(x_h))^2\right] \le \mathbb{E}^{\pi^\star}\left[\mathbb{E}\left[(r_h + V_{h+1}^{\pi^\star}(x_{h+1}) - V_h^{\pi^\star}(x_h))^2 \mid x_h, a_h\right]\right]$$

$$= \mathbb{E}^{\pi^\star}\left[\mathbb{E}^{\pi^\star}\left[(r_h + V_{h+1}^{\pi^\star}(x_{h+1}) - V_h^{\pi^\star}(x_h))^2 \mid x_h\right]\right]$$

$$= \mathbb{E}^{\pi^\star}\left[\mathrm{Var}^{\pi^\star}\left[r_h + V_{h+1}^{\pi^\star}(x_{h+1}) \mid x_h\right]\right],$$

so that

$$\sigma_{\pi^\star}^2 \le \mathbb{E}^{\pi^\star}\left[\sum_{h=1}^{H} \mathrm{Var}^{\pi^\star}\left[r_h + V_{h+1}^{\pi^\star}(x_{h+1}) \mid x_h\right]\right]$$

$$\le \mathbb{E}^{\pi^\star}\left[\sum_{h=0}^{H} \mathrm{Var}^{\pi^\star}\left[r_h + V_{h+1}^{\pi^\star}(x_{h+1}) \mid x_h\right]\right] = \mathrm{Var}^{\pi^\star}\left[\sum_{h=1}^{H} r_h\right] \le R^2,$$

where the second to last inequality follows from Lemma D.5.

$\square$

# Part II

# Additional Results

## H    Additional Lower Bounds

This section contains additional lower bounds that complement the results in Sections 2 and 3:

- Appendix H.1 shows that the conclusion of Theorem H.1 continues to hold even for online imitation learning in an *active* sample complexity framework.

- Appendix H.2 presents an instance-dependent lower bound for stochastic experts, complementing the minimax lower bound in Theorem G.1.

- Appendix H.3 investigates the extent to which Theorems 2.1 and 3.1 are tight on a per-policy basis.

### H.1    Lower Bounds for Online Imitation Learning in Active Interaction Model

For the online imitation learning setting introduced in Section 1.1, we measure sample complexity in terms of the total number of episodes of online interaction, and expert feedback is available in every episode. In this section, we consider a more permissive sample complexity framework inspired by active learning [40, 75]. Here, as in Section 1.1, the learner interacts with the underlying MDP $M^\star$ through multiple episodes. At each episode $i \in [n]$ the learner executes a policy $\pi^i = \{\pi_h^i : \mathcal{X} \to \Delta(\mathcal{A})\}_{h=1}^H$, and at any step $h$ in the episode, they can decide whether to query the expert for an action $a_h^\star \sim \pi_h^\star(x_h)$ at the current state $x_h$. We set $M^i = 1$ if the learner queries the expert at any point during episode $i$ and set $M^i = 0$ otherwise, and define the *active sample complexity* $M := \sum_{i=1}^n M^i$ as the total number of queries.

It is clear that the active sample complexity satisfies $m \leq n$, and in some cases we might hope for it to be much smaller than the total number of episodes, at least for a well-designed algorithm. While this can indeed be the case for MDPs that satisfies (fairly strong) distributional assumptions [75], we will show that the lower bound in Theorem 2.2 continues to hold in this framework (up to a logarithmic factor), meaning that online interaction in the active sample complexity framework cannot improve over LogLossBC in general.

**Theorem H.1** (Lower bound for deterministic experts in active sample complexity framework)**.** *For any $m \in \mathbb{N}$ and $H \in \mathbb{N}$, there exists a reward-free MDP $M^\star$ with $|\mathcal{X}| = |\mathcal{A}| = m + 1$, a class of reward functions $\mathcal{R}$ with $|\mathcal{R}| = m + 1$, and a class of deterministic policies $\Pi$ with $\log|\Pi| = \log(m)$ with the following property. For any online imitation learning algorithm in the active sample complexity framework that has sample complexity $\mathbb{E}[M] \leq c \cdot m$ for an absolute constant $c > 0$, there exists a deterministic reward function $r = \{r_h\}_{h=1}^H$ with $r_h \in [0, 1]$ and (optimal) expert policy $\pi^\star \in \Pi$ with $\mu = 1$ such that the expected suboptimality is lower bounded as*

$$\mathbb{E}[J(\pi^\star) - J(\widehat{\pi})] \geq c \cdot \frac{H}{m}$$

*for an absolute constant $c > 0$. In addition, the dynamics, rewards, and expert policies are all stationary.*

Since this example has $\log|\Pi| = \log(M)$, it follows that the sample complexity bound for LogLossBC in Theorem 2.1 (which uses $M = n$) can be improved by no more than a $\log(n)$ factor through online interaction in the active framework.

**Proof of Theorem H.1.**    Let $m \in \mathbb{N}$ and $H \in \mathbb{N}$ be fixed. We first specify the dynamics for the reward-free MDP $M^\star$. Set $\mathcal{X} = \{\maltese_1, \ldots, \maltese_m\}$ and $\mathcal{A} = \{\mathfrak{a}, \mathfrak{b}\}$. The initial state distribution is $P_0 = \mathrm{unif}(\maltese_1, \ldots, \maltese_m)$. The transition dynamics are $P_h(x' \mid x, a) = \mathbb{I}\{x' = x\}$ for all $h$; that is, $\maltese_1, \ldots, \maltese_m$ are all self-looping terminal states.

Let a *problem instance* $\mathcal{I} = (M^\star, r, \pi^\star)$ refer to a tuple consisting of the reward-free MDP $M^\star$, a reward function $r = \{r_h\}_{h=1}^{H}$, and an expert policy $\pi^\star$. We consider $m+1$ problem instances $\mathcal{I}^0, \ldots, \mathcal{I}^m$ parameterized by a collection of policies $\Pi = \{\pi^0, \ldots, \pi^m\}$ and reward functions $\mathcal{R} = \{r^0, \ldots, r^m\}$.

- For problem instance $\mathcal{I}^0 = (M^\star, r^0, \pi^0)$, the expert policy is $\pi^0$, which sets $\pi_h^0(x) = \mathfrak{a}$ for all $x \in \mathcal{X}$ and $h \in [H]$. The reward function $r^0$ sets $r_h(x, a) = \mathbb{I}\{a = \mathfrak{a}\}$ for all $x \in \mathcal{X}$ and $h \in [H]$.

- For each problem instance $\mathcal{I}^j = (M^\star, r^j, \pi^j)$, the expert policy is $\pi^j$, which for all $h \in [H]$ sets $\pi_h^j(x) = \mathfrak{a}$ for $x \neq \mathfrak{x}_j$ and sets $\pi^h(\mathfrak{x}_j) = \mathfrak{b}$. The reward function $r^j$ sets $r_h(x, a) = \mathbb{I}\{a = \mathfrak{a}, x \neq \mathfrak{x}_j\} + \mathbb{I}\{a = \mathfrak{b}, x = \mathfrak{x}_j\}$ for all $h \in [H]$.

Let $J^j$ denote the expected reward under instance $j$. Note that all instances satisfy $\mu = 1$, and that $\pi^j$ is an optimal policy for each instance $j$.

Going forward, we fix the online imitation learning algorithm under consideration and let $\mathbb{P}^j$ denote the law of $o^1, \ldots, o^n$ when $\mathfrak{a}$ when we execute the algorithm on instance $\mathcal{I}^j$; let $\mathbb{E}^j[\cdot]$ denote the corresponding expectation. In addition, for any policy $\pi$, let $\mathbb{P}^{\pi^j|\pi}$ denote the law of $o = (x_1, a_1, a_1^\star), \ldots, (x_H, a_H, a_H^\star)$ when we execute $\pi$ in the online imitation learning framework when the underlying instance is $\mathcal{I}^j$, with the convention that $a_h^\star = \perp$ if the learner does not query the expert in episode $j$.

Our aim is to lower bound

$$\max_{j \in \{0, \ldots, m\}} \mathbb{E}^j[J^j(\pi^j) - J^j(\widehat{\pi})]$$

To this end, define $\rho_j(\pi, \pi') = \sum_{h=1}^{H} \mathbb{E}_{a_h \sim \pi_h(\mathfrak{x}_j), a_h' \sim \pi_h'(\mathfrak{x}_j)} \mathbb{I}\{a_h \neq a_h'\}$ and $\rho(\pi, \pi') = \frac{1}{m} \rho_j(\pi, \pi')$, and observe that

$$\mathbb{E}^0[J^0(\pi^0) - J^0(\widehat{\pi})] = \mathbb{E}^0\left[\frac{1}{m} \sum_{j=1}^{m} \sum_{h=1}^{H} \mathbb{E}_{a_h \sim \widehat{\pi}_h(\mathfrak{x}_j)}[\mathbb{I}\{a_h \neq \pi_h^0(\mathfrak{x}_j)\}]\right]$$

$$= \mathbb{E}^0[\rho(\widehat{\pi}, \pi^0)] \geq \frac{H}{2m} \cdot \mathbb{P}^0\left[\rho(\widehat{\pi}, \pi^0) \geq \frac{H}{2m}\right].$$

Next, note that for any $i \in [m]$, if $\rho(\widehat{\pi}, \pi^0) < \frac{H}{2m}$, then $\rho_j(\widehat{\pi}, \pi^0) < \frac{H}{2}$, which means that $\rho_j(\widehat{\pi}, \pi^j) \geq \frac{H}{2}$. It follows that

$$\mathbb{E}^j[J^j(\pi^j) - J^j(\widehat{\pi})] = \mathbb{E}^j\left[\frac{1}{m} \rho_j(\widehat{\pi}, \pi^j)\right] \geq \frac{H}{2m} \mathbb{P}^j\left[\rho(\widehat{\pi}, \pi^0) < \frac{H}{2m}\right],$$

and if we define $\bar{\mathbb{P}} = \mathbb{E}_{j \sim \text{unif}([m])} \mathbb{P}^j$, then

$$\mathbb{E}_{j \sim \text{unif}([m])} \mathbb{E}^j[J^j(\pi^j) - J^j(\widehat{\pi})] \geq \frac{H}{2m} \bar{\mathbb{P}}\left[\rho(\widehat{\pi}, \pi^0) < \frac{H}{2m}\right].$$

Combining these observations, we find that

$$\max_{i \in \{0, \ldots, m\}} \mathbb{E}^j[J^j(\pi^j) - J^j(\widehat{\pi})] \geq \frac{H}{4m}\left(\mathbb{P}^0\left[\rho(\widehat{\pi}, \pi^0) \geq \frac{H}{2m}\right] + \bar{\mathbb{P}}\left[\rho(\widehat{\pi}, \pi^0) < \frac{H}{2m}\right]\right)$$

$$\geq \frac{H}{4m}(1 - D_{\mathsf{TV}}(\mathbb{P}^0, \bar{\mathbb{P}})).$$

It remains to bound the total variation distance. Next, using Lemma D.2 of Foster et al. [36], we can bound

$$D_{\mathsf{TV}}^2(\mathbb{P}^0, \bar{\mathbb{P}}) \leq D_{\mathsf{H}}^2(\mathbb{P}^0, \bar{\mathbb{P}}) \leq \mathbb{E}_{j \sim \text{unif}[m]}[D_{\mathsf{H}}^2(\mathbb{P}^0, \mathbb{P}^j)] \leq 7 \mathbb{E}_{j \sim \text{unif}[m]} \mathbb{E}^0\left[\sum_{t=1}^{n} D_{\mathsf{H}}^2\left(\mathbb{P}^{\pi^0|\pi^t}, \mathbb{P}^{\pi^j|\pi^t}\right)\right].$$

Since the feedback the learner receives for a given episode $t$ is identical under instances $\mathcal{I}^0$ and $\mathcal{I}^j$ is identical unless i) $x_1 = \mathfrak{x}_j$, and ii) the learner decides to query the expert for feedback (i.e., $M^t = 1$), we can bound

$$D_{\mathsf{H}}^2\left(\mathbb{P}^{\pi^0|\pi^t}, \mathbb{P}^{\pi^j|\pi^0}\right) \leq 2 \mathbb{P}^{\pi^0|\pi^t}[x_1^t = \mathfrak{x}_j, M^t = 1]$$

and hence

$$\mathbb{E}_{j\sim\mathrm{unif}[m]}\,\mathbb{E}^{\mathrm{o}}\left[\sum_{t=1}^{n}D_{\mathsf{H}}^{2}\!\left(\mathbb{P}^{\pi^{0}|\pi^{t}},\mathbb{P}^{\pi^{j}|\pi^{t}}\right)\right]\leq 2\,\mathbb{E}_{j\sim\mathrm{unif}[m]}\,\mathbb{E}^{\mathrm{o}}\left[\sum_{t=1}^{n}\mathbb{P}^{\pi^{0}|\pi^{t}}[x_{1}^{t}=\mathbf{x}_{j},M^{t}=1]\right]$$

$$=\frac{2}{m}\,\mathbb{E}^{\mathrm{o}}\left[\sum_{t=1}^{n}\sum_{j=1}^{m}\mathbb{P}^{\pi^{0}|\pi^{t}}[x_{1}^{t}=\mathbf{x}_{j},M^{t}=1]\right]$$

$$=\frac{2}{m}\,\mathbb{E}^{\mathrm{o}}\left[\sum_{t=1}^{n}\mathbb{P}^{\pi^{0}|\pi^{t}}[M^{t}=1]\right]$$

$$=\frac{2}{m}\,\mathbb{E}^{\mathrm{o}}[M].$$

It follows that if $\mathbb{E}^{\mathrm{o}}[M]\leq m/56$, then $D_{\mathsf{TV}}\big(\mathbb{P}^{\mathrm{o}},\bar{\mathbb{P}}\big)\leq 1/2$, so that the algorithm must have

$$\max_{i\in\{0,\dots,m\}}\mathbb{E}^{j}[J^{j}(\pi^{j})-J^{j}(\widehat{\pi})]\geq\frac{H}{8m}.$$

$\square$

## H.2 An Instance-Dependent Lower Bound for Stochastic Experts

In this section, we further investigate the optimality of LogLossBC for stochastic experts (Theorem 3.1). Recall that when $\log|\Pi|=O(1)$ the leading-order term in Theorem 3.1 scales as roughly $\sqrt{\sigma_{\pi^{\star}}^{2}/n}$, where the salient quantity is the *variance*

$$\sigma_{\pi^{\star}}^{2}:=\sum_{h=1}^{H}\mathbb{E}^{\pi^{\star}}\left[\left(Q_{h}^{\pi^{\star}}(x_{h},\pi^{\star}(x_{h}))-Q_{h}^{\pi^{\star}}(x_{h},a_{h})\right)^{2}\right]$$

for the expert policy $\pi^{\star}$. Theorem G.1 shows that this is optimal qualitatively, in the sense that for any value $\sigma^{2}$, there exists a class of MDPs where the $\sigma_{\pi^{\star}}^{2}\leq\sigma^{2}$, and where the minimax rate is at least $\sqrt{\sigma^{2}/n}$.

In what follows, we will prove that for the special case of *autoregressive* MDPs (that is, the special case of the imitation learning problem in which the state takes the form $x_{h}=a_{1:h-1}$; cf. Appendix B.3), Theorem G.1 is optimal on a *per-policy* basis. Concretely, we prove a *local minimax* lower bound [28] which states that for any policy $\pi^{\star}$ and any reward function $r^{\star}$, there exists a difficult "alternative" policy $\widetilde{\pi}$, such that in worst case over rewards $r\in\{-r^{\star},+r^{\star}\}$ and expert policies $\pi\in\{\pi^{\star},\widetilde{\pi}\}$, any algorithm must have regret at least $\sqrt{\sigma^{2}/n}$.

**Theorem H.2.** *Consider the offline imitation learning setting, and let $M^{\star}$ be an autoregressive MDP. Let a reward function $r^{\star}$ with $\sum_{h=1}^{H}r_{h}^{\star}\in[0,R]$ almost surely be fixed, and let an expert policy $\pi^{\star}$ be given. For any $n\in\mathbb{N}$, there exists an alternative policy $\widetilde{\pi}$ such that*

$$\min_{\mathrm{Alg}}\max_{\pi\in\{\pi^{\star},\widetilde{\pi}\}}\max_{r\in\{r^{\star},-r^{\star}\}}\mathbb{P}\left[J(\pi)-J(\widehat{\pi})\geq c\cdot\sqrt{\frac{\sigma_{\pi^{\star}}^{2}}{n}}\right]\geq\frac{1}{4}$$

*for all $n\geq c'\cdot\frac{R^{2}}{\sigma_{\pi^{\star}}^{2}}$, where $c,c'>0$ are absolute constants.*

Theorem H.2 suggests that the leading term in Theorem 3.1 cannot be improved substantially without additional assumptions, on a (nearly) per-instance basis. The restriction to $n\geq c'\cdot\frac{R^{2}}{\sigma_{\pi^{\star}}^{2}}$ in Theorem H.2 is somewhat natural, as this corresponds to the regime in which the $\sqrt{\sigma_{\pi^{\star}}^{2}/n}$ term in Theorem 3.1 dominates the lower-order term.

**Proof of Theorem H.2.** We begin by observing that for any $\Delta>0$,

$$\min_{\mathrm{Alg}}\max_{\pi\in\{\pi^{\star},\widetilde{\pi}\}}\max_{r\in\{r^{\star},-r^{\star}\}}\mathbb{P}[J_{r}(\pi)-J_{r}(\widehat{\pi})\geq\Delta]\geq\min_{\mathrm{Alg}}\max_{\pi\in\{\pi^{\star},\widetilde{\pi}\}}\mathbb{P}[|J_{r^{\star}}(\pi)-J_{r^{\star}}(\widehat{\pi})|\geq\Delta].$$

with the convention that $J_r(\pi)$ denotes the expected reward under $r$; we abbreviate $J(\pi) \equiv J_{r^\star}(\pi)$ going forward. Let $\mathbb{P}_n^\pi$ denote the law of the offline imitation learning dataset under $\pi$. If we set $\Delta = |J(\pi^\star) - J(\widetilde{\pi})|/2$, then by the standard Le Cam two-point argument, we have that

$$
\max\Big\{\mathbb{P}_n^{\pi^\star}[|J(\pi^\star) - J(\widehat{\pi})| \geq \Delta], \mathbb{P}_n^{\widetilde{\pi}}[|J(\widetilde{\pi}) - J(\widehat{\pi})| \geq \Delta]\Big\}
$$

$$
\geq \frac{1}{2}\Big(1 - \mathbb{P}_n^{\pi^\star}[|J(\pi^\star) - J(\widehat{\pi})| < \Delta] + \mathbb{P}_n^{\widetilde{\pi}}[|J(\widetilde{\pi}) - J(\widehat{\pi})| \geq \Delta]\Big)
$$

$$
\geq \frac{1}{2}\Big(1 - \mathbb{P}_n^{\pi^\star}[|J(\widetilde{\pi}) - J(\widehat{\pi})| \geq \Delta] + \mathbb{P}_n^{\widetilde{\pi}}[|J(\widetilde{\pi}) - J(\widehat{\pi})| \geq \Delta]\Big)
$$

$$
\geq \frac{1}{2}\Big(1 - D_{\mathsf{TV}}\Big(\mathbb{P}_n^{\pi^\star}, \mathbb{P}_n^{\widetilde{\pi}}\Big)\Big) \geq \frac{1}{2}\Big(1 - \sqrt{n \cdot D_{\mathsf{H}}^2(\mathbb{P}^{\pi^\star}, \mathbb{P}^{\widetilde{\pi}})}\Big),
$$

where the final inequality uses the standard tensorization property for Hellinger distance (e.g., Wainwright [91]).

We will proceed by showing that

$$
\omega_{\pi^\star}(\varepsilon) := \sup_\pi\Big\{|J(\pi) - J(\pi^\star)| \mid D_{\mathsf{H}}^2\Big(\mathbb{P}^{\pi^\star}, \mathbb{P}^\pi\Big) \leq \varepsilon^2\Big\} \geq \Omega(1) \cdot \sqrt{\sigma_{\pi^\star}^2 \cdot \varepsilon^2}, \tag{19}
$$

for any $\varepsilon > 0$ sufficiently small, from which the result will follow by setting $\varepsilon^2 \propto 1/n$ and

$$
\widetilde{\pi} = \arg\max_\pi\Big\{|J(\pi) - J(\pi^\star)| \mid D_{\mathsf{H}}^2\Big(\mathbb{P}^{\pi^\star}, \mathbb{P}^\pi\Big) \leq \varepsilon^2\Big\} \geq \Omega(1) \cdot \sqrt{\sigma_{\pi^\star}^2 \cdot \varepsilon^2}.
$$

To prove this, we will appeal to the following technical lemma.

**Lemma H.1.** *For any distribution $\mathbb{Q}$ and function $h$ with $|h| \leq R$ almost surely, it holds that for all $0 \leq \varepsilon^2 \leq \frac{\mathrm{Var}_{\mathbb{Q}}[h]}{4R^2}$, there exists a distribution $\mathbb{P}$ such that*

*1. $\mathbb{E}_{\mathbb{P}}[h] - \mathbb{E}_{\mathbb{Q}}[h] \geq 2^{-3}\sqrt{\mathrm{Var}_{\mathbb{Q}}[h] \cdot \varepsilon^2}$*

*2. $D_{\mathsf{KL}}(\mathbb{Q} \,\|\, \mathbb{P}) \leq \varepsilon^2$.*

Since stochastic policies $\pi$ in the autoregressive MDP $M^\star$ are equivalent to arbitrary joint laws over the sequence $a_{1:H}$ (via Bayes' rule) and $J(\pi) = \mathbb{E}^\pi\Big[\sum_{h=1}^H r_h^\star\Big]$, Lemma H.1 implies that for any $\varepsilon^2 \leq \mathrm{Var}^{\pi^\star}\Big[\sum_{h=1}^H r_h^\star\Big]/4R^2$, there exists a policy $\widetilde{\pi}$ such that (i) $D_{\mathsf{H}}^2\big(\mathbb{P}^{\pi^\star}, \mathbb{P}^{\widetilde{\pi}}\big) \leq D_{\mathsf{KL}}\big(\mathbb{P}^{\pi^\star} \,\|\, \mathbb{P}^{\widetilde{\pi}}\big) \leq \varepsilon^2$, and (ii)

$$
J(\widetilde{\pi}) - J(\pi^\star) \geq 2^{-3}\sqrt{\mathrm{Var}^{\pi^\star}\left[\sum_{h=1}^H r_h^\star\right] \cdot \varepsilon^2}.
$$

This establishes Eq. (19). The result now follows by setting $\varepsilon^2 = \frac{c}{n}$ for an absolute constant $c > 0$ so that $\sqrt{n \cdot D_{\mathsf{H}}^2(\mathbb{P}^{\pi^\star}, \mathbb{P}^{\widetilde{\pi}})} \leq 1/2$, which is admissible whenever $n \geq c' \cdot \frac{R^2}{\sigma_{\pi^\star}^2}$. Finally, we observe that for any deterministic MDP, by Lemma D.5,

$$
\mathrm{Var}^{\pi^\star}\left[\sum_{h=1}^H r_h\right] = \mathbb{E}^{\pi^\star}\left[\sum_{h=1}^H \mathrm{Var}^{\pi^\star}\Big[r_h + V_{h+1}^{\pi^\star}(x_{h+1}) \mid x_h\Big]\right] = \mathbb{E}^{\pi^\star}\left[\sum_{h=1}^H (Q_h^{\pi^\star}(x_h, a_h) - V_h^{\pi^\star}(x_h))^2\right] = \sigma_{\pi^\star}^2,
$$

since deterministic MDPs satisfy

$$
Q_h^{\pi^\star}(x_h, a_h) = r_h(x_h, a_h) + V_{h+1}^{\pi^\star}(x_{h+1})
$$

almost surely, and since $\mathbb{E}^{\pi^\star}\big[Q_h^{\pi^\star}(x_h, a_h) \mid x_h\big] = V_h^{\pi^\star}(x_h)$.

$\square$

**Proof of Lemma H.1.** Recall that we assume the domain is countable, so that $\mathbb{Q}$ admits a probability mass function $q$. We will define $\mathbb{P}$ via the probability mass function

$$
p(x) = \frac{q(x)e^{\eta h(x)}}{\sum_{x'} q(x')e^{\eta h(x')}}
$$

for a parameter $\eta > 0$. We begin by observing that

$$D_{\mathsf{KL}}(\mathbb{Q} \,\|\, \mathbb{P}) = \log\big(\mathbb{E}_{\mathbb{Q}}\big[e^{\eta h}\big]\big) - \eta\,\mathbb{E}_{\mathbb{Q}}[h] = \log\Big(\mathbb{E}_{\mathbb{Q}}\Big[e^{\eta(h - \mathbb{E}_{\mathbb{Q}}[h])}\Big]\Big).$$

We now use the following lemma.

**Lemma H.2.** *For any random variable $X$ with $|X| \le R$ almost surely and any $\eta \in (0, (2R)^{-1})$,*

$$\frac{\eta^2}{8}\mathrm{Var}[X] \le \log\Big(\mathbb{E}\Big[e^{\eta(X - \mathbb{E}[X])}\Big]\Big) \le \eta^2\mathrm{Var}[X].$$

Hence, as long as $\eta \le (2R)^{-1}$,

$$D_{\mathsf{KL}}(\mathbb{Q} \,\|\, \mathbb{P}) \le \eta^2\mathrm{Var}_{\mathbb{Q}}[h].$$

We set $\eta = \min\Big\{\sqrt{\frac{\varepsilon^2}{\mathrm{Var}_{\mathbb{Q}}[h]}}, \frac{1}{2R}\Big\}$ so that $D_{\mathsf{KL}}(\mathbb{Q} \,\|\, \mathbb{P}) \le \varepsilon^2$.

Next, we compute that

$$0 \le D_{\mathsf{KL}}(\mathbb{P} \,\|\, \mathbb{Q}) = \eta\,\mathbb{E}_{\mathbb{P}}[h] - \log\big(\mathbb{E}_{\mathbb{Q}}\big[e^{\eta h}\big]\big),$$

so that

$$\mathbb{E}_{\mathbb{P}}[h] - \mathbb{E}_{\mathbb{Q}}[h] \ge \eta^{-1}\log\big(\mathbb{E}_{\mathbb{Q}}\big[e^{\eta h}\big]\big) - \mathbb{E}_{\mathbb{Q}}[h] = \eta^{-1}\log\Big(\mathbb{E}_{\mathbb{Q}}\Big[e^{\eta(h - \mathbb{E}_{\mathbb{Q}}[h])}\Big]\Big).$$

Since $\eta \le (2R)^{-1}$, Lemma H.2 yields

$$\mathbb{E}_{\mathbb{P}}[h] - \mathbb{E}_{\mathbb{Q}}[h] \ge \frac{\eta}{8}\mathrm{Var}_{\mathbb{Q}}[h] = \frac{1}{8}\sqrt{\mathrm{Var}_{\mathbb{Q}}[h] \cdot \varepsilon^2}$$

as long as $\varepsilon^2 \le \frac{\mathrm{Var}_{\mathbb{Q}}[h]}{4R^2}$.

$\square$

**Proof of Lemma H.2.** Note that $e^x \le 1 + x + (e - 2)x^2 \le 1 + x + x^2$ whenever $|x| \le 1$, and similarly $e^x \ge 1 + x + \frac{x^2}{4}$ for $|x| \le 1$. It follows that if $\eta \le (2R)^{-1}$,

$$1 + \frac{\eta^2}{4}\mathrm{Var}(X) \le \mathbb{E}\Big[e^{\eta(X - \mathbb{E}[X])}\Big] \le 1 + \eta^2\mathrm{Var}(X).$$

We conclude by using that $\frac{x}{2} \le \log(1 + x) \le x$ for $x \in [0, 1]$.

$\square$

### H.3 Tightness of the Hellinger Distance Reduction

Theorem 2.1 and Theorem 3.1 are supervised learning reductions that bound the regret of any policy $\widehat{\pi}$ in terms of its Hellinger distance $D_{\mathsf{H}}^2\big(\mathbb{P}^{\widehat{\pi}}, \mathbb{P}^{\pi^\star}\big)$ to the expert policy $\pi^\star$. The following result shows that these reductions are tight in a fairly strong instance-dependent sense: Namely, for any pair of policies $\widehat{\pi}$ and $\pi^\star$, and for any reward-free MDP $M^\star$, it is possible to design a reward function $r = \{r_h\}_{h=1}^H$ for which each term in Eq. (9) of Theorem 3.1 is tight, and such that Theorem 2.1 is tight; the only caveat is that we require the reward function to be *non-Markovian*, in the sense that $r_h$ depends on the full history $x_{1:h}$ and $a_{1:h}$.

**Theorem H.3** (Converse to Theorems 2.1 and 3.1)**.** *Let a reward-free MDP $M^\star$ and a pair of (potentially stochastic) policies $\widehat{\pi}$ and $\pi^\star$ be given.*

*1. For any $R > 0$, there exists a non-Markovian reward function $r = \{r_h\}_{h=1}^H$ with $\sum_{h=1}^H r_h \in [0, R]$ such that*

$$J(\pi^\star) - J(\widehat{\pi}) \ge \frac{R}{6} \cdot D_{\mathsf{H}}^2\big(\mathbb{P}^{\widehat{\pi}}, \mathbb{P}^{\pi^\star}\big). \tag{20}$$

2. *For any $\sigma^2 > 0$, there exists a non-Markovian reward function $r = \{r_h\}_{h=1}^H$ for which $\sigma_{\pi^\star}^2 := \sum_{h=1}^H \mathbb{E}^{\pi^\star}\left[(Q_h^{\pi^\star}(x_{1:h}, a_{1:h}) - V_h^{\pi^\star}(x_{1:h}, a_{1:h-1}))^2\right] \leq \sigma^2$, and such that[22]*

$$J(\pi^\star) - J(\widehat{\pi}) \geq \frac{1}{9}\sqrt{\sigma^2 \cdot D_{\mathsf{H}}^2(\mathbb{P}^{\widehat{\pi}}, \mathbb{P}^{\pi^\star})}. \tag{21}$$

3. *For any $R > 0$ and $\sigma^2 > 0$, there exists a non-Markovian reward function $r = \{r_h\}_{h=1}^H$ with $\sum_{h=1}^H r_h \in [0, R]$ and $\sigma_{\pi^\star}^2 \leq \sigma^2$ simultaneously such that*

$$J(\pi^\star) - J(\widehat{\pi}) \geq \frac{1}{9} \min\left\{ \sqrt{\sigma^2 \cdot D_{\mathsf{H}}^2(\mathbb{P}^{\widehat{\pi}}, \mathbb{P}^{\pi^\star})}, R \cdot D_{\mathsf{H}}^2(\mathbb{P}^{\widehat{\pi}}, \mathbb{P}^{\pi^\star}) \right\}.$$

Eq. (20) shows that there exist reward functions with bounded range for which Theorem 2.1 and the lower-order term in Eq. (9) of Theorem 3.1 are tight, while Eq. (21) shows that there exist reward functions with bounded variance (but not necessarily bounded range) for which the leading term in Eq. (9) or Theorem 3.1 is tight.

Note that for some MDPs, the state $x_h$ already contains the full history $x_{1:h-1}, a_{1:h-1}$, so the assumption of non-Markovian rewards is without loss of generality. For MDPs that do not have this property, Theorem H.3 leaves open the possibility that Theorems 2.1 and 3.1 can be improved on a per-MDP basis.

**Proof of Theorem H.3.** Consider a pair of measures $\mathbb{P}$ and $\mathbb{Q}$, and set $\bar{\mathbb{P}} := \frac{1}{2}(\mathbb{P} + \mathbb{Q})$. Consider the function

$$h = 1 - \frac{1}{2}\frac{\mathbb{Q}}{\bar{\mathbb{P}}} \in [0, 1].$$

Using Lemma D.4, we observe that

$$\mathbb{E}_{\mathbb{P}}[h] - \mathbb{E}_{\mathbb{Q}}[h] = 2\left(\mathbb{E}_{\bar{\mathbb{P}}}[h] - \mathbb{E}_{\mathbb{Q}}[h]\right) = \mathbb{E}_{\mathbb{Q}}\left[\frac{\mathbb{Q}}{\bar{\mathbb{P}}}\right] - \mathbb{E}_{\bar{\mathbb{P}}}\left[\frac{\mathbb{Q}}{\bar{\mathbb{P}}}\right] = D_{\chi^2}(\mathbb{Q} \| \bar{\mathbb{P}}) \geq \frac{1}{6}D_{\mathsf{H}}^2(\mathbb{Q}, \mathbb{P}). \tag{22}$$

We also observe that by concavity of variance,

$$\frac{1}{2}\left(\mathrm{Var}_{\mathbb{P}}[h] + \mathrm{Var}_{\mathbb{Q}}[h]\right) \leq \mathrm{Var}_{\bar{\mathbb{P}}}[h] = \frac{1}{4}\mathbb{E}_{\bar{\mathbb{P}}}\left[\left(\frac{\mathbb{Q}}{\bar{\mathbb{P}}} - \mathbb{E}_{\bar{\mathbb{P}}}\left[\frac{\mathbb{Q}}{\bar{\mathbb{P}}}\right]\right)^2\right] = D_{\chi^2}(\mathbb{Q} \| \bar{\mathbb{P}}) \leq D_{\mathsf{H}}^2(\mathbb{Q}, \mathbb{P}). \tag{23}$$

To apply this observation to the theorem at hand, let a parameter $B > 0$ be given, let $\bar{\mathbb{P}} := \frac{1}{2}(\mathbb{P}^{\pi^\star} + \mathbb{P}^{\widehat{\pi}})$, and consider the non-Markov reward function $r$ that sets $r_1, \ldots, r_{h-1} = 0$ and

$$r_H(\tau) = B \cdot \left(1 - \frac{1}{2}\frac{\mathbb{P}^{\widehat{\pi}}}{\bar{\mathbb{P}}}\right) \in [0, B].$$

Then by Eq. (22), we have that

$$J(\pi^\star) - J(\widehat{\pi}) \geq \frac{B}{6} \cdot D_{\mathsf{H}}^2\left(\mathbb{P}^{\widehat{\pi}}, \mathbb{P}^{\pi^\star}\right).$$

At the same time, by Eq. (23), we have that

$$\mathrm{Var}^{\pi^\star}\left[\sum_{h=1}^H r_h\right] = \mathrm{Var}^{\pi^\star}[r_H] \leq 2B^2 \cdot D_{\mathsf{H}}^2\left(\mathbb{P}^{\widehat{\pi}}, \mathbb{P}^{\pi^\star}\right),$$

and by Proposition 3.1,

$$\sigma_{\pi^\star}^2 \leq \mathrm{Var}^{\pi^\star}\left[\sum_{h=1}^H r_h\right].$$

---

[22]Note that since the reward function under consideration is non-Markovian, the value functions $Q_h^{\pi^\star}$ and $V_h^{\pi^\star}$ depend on the full history $x_{1:h}, a_{1:h-1}$.

To conclude, note that if we set $B^2 = R^2$, then $\sum_{h=1}^{H} r_h \in [0, R]$ and

$$J(\pi^\star) - J(\widehat{\pi}) \geq \frac{R}{6} \cdot D_{\mathsf{H}}^2\left(\mathbb{P}^{\widehat{\pi}}, \mathbb{P}^{\pi^\star}\right).$$

Meanwhile, if we set

$$B^2 = \frac{\sigma^2}{2 D_{\mathsf{H}}^2(\mathbb{P}^{\widehat{\pi}}, \mathbb{P}^{\pi^\star})},$$

then $\sigma_{\pi^\star}^2 \leq \sigma^2$ and

$$J(\pi^\star) - J(\widehat{\pi}) \geq \frac{1}{9}\sqrt{\sigma^2 \cdot D_{\mathsf{H}}^2(\mathbb{P}^{\widehat{\pi}}, \mathbb{P}^{\pi^\star})}.$$

Finally, if we set

$$B^2 = \frac{\sigma^2}{2 D_{\mathsf{H}}^2(\mathbb{P}^{\widehat{\pi}}, \mathbb{P}^{\pi^\star})} \wedge R^2.$$

Then $\sum_{h=1}^{H} r_h \in [0, R]$, $\sigma_{\pi^\star}^2 \leq \sigma^2$, and

$$J(\pi^\star) - J(\widehat{\pi}) \geq \frac{B}{6} \cdot D_{\mathsf{H}}^2\left(\mathbb{P}^{\widehat{\pi}}, \mathbb{P}^{\pi^\star}\right) \geq \min\left\{\frac{1}{9}\sqrt{\sigma^2 \cdot D_{\mathsf{H}}^2(\mathbb{P}^{\widehat{\pi}}, \mathbb{P}^{\pi^\star})}, \frac{R}{6} \cdot D_{\mathsf{H}}^2\left(\mathbb{P}^{\widehat{\pi}}, \mathbb{P}^{\pi^\star}\right)\right\}.$$

$\square$

# I  Benefits of Online Interaction

Our results in Sections 2 and 3 show that the benefits of online interaction in imitation learning—to the extent that horizon is concerned—are more limited than previously thought. We expect that in practice, online interaction will likely lead to benefits, but only in a problem-dependent sense. To this end, we first discuss the role of misspecification and the realizability assumption used by our results, then highlight several special cases in which online interaction is indeed beneficial, but in a policy class-dependent fashion not captured by existing theory. In particular, we identify three phenomena which lead to improved sample complexity: (i) *representational benefits*; (ii) *value-based feedback*; and (iii) *exploration*. Our results in this section can serve as a starting point toward developing a more fine-grained understanding of algorithms and sample complexity of imitation learning.

## I.1  The Role of Misspecification

This paper (for both deterministic and stochastic experts) focuses on the realizable setting in which $\pi^\star \in \Pi$. It is natural to ask how the role of horizon in imitation learning changes under misspecification, and whether online interaction brings greater benefits in this case. This is a subtle issue, as there are various incomparable notions of misspecification error which can lead to different forms of horizon dependence. For example, for deterministic experts, if $\Pi$ is misspecified in the sense that $\inf_{\pi \in \Pi} L_{\mathsf{bc}}(\pi) \leq \varepsilon_{\mathsf{apx}}$, the indicator-loss behavior cloning algorithm in ?? achieves $J(\pi^\star) - J(\widehat{\pi}) \lesssim RH \cdot \left(\frac{\log(|\Pi|\delta^{-1})}{n} + \varepsilon_{\mathsf{apx}}\right)$, which is tight in general. In other words, the dependence on $\varepsilon_{\mathsf{apx}}$ is not horizon-independent. On the other hand, as we show in Appendix E, if we assume that $\inf_{\pi \in \Pi} D_{\chi^2}\left(\mathbb{P}^{\pi^\star} \| \mathbb{P}^{\pi}\right) \leq \varepsilon_{\mathsf{apx}}$, a stronger notion of misspecification error, then LogLossBC achieves a horizon-independent guarantee of the form $J(\pi^\star) - J(\widehat{\pi}) \lesssim R \cdot \left(\frac{\log(|\Pi|\delta^{-1})}{n} + \varepsilon_{\mathsf{apx}}\right)$. We leave a detailed investigation of tradeoffs between misspecification and horizon (as well as interplay with online versus offline IL) for future work; by giving the first horizon-independent treatment for the realizable setting, we hope that our results can serve as a starting point.

## I.2  Representational Benefits

The classical intuition behind algorithms like Dagger and Aggrevate (which Definition 1.1 attempts to quantify) is *recoverability*: through online access, we can learn to correct the mistakes of an imperfect policy. Our results in Sections 2 and 3 show that recoverability has limited benefits for stationary policy classes as far as horizon is concerned. In spite of this, the following proposition shows that recoverability can have pronounced benefits for *representational* reasons, even with constant horizon.

**Proposition I.1** (Representational benefits of online IL). *For any $N \in \mathbb{N}$, there exists a class $\mathcal{M}$ of MDPs with $H = 2$ and a policy class $\Pi$ with $\log|\Pi| = O(N)$ such that*

- *There is an online imitation learning algorithm that achieves $J(\pi^\star) - J(\widehat{\pi}) = 0$ with probability at least $1 - \delta$ using $O(\log(\delta^{-1}))$ episodes for any MDP $M^\star \in \mathcal{M}$ and expert policy $\pi^\star \in \Pi$. In particular, this can be achieved by* Dagger.

- *Any proper offline imitation learning algorithm requires $n = \Omega(N)$ trajectories to learn a non-trivial policy with $J(\pi^\star) - J(\widehat{\pi}) \le c$ for an absolute constant $c > 0$.[23]*

The idea behind this construction is as follows: The behavior of the (stochastic) expert policy at step $h = 1$ is very complex, and learning to imitate it well in distribution (e.g., with respect to total variation or Hellinger distance) is a difficult representation learning problem (in the language of Section 2, e.g., Theorem 2.1, we must take $\log|\Pi_1|$ very large in order to realize $\pi_1^\star$). For offline imitation learning, we have no choice but to imitate $\pi_1^\star$ well at $h = 1$, leading to the lower bound in Proposition I.1. With online access though, we can give up on learning $\pi_1^\star$ well, and instead learn to correct our mistake at step $h = 2$. For the construction in Proposition I.1, this a much easier representation learning problem, and requires very low sample complexity (i.e., we can realize $\pi_2^\star$ with a class $\Pi_2$ for which $\log|\Pi_2|$ is small). We conclude that Dagger can indeed lead to substantial benefits over offline IL, but for representational reasons unrelated to horizon, and not captured by existing theory. While this example is somewhat contrived, it suggests that potential to develop a deeper understanding of representational benefits in imitation learning, which we leave as a promising direction for future work.

### I.3 Benefits of Value-Based Feedback

Beginning with the work of Ross and Bagnell [71] on Aggrevate, many works (e.g., Sun et al. [78]) consider a *value-based feedback* variant of the online IL framework (Section 1.1) where in addition to (or instead of) observing $a_h^\star$, the learner observes the expert's advantage function $A_h^{\pi^\star}(x_h, \cdot) := Q_h^{\pi^\star}(x_h, \pi_h^\star(x_h)) - Q_h^{\pi^\star}(x_h, \cdot)$ or value function $Q_h^{\pi^\star}(x_h, \cdot)$ at every state visited by the learner (see Appendix I.5.2 for details, which are deferred to the appendix for space). While such feedback intuitively seems useful, existing theoretical guarantees—to the best of our knowledge—[71, 78] only show that algorithms like Aggrevate are no worse than non-value based methods like Dagger, and do not quantify situations in which value-based feedback actually leads to improvement.[24]

The following result shows that i) value-based feedback can lead to arbitrarily large improvement over non-value based feedback for representational reasons similar to Proposition I.1 (that is for a complicated stochastic expert, learning to optimize a fixed value function can be much easier than learning to imitate the expert well in TV distance), but ii) it is only possible to exploit value-based feedback in this fashion under online interaction (that is, even if we annotate the trajectories for offline imitation learning with $A_h^{\pi^\star}(x_h, \cdot)$ for the visited states, this cannot lead to improvement in sample complexity).

**Proposition I.2** (Benefits of value-based feedback (informal)). *For any $N \in \mathbb{N}$, there is a class of MDPs $\mathcal{M}$ with $H = 2$ and a policy class $\Pi$ with $\log|\Pi| = O(N)$ such that*

- *There is an online imitation learning algorithm with value-based feedback that achieves $J(\pi^\star) - J(\widehat{\pi}) = 0$ with probability at least $1 - \delta$ using $O(\log(\delta^{-1}))$ episodes for every MDP $M^\star \in \mathcal{M}$ and expert $\pi^\star \in \Pi$. In particular, this can be achieved by* Aggrevate.

- *Any proper offline imitation learning algorithm (with value-based feedback) or proper online imitation learning algorithm (without valued-based feedback) requires $n = \Omega(N)$ trajectories to learn a non-trivial policy with $J(\pi^\star) - J(\widehat{\pi}) \le c$ for an absolute constant $c > 0$.[25]*

As with Proposition I.1, this example calls for a fine-grained policy class-dependent theory, which we hope to explore more deeply in future work.

---

[23]We expect that this result extends to *improper* offline IL algorithms for which $\widehat{\pi} \notin \Pi$, but a more complicated construction is required; we leave this for the next version of the paper.

[24]These results are reductions which bound regret in terms of different notions of supervised learning performance, which makes it somewhat difficult to compare them or derive concrete end-to-end guarantees.

[25]As with Proposition I.1, we expect that this lower bound can be extended to improper learners, but a more complicated construction is required.

## I.4 Benefits from Exploration

A final potential benefit of online interaction arises in *exploration*. One might hope that with online access, we can directly guide the MDP to informative states that will help to identify the optimal policy faster. The following proposition gives an example in which deliberate exploration can lead to arbitrarily large improvement over offline imitation learning, as well as over naive online imitation learning algorithms like Dagger that do not deliberately explore.

**Proposition I.3** (Benefits of exploration for online IL)**.** *For any $n \in \mathbb{N}$ and $H \in \mathbb{N}$, there exists an MDP $M^\star$ and a class of deterministic policies $\Pi$ with $|\Pi| = 2$ with the following properties.*

1. *There exists an online imitation learning algorithm that returns a policy $\widehat{\pi}$ such that $J(\pi^\star) - J(\widehat{\pi}) = 0$ with probability at least $1 - \delta$ using $O(\log(\delta^{-1}))$ episodes, for* all *possible reward functions (i.e., even if $\mu = H$).*

2. *For any offline imitation learning algorithm, there exists a deterministic reward function $r = \{r_h\}_{h=1}^H$ and expert policy $\pi^\star \in \Pi$ with $\mu = 1$ such that any algorithm must have $\mathbb{E}[J(\pi^\star) - J(\widehat{\pi})] \geq \Omega(1) \cdot \frac{H}{n}$. In addition,* Dagger *has regret $\mathbb{E}[J(\pi^\star) - J(\widehat{\pi})] \geq \Omega(1) \cdot \frac{H}{n}$.*

The idea behind this construction is simple: We take the lower bound construction from Theorem 2.2 and augment it with a "revealing" which directly reveals the identity of the underlying expert. The true expert never visits this state, so offline imitation learning algorithms cannot exploit it (standard online IL algorithms like Dagger and relatives do not exploit the revealing state for the same reason),[26] but a well-designed online IL algorithm that deliberately navigates to the revealing state can use it to identify $\pi^\star$ extremely quickly.

As with the previous examples, this construction is somewhat contrived, but it suggests that directly maximizing information acquisition may be a useful algorithm design paradigm for online IL, and we hope to explore this more deeply in future work.

## I.5 Proofs

### I.5.1 Proof of Proposition I.1

**Proof of Proposition I.1.** Let $N \in \mathbb{N}$ be given. We set $\mathcal{X} = \{\mathfrak{x}, \mathfrak{y}, \mathfrak{z}\}$, $\mathcal{A} = [N] \cup \{\mathfrak{a}, \mathfrak{b}\}$, and $H = 2$. We consider a family of problem instances $\{(M, \pi^\star, r)\}$ indexed by a subset $S \subset [N]$ with $|\mathcal{S}| = N/2$ and an action $a^\star \in \{\mathfrak{a}, \mathfrak{b}\}$ as follows. For a given pair $(S, a^\star)$:

- The dynamics are as follows. We have $x_1 = \mathfrak{x}$ deterministically. For simplicity, we assume that only actions in $[N]$ are available at step $h = 1$. If $a_1 \in S$, then $x_2 = \mathfrak{y}$, otherwise $x_2 = \mathfrak{z}$.

- The reward at step 1 is $r_1(\cdot, \cdot) = 0$, and the reward at step 2 is given by $r_2(\mathfrak{y}, \cdot) = 1$ and $r_2(\mathfrak{z}, a) = \mathbb{I}\{a = a^\star\}$.

- The expert $\pi^\star$ sets $\pi^\star(\mathfrak{x}) = \mathrm{unif}(S)$, $\pi^\star(\mathfrak{y}) = \mathrm{unif}(\{\mathfrak{a}, \mathfrak{b}\})$, and $\pi^\star(\mathfrak{z}) = a^\star$.

Let us refer to the problem instance above as $\mathcal{I}_{S,a^\star} = \{(M_{S,a^\star}, \pi^\star_{S,a^\star}, r_{S,a^\star})\}$, and let $J_{S,a^\star}(\pi)$ denote the expected reward under this instance.

**Upper bound for online imitation learning.** Consider the algorithm that sets $\widehat{\pi}_1^i = \mathrm{unif}([N])$ for each $i \in [n]$. If we play for $n = \log_2(\delta^{-1})$ episodes, we will see $x_2 = \mathfrak{z}$ in at least one episode with probability at least $1 - \delta$, at which point we will observe $a^\star = \pi^\star(\mathfrak{z})$, and we can return the policy $\widehat{\pi}$ that sets $\widehat{\pi}_1(\mathfrak{x}) = \mathrm{unif}([N])$ and $\widehat{\pi}_2(\cdot) = a^\star$; this policy has zero regret.

Note that if we define $\Pi = \{\pi^\star_{S,a^\star}\}_{|S|=N/2, a^\star \in \{\mathfrak{a},\mathfrak{b}\}}$ as the natural policy class for the family of instances above, then the algorithm above is equivalent to running Dagger with the online learning algorithm that, at iteration $i$, sets

$$\widehat{\pi}_h^i = \mathrm{unif}\big(\{\pi \in \Pi_h \mid \pi_2(\mathfrak{z}) = a_2^{\star,j} \ \forall j < i : x_2^j = \mathfrak{z}\}\big),$$

and choosing the final policy as $\widehat{\pi} = \widehat{\pi}^i$ for any iteration $i$ after $x_2 = \mathfrak{z}$ is encountered.

---

[26]This phenomenon is also distinct from "active" online imitation learning algorithms [75] which can obtain improved sampling complexity under strong distributional assumptions in the vein of active learning [40], but still do not deliberately explore.

**Lower bound for offline imitation learning.** Consider the offline imitation learning setting. When the underlying instance is $\mathcal{I}_{S,a^\star}$, we observe a dataset $\mathcal{D}$ consisting of $n$ trajectories generated by executing $\pi_{S,a^\star}^\star$ in $M_{S,a^\star}$. The trajectories never visit the state $\mathfrak{z}$, so $a^\star$ is not identifiable, and we can do no better than guessing $a^\star$ uniformly in this state. Letting $\mathbb{E}_{S,a^\star}$ denote the law of $\mathcal{D}$ under instance $\mathcal{I}_{S,a^\star}$, we have $J_{S,a^\star}(\widehat{\pi}) = \widehat{\pi}_1(S \mid \mathfrak{x}) + \widehat{\pi}_1(S^c \mid \mathfrak{x})\widehat{\pi}_2(a^\star \mid \mathfrak{z})$. It follows that for any $S$, since the law of $\mathcal{D}$ does not depend on $a^\star$,

$$\max_{a^\star \in \{\mathfrak{a},\mathfrak{b}\}} \mathbb{E}_{S,a^\star}\left[J_{S,a^\star}(\pi_{S,a^\star}^\star) - J_{S,a^\star}(\widehat{\pi})\right] \geq \mathbb{E}_{S,\mathfrak{a}}[1 - \widehat{\pi}_1(S \mid \mathfrak{x}) - \widehat{\pi}_1(S^c \mid \mathfrak{x})/2]$$

$$= \frac{1}{2}\mathbb{E}_{S,\mathfrak{a}}[1 - \widehat{\pi}_1(S \mid \mathfrak{x})].$$

Note that if $\widehat{\pi}$ is proper in the sense that $\widehat{\pi}_1(\cdot\mathfrak{x}) = \mathrm{unif}(\widehat{S})$ for some $\widehat{S} \subset [N]$ with $|\widehat{S}| = N/2$, we have $1 - \widehat{\pi}_1(S \mid \mathfrak{x}) = 1 - \frac{2}{N}|\widehat{S} \cup S|$. We conclude that if $\mathbb{E}_{S,a^\star}\left[J_{S,a^\star}(\pi_{S,a^\star}^\star) - J_{S,a^\star}(\widehat{\pi})\right] \leq \frac{1}{8}$, then $\mathbb{E}_{S,a^\star}\left[|\widehat{S} \cap S|\right] \geq \frac{3}{8}N$. From here, it follows from standard lower bounds for discrete distribution estimation (e.g., Canonne [18]) that any such estimator $\widehat{S}$ requires $n = \Omega(N)$ samples for a worst-case choice of $S$. $\qquad\square$

### I.5.2 Background and Proof for Proposition I.2

Before proving Proposition I.2, we first formally introduce the value-based feedback model we consider.

**Background on value-based feedback.** We can consider two models for imitation learning with value-based feedback, inspired by Ross and Bagnell [71], Sun et al. [78].

- **Offline setting.** In the offline setting, we receive $n$ trajectories $(x_1, a_1), \ldots, (x_H, a_H)$ generated by executing $\pi^\star$ in $M^\star$. For each state in each such trajectory, we observe $A_h^{\pi^\star}(x_h, \cdot)$, where $A_h^{\pi^\star}(x, a) = Q_h^{\pi^\star}(x, \pi^\star(x)) - Q_h^{\pi^\star}(x, a)$ is the advantage function for $\pi^\star$.[27]

- **Online setting.** The online setting is as follows. There are $n$ at episodes. For each episode $i$, we execute a policy $\widehat{\pi}^i$, and receive a "trajectory" $o^i = (x_1^i, a_1^i, a_1^{\star,i}), \ldots, (x_H^i, a_H^i, a_H^{\star,i})$, where $a_h^i \sim \widehat{\pi}^i(x_h^i)$ and $a_h^{\star,i} \sim \pi^\star(x_h^i)$. In addition, for each state in the trajectory, we observe $A_h^{\pi^\star}(x_h, \cdot)$. After the $n$ episodes conclude, we output a final policy $\widehat{\pi}$ on which performance is evaluated.

**Proof of Proposition I.2.** We only sketch the proof, as it is quite similar to Proposition I.1. Let $N \in \mathbb{N}$ be given. We set $\mathcal{S} = \{\mathfrak{x}, \mathfrak{y}, \mathfrak{z}\}$, $\mathcal{A} = [N]$, and $H = 2$. We consider a class of problem instances $\{(M, \pi^\star, r)\}$ indexed by sets $S_1, S_2 \subset [N]$ with $|\mathcal{S}_1| = |\mathcal{S}_2| = N/2$ defined as follows. For a given pair $(S_1, S_2)$:

- The dynamics are as follows. We have $x_1 = \mathfrak{x}$ deterministically. If $a_1 \in S_1$, then $x_2 = \mathfrak{y}$, otherwise $x_2 = \mathfrak{z}$.

- The reward function sets $r_1(\mathfrak{x}, \cdot) = 0$, $r_2(\mathfrak{y}, \cdot) = 1$, and $r_2(\mathfrak{z}, a) = \mathbb{I}\{a \in S_2\}$.

- The expert $\pi^\star$ sets $\pi^\star(\mathfrak{x}) = \mathrm{unif}(S_1)$, $\pi^\star(\mathfrak{z}) = \mathrm{unif}(S_2)$, and $\pi^\star(\mathfrak{y}) = \mathrm{unif}([N])$

We refer to the problem instance above as $\mathcal{I}_{S_1,S_2} = (M_{S_1,S_2}, \pi_{S_1,S_2}^\star, r_{S_1,S_2})$, and let $J_{S_1,S_2}(\pi)$ denote the expected reward under this instance.

**Upper bound for online imitation learning with value-based feedback.** Consider an algorithm that sets $\widehat{\pi}_1^i = \mathrm{unif}([N])$ for each $i \in [n]$. If we play for $n = \log_2(\delta^{-1})$ episodes, we will see $x_2 = \mathfrak{z}$ in at least one episode with probability at least $1 - \delta$, at which point we will observe $A_2^{\pi^\star}(\mathfrak{z}, \cdot)$. We can pick an arbitrary action with $A_2^{\pi^\star}(\mathfrak{z}, \cdot) = 0$ and return the policy $\widehat{\pi}$ that sets $\widehat{\pi}_1(\mathfrak{x}) = \mathrm{unif}([N])$ and $\widehat{\pi}_2(\cdot) = a$; this policy has zero regret.

Note that if we define $\Pi = \left\{\pi_{S_1,S_2}^\star\right\}_{|S_1|=|S_2|=N/2}$ as the natural policy class for the family of instances above, then the algorithm above is equivalent to running `Aggrevate` with the online learning algorithm that, at iteration $i$, sets

$$\widehat{\pi}_h^i = \mathrm{unif}\left(\left\{\pi \in \Pi_h \mid \pi_2(\mathfrak{z}) \in \arg\max_a A_2^{\pi^\star}(x_2^j, a) \;\forall j < i : x_2^j = \mathfrak{z}\right\}\right),$$

---

[27]Our results are not sensitive to whether the learner observes the advantage function or the value function itself; we choose this formulation for concreteness.

and choosing the final policy as $\widehat{\pi} = \widehat{\pi}^i$ for any iteration $i$ after $x_2 = \mathfrak{z}$ is encountered.

**Lower bound for offline imitation learning.** Consider the offline imitation learning setting. When the underlying instance is $\mathcal{I}_{S_1, S_1}$, we observe a dataset $\mathcal{D}$ consisting of $n$ trajectories generated by executing $\pi^\star_{S_1, S_2}$ in $M_{S_1, S_2}$. The trajectories never visit the state $\mathfrak{z}$, so $S_2$ is not identifiable, and we can do no better than guessing uniformly in this state. Letting $\mathbb{E}_{S_1, S_2}$ denote the law of $\mathcal{D}$ under instance $\mathcal{I}_{S_1, S_2}$, we have $J_{S_1, S_2}(\widehat{\pi}) = \widehat{\pi}_1(S_1 \mid \mathfrak{x}) + \widehat{\pi}_1(S_1^c \mid \mathfrak{x})\widehat{\pi}_2(S_2 \mid \mathfrak{z})$. It follows that for any $(S_1, S_2)$, since the law of $\mathcal{D}$ does not depend on $S_2$,

$$\max_{S_2 : |S_2| = N/2} \mathbb{E}_{S_1, S_2}\big[J_{S_1, S_2}(\pi^\star_{S_1, S_2}) - J_{S_1, S_2}(\widehat{\pi})\big] \geq \mathbb{E}_{S_1}[1 - \widehat{\pi}_1(S_1 \mid \mathfrak{x}) - \widehat{\pi}_1(S_1^c \mid \mathfrak{x})/2]$$

$$= \frac{1}{2}\mathbb{E}_{S_1}[1 - \widehat{\pi}_1(S_1 \mid \mathfrak{x})],$$

with the convention that $\mathbb{E}_{S_1}$ denotes the law of $\mathcal{D}$ for an arbitrary choice of $\mathcal{S}_2$. If $\widehat{\pi}$ is proper in the sense that $\widehat{\pi}_1(\cdot \mathfrak{x}) = \mathrm{unif}(\widehat{S_1})$ for some $\widehat{S_1} \subset [N]$ with $|\widehat{S_1}| = N/2$, we have $1 - \widehat{\pi}_1(S_1 \mid \mathfrak{x}) = 1 - \frac{2}{N}|\widehat{S_1} \cup S_1|$. We conclude that if $\mathbb{E}_{S_1, S_2}\big[J_{S_1, S_2}(\pi^\star_{S_1, S_2}) - J_{S_1, S_2}(\widehat{\pi})\big] \leq \frac{1}{8}$, then $\mathbb{E}_{S_1,}\big[|\widehat{S_1} \cap S_1|\big] \geq \frac{3}{8}N$. From here, it follows from standard lower bounds for discrete distribution estimation (e.g., Canonne [18]) that any such estimator $\widehat{S}$ requires $n = \Omega(N)$ samples for a worst-case choice of $S$.

**Lower bound for online imitation learning without value-based-feedback.** Consider an online imitation learning algorithm that does not receive value-based feedback. We claim, via an argument similar to the one above, that if the algorithm that ensures

$$\mathbb{E}_{S_1, S_2}\big[J_{S_1, S_2}(\pi^\star_{S_1, S_2}) - J_{S_1, S_2}(\widehat{\pi})\big] \leq c$$

on all instances for a sufficiently small absolute constant $c$, then it can be used to produce estimators $\widehat{S_1}, \widehat{S_2} \subset [N]$ such that with constant probability, either $\big|\widehat{S_1} \cap S_1\big| \geq \frac{3}{8}N$ or $\big|\widehat{S_2} \cap S_2\big| \geq \frac{3}{8}N$. From here, it should follow from standard arguments that this requires $n = \Omega(N)$ samples for a worst-case choice of $S_1$ and $S_2$.

$\square$

### I.5.3 Proof of Proposition I.3

**Proof of Proposition I.3.** We consider a slight variant of the construction from Theorem 2.2. Let $n$ and $H$ be given, and let $\Delta \in (0, 1/3)$ be a parameter whose value will be chosen later. We first specify the dynamics for $M^\star$. Set $\mathcal{X} = \{\mathfrak{x}, \mathfrak{y}, \mathfrak{z}\}$ and $\mathcal{A} = \{\mathfrak{a}, \mathfrak{b}, \mathfrak{c}\}$. The initial state distribution sets $P_0(\mathfrak{x}) = 1 - \Delta$ and $P_0(\mathfrak{y}) = \Delta$. The transition dynamics are:

- $P_h(x' = \cdot \mid x = \mathfrak{x}, a) = \mathbb{I}_{\mathfrak{x}} \cdot \mathbb{I}\{a \in \{\mathfrak{a}, \mathfrak{b}\}\} + \mathbb{I}_{\mathfrak{z}} \cdot \mathbb{I}\{a = \mathfrak{c}\}$.

- $P_h(x' \mid x, a) = \mathbb{I}\{x' = x\}$ for $x \in \{\mathfrak{y}, \mathfrak{z}\}$.

In other words, $\mathfrak{y}$ and $\mathfrak{z}$ are terminal states. For state $\mathfrak{x}$, actions $\mathfrak{a}$ and $\mathfrak{b}$ are self-loops, but action $\mathfrak{c}$ transitions to $\mathfrak{z}$.

The expert policies are $\pi^\mathfrak{a}$, which sets $\pi^\mathfrak{a}_h(x) = \mathfrak{a}$ for all $h$ and $x \in \mathcal{X}$, and $\pi^\mathfrak{b}$, which sets $\pi^\mathfrak{b}_h(\mathfrak{x}) = \mathfrak{a}$ and sets $\pi^\mathfrak{b}_h(\mathfrak{y}) = \pi^\mathfrak{b}_h(\mathfrak{z}) = \mathfrak{b}$. We have $\Pi = \{\pi^\mathfrak{a}, \pi^\mathfrak{b}\}$.

We consider two problem instances for the lower bound, $\mathcal{I}^\mathfrak{a} = (M^\star, \pi^\mathfrak{a}, r^\mathfrak{a})$, and $\mathcal{I}^\mathfrak{b} = (M^\star, \pi^\mathfrak{b}, r^\mathfrak{b})$. For problem instance $\mathcal{I}^\mathfrak{a}$, the expert policy is $\pi^\mathfrak{a}$. We set $r^\mathfrak{a}_h(\mathfrak{x}, \cdot) = r^\mathfrak{a}_h(\mathfrak{z}, \cdot) = 0, r^\mathfrak{a}_h(\mathfrak{y}, a) = \mathbb{I}\{a = \mathfrak{a}\}$ for all $h$. On the other hand, for problem instance $\mathcal{I}^\mathfrak{b}$, the expert policy is $\pi^\mathfrak{b}$. We set $r^\mathfrak{b}_h(\mathfrak{x}, \cdot) = r^\mathfrak{b}_h(\mathfrak{z}, \cdot) = 0, r^\mathfrak{b}_h(\mathfrak{y}, a) = \mathbb{I}\{a = \mathfrak{b}\}$ for all $h$. Note that both of these choices for the reward function satisfy $\mu = 1$, and that $\pi^\mathfrak{a}$ and $\pi^\mathfrak{b}$ are optimal policies for the respective instances. Let $J^\mathfrak{a}$ denote the expected reward function for instance $\mathfrak{a}$, and likewise for $\mathfrak{b}$.

**Upper bound on online sample complexity.** We consider the following online algorithm. For episodes $t = 1, \ldots, :$

- If $x_1 \neq \mathfrak{x}$, proceed to the next episode.

- If $x_1 = \mathfrak{x}$, take action $\mathfrak{c}$, and observe $a_2 = \pi^\star(\mathfrak{z})$. If $a_2 = \mathfrak{a}$, return $\widehat{\pi} = \pi^\mathfrak{a}$, and if $a_2 = \mathfrak{b}$, return $\widehat{\pi} = \pi^\mathfrak{b}$.

For any $\Delta \le e^{-1}$, this algorithm will terminate after $\log(1/\delta)$ episodes with probability at least $1 - \delta$, and whenever the algorithm terminates, it is clear that $\widehat{\pi} = \pi^\star$. In particular, this leads to zero regret for *any choice of reward function.*

**Lower bound on offline sample complexity.** By setting $\Delta \propto \frac{1}{n}$, an argument essentially identical to the proof of Theorem 2.2 shows that any offline imitation learning algorithm must have

$$\max\{\mathbb{E}^{\mathrm{a}}[J^{\mathrm{a}}(\pi^{\mathrm{a}}) - J^{\mathrm{a}}(\widehat{\pi})], \mathbb{E}^{\mathrm{b}}[J^{\mathrm{b}}(\pi^{\mathrm{b}}) - J^{\mathrm{b}}(\widehat{\pi})]\} \gtrsim \Delta H \gtrsim \frac{H}{n}.$$

For the sake of avoiding repetition, we omit the details. Finally, we observe that since neither policy in $\Pi$ takes the action $\mathfrak{c}$, Dagger—when equipped with any online learning algorithm that predicts from a mixture of policies in $\Pi$, such as in Proposition E.2)—will never take the action $\mathfrak{c}$, and hence is subject to the $\frac{H}{n}$ lower bound from Theorem 2.2 as well.

$\square$

