# OpenReview forum: "Is Behavior Cloning All You Need? Understanding Horizon in Imitation Learning"
_NeurIPS.cc/2024/Conference — NeurIPS 2024 spotlight_

### Official Review · Reviewer_FtQL · 2024-07-09

**Soundness:** 3
**Presentation:** 3
**Contribution:** 3
**Rating:** 7
**Confidence:** 4

**Summary:**

This paper presents a thorough analysis of imitation learning (IL), focusing on the gap between offline and online IL. The authors demonstrate that behavior cloning (BC) with logarithmic loss can achieve horizon-independent sample complexity under specific conditions, providing valuable insights into the theoretical underpinnings of IL.

**Strengths:**

- This paper presents extensive theoretical results for offline IL and online IL under the general function approximation class. It builds a sharp separation between offline and online algorithms, which the reviewer appreciates a lot.
- The reviewer believes that the analysis tools in this paper can be further applied in future studies of imitation learning and sequential decision making.

**Weaknesses:**

- The theoretical results have a gap with practice. For example, the reviewer finds it odd that the conclusion of a 'horizon-free' guarantee is obtained by assuming the policy class is well-controlled. Practitioners often choose a large function class to ensure the approximation error is small.
- Some key related work is missing.

**Questions:**

**Experiments:**

- There is a gap between the developed theory and practice. It is not safe to conclude that the theory can explain the empirical results in Figure 1. There are possible theoretical explanations for empirical results that differ from this paper. Specifically, [1] showed that MuJoCo locomotion controls have deterministic transitions, thus the statistical estimation error comes from the initial step, and there are no compounding errors. The reviewer believes the explanation provided in [1] is also true for the studied Atari environments.
- The reviewer believes that practitioners choose stationary policies in practice because the MDP formulation for practical tasks has stationary transitions, rather than the non-stationary ones studied in this paper. For stationary-transition MDPs, the estimation error is, in fact, smaller.

**Related Work:**

- [2] studied that adversarial imitation learning algorithms can achieve horizon-free sample complexity for structured MDPs. Their results are also important for understanding the horizon in imitation learning. This work should be reviewed.

[1] Li, Ziniu, et al. "Rethinking ValueDice: Does it really improve performance?." *arXiv preprint arXiv:2202.02468* (2022).

[2] Xu, Tian, et al. "Understanding adversarial imitation learning in small sample regime: A stage-coupled analysis." *arXiv preprint arXiv:2208.01899* (2022).

**Limitations:**

To the reviewer's understanding, this paper mainly focuses on the model selection problem under general function class approximation. This does not provide many new insights on the compounding errors issues and horizon dependence issues in imitation learning. The title may be revised to reflect the contribution.

---

> ### Author Rebuttal · Authors · 2024-08-06
>
> We thank the reviewer for their positive review and their helpful comments! Please see responses to individual questions below.
>
> > The theoretical results have a gap with practice. For example, the reviewer finds it odd that the conclusion of a 'horizon-free' guarantee is obtained by assuming the policy class is well-controlled. Practitioners often choose a large function class to ensure the approximation error is small.
>
> We assume that the reviewer is referring to the fact that our results assume realizability and scale with the complexity of the policy class. We remark:
>
> - Realizability is a standard assumption in RL theory, and is used as a starting point for virtually all existing theoretical results, cf. “Reinforcement learning: Theory and algorithms” (Agarwal, Jiang, Kakade, and Sun).
>
> - Our main results (Theorems 2.1 and 2.3) do not assume that $\Pi$ is well controlled. Rather, these theorems are *reductions* (in the spirit of prior work on IL) showing that $H$-independent rollout performance is achievable when an appropriate notion of supervised learning error is small. Thus if the policy $\hat{\pi}$ has good supervised learning performance (regardless of whether it belongs to a finite class or enjoys a provable generalization bound), our theorems imply that it will have good rollout performance. Corollaries 2.1 and 2.2 instantiate these results for finite classes for concreteness.
>
> - We note that prior work does not achieve $H$-free guarantees *even in the case where $\Pi$ is well controlled*. Our results show that with properly normalized rewards, generalization error of $\Pi$ is the *only* source of $H$ dependence. In some sense, this is the strongest guarantee one might hope for, as the complexity of $\Pi$ influences regret even absent sequential structure in the IL problem. For stationary policies with parameter sharing, our results can be instantiated to give end-to-end guarantees with no dependence on $H$.
>
> - Our most general guarantees for LogLossBC (see Appendix E and Theorem E.1) support infinite policy classes and misspecification, allowing one to trade off policy class complexity with approximation error. The conclusion is the same as for our main results: As long as the supervised learning error (policy class complexity + approximation error) is small, the rollout performance will be horizon-independent.
>
> > Some key related work is missing. [2] studied that adversarial imitation learning algorithms can achieve horizon-free sample complexity for structured MDPs. Their results are also important for understanding the horizon in imitation learning.
>
> Thank you for pointing us to this work, which we are happy to discuss in the final version. While [2] contains results concerning $H$-independence, their findings are quite restrictive compared to our paper ( e.g. their notion of horizon independence is not consistent with that in prior work in RL, e.g., Jiang & Agarwal ‘18). Also, their work:
> - Is restricted to tabular MDPs and policies (while our work considers general MDPs and function approximation).
> - Requires knowledge of the dynamics of the MDP (while our work considers the purely offline imitation learning setting, with no knowledge beyond expert trajectories).
> - Only achieves horizon-independence for a restricted class of MDPs called RBAS-MDPs (while our work achieves horizon independence for *any MDP*, as long as the rewards are appropriately normalized).
>
> While the work of [2] is related and we are happy to cite it, we believe that all claims of novelty and significance of our work stand.
>
> > There is a gap between the developed theory and practice. It is not safe to conclude that the theory can explain the empirical results in Figure 1. There are possible theoretical explanations for empirical results that differ from this paper. Specifically, [1] showed that MuJoCo locomotion controls have deterministic transitions, thus the statistical estimation error comes from the initial step, and there are no compounding errors. The reviewer believes the explanation provided in [1] is also true for the studied Atari environments.
>
> We emphasize that our experiments are intended to *validate* our theory, particularly the claim that log-loss BC can achieve $H$-independence. We do not claim at any point in the paper that our theory predicts the precise behavior in Figure 1; rather, we claim that Figure 1 validates the theoretical prediction that regret does not degrade as a function of horizon.
>
> Regarding the point about deterministic transitions in MuJoCo and Atari, we emphasize that even if dynamics are deterministic, one can still experience compounding errors due to stochasticity in the *learned policy*. This phenomenon has been widely observed [11,15, 39].
>
> > The reviewer believes that practitioners choose stationary policies in practice because the MDP formulation for practical tasks has stationary transitions, rather than the non-stationary ones studied in this paper. For stationary-transition MDPs, the estimation error is, in fact, smaller.
>
> We believe there may be some confusion here.  A key feature of our analysis is that it supports both non-stationary *and* stationary policies (note that while we use the notation $\pi_h$ throughout the paper, stationary policies are simply a special case in which $\pi_h=\pi$ for all $h$), and we state a number of improved guarantees special to *stationary* policies, e.g. discussion at the bottom of page 6. If the reviewer has a specific question regarding these results that we can clarify, we would greatly appreciate it.
>
> > To the reviewer's understanding, this paper mainly focuses on the model selection problem under general function class approximation. This does not provide many new insights on the compounding errors issues and horizon dependence issues in imitation learning. The title may be revised to reflect the contribution.
>
> We believe this summary is misguided; please see the response to question #1.

---

> > ### Comment · Reviewer_FtQL · 2024-08-12
> >
> > I appreciate the detailed explanation. I am ready to increase my review score, but I still have some concerns and futher comments are provided below.
> >
> > **Comment 1:** I clearly understand the response about the 'horizon-free' guarantee. I would like to suggest adding a figure to the paper to clarify the concepts between supervised learning and imitation learning. I also have an additional comment about the reward notion 'R', which I elaborate on below.
> >
> > **Comment 2:** Could the authors clarify whether the rollout policy for expected regret in Figures 1 and 2 is deterministic or stochastic? I could not find this detail in the Appendix (although I see that the Appendix mentions the expert policy for data collection is deterministic). I ask this because I am conjecturing the following: if the learner's policy is deterministic, the 'unnormalized' regret is also independent of H for the tested environments. This is why I reference the previous work [1] in my review comment. I noticed the authors mentioned using a stochastic policy, but it does not make sense to consider stochastic policies for such tasks.
> >
> > I want to clarify that I agree the provided empirical results are consistent with the developed theory. However, I am pointing out other possibilities: practitioners usually care about absolute performance, relating to the notions of regret and 'R'. For BC on these tasks, the gap may be independent of the horizon because of deterministic transitions. For the same reason, due to an interest in absolute performance, practitioners also care about adversarial training algorithms like GAIL, which should a gap of $O(1/H * R)$ when using the notion of expected regret, as such methods are observed to achieve good performance regardless of horizon length.

---

> > > ### Author Response · Authors · 2024-08-12
> > > **Response to Comment**
> > >
> > > Thank you for your interest! We are happy to clarify these points.
> > >
> > > ### **Regarding comment 1:**
> > > We are happy to include a figure that better explains the distinction between supervised learning and IL in the next version of the paper.
> > >
> > > ### **Regarding comment 2:**
> > > This is a subtle issue.  One point we should have mentioned earlier is that even though the transitions of the MDP are deterministic, the *initial state is stochastic*, both in MuJoCo and in Atari, which is the default in the Farama Foundation’s Gymnasium.  In this setting, even if every policy is fully deterministic, we expect unnormalized regret to depend on horizon. This is true in theory (in fact, our lower bound for deterministic experts, Theorem 2.2 in our paper, covers exactly this setting) but also intuitively holds true in MuJoCo experiments for the following reason.  Suppose the initial state is drawn so that the learner makes a mistake, leading the walker to fall over; then the reward after the walker has fallen will be zero, even though the expert is able to accrue reward proportional to the horizon by continuing to walk.  To summarize, even when both expert and imitator are deterministic and the MDP has deterministic transitions, as long as the initial state is stochastic, we expect the *unnormalized* regret to scale with the horizon.
> > >
> > > ### **Regarding GAIL:**
> > > We agree with the reviewer that IRL methods can be beneficial in practice. However, regarding the comment about GAIL specifically, we emphasize that in a minimax sense, the regret *can* be larger than $R / H$, as shown by our Theorem 2.2; indeed, the lower bound in Theorem 2.2 demonstrates that linear scaling with $H$ is necessary for *any algorithm* (information-theoretically) in a worst-case sense when we allow dense rewards (note that in the setting of Theorem 2.2, we have $R/H = 1$). We also remark that GAIL assumes access to a global simulator, allowing the learner to sample trajectories from the MDP according to a given policy, which goes beyond the access model we study for behavior cloning. One interesting takeaway from our work (Theorem 2.2) is the observation that in the worst case, this additional assumption does not lead to improved performance.  However, we emphasize that we agree with the reviewer that IRL methods likely have benefits in practice, and we think that it is an interesting question to better understand the extent to which such improved performance can be proved for some “nice” sub-class of MDPs.

---

> > > > ### Comment · Reviewer_FtQL · 2024-08-13
> > > >
> > > > Thanks a lot for the discussion! All of my concerns have been addressed. I have changed my score from 5 to 7.

---

### Official Review · Reviewer_DwTf · 2024-07-09

**Soundness:** 2
**Presentation:** 3
**Contribution:** 2
**Rating:** 4
**Confidence:** 3

**Summary:**

This paper analyzes the sample complexity of offline behavior cloning w.r.t. the trajectory horizon. The theoretical result reveals that offline behavior cloning actually does not suffer more from the long horizon than the online BC, under two assumptions.

**Strengths:**

- The theory to further understand the complexity of BC is important.
- The presentation is clear and easy to follow.

**Weaknesses:**

- The result is intuitive since with "parameter sharing", we assume the policy generalizes to longer horizon states.
- One important assumption made in this paper, i.e. the policy class using parameter sharing, is unclear. For example, how the parameter sharing will affect practical learning? Does it introduce any bias for the policy learning?
- The experiments are not solid. For example, the Atari figure seems like the Expected regret is random w.r.t. horizon. It is natural to question whether the MuJoCo results really fit the theoretical prediction or simply are due to the task properties.

**Questions:**

N/A

**Limitations:**

See weakness.

---

> ### Author Rebuttal · Authors · 2024-08-06
>
> We thank the reviewer for their positive review and their helpful comments! Please see responses to individual questions below.
>
> > The result is intuitive since with "parameter sharing", we assume the policy generalizes to longer horizon states.
>
> While it is certainly intuitive that smaller policy classes yield improved rates, we emphasize that our work is the first to demonstrate that under the assumption of parameter sharing, horizon-free regret is possible with BC in general MDPs; we would like to push back on the idea that it is intuitive that this result holds in such generality, as this finding runs counter to the intuition expressed in many prior works. In particular, one of the key consequences of our work is that under the assumptions of parameter sharing and sparse rewards, the key challenge of compounding error is not present in BC when using Log loss, a finding which we believe is somewhat surprising in light of previous work.
>
>
> > One important assumption made in this paper, i.e. the policy class using parameter sharing, is unclear. For example, how the parameter sharing will affect practical learning? Does it introduce any bias for the policy learning?
>
> We would like to emphasize that parameter sharing is not actually an “assumption” in our paper, but rather is an important special case of our general results.  Indeed, our main results hold in the general case of arbitrary policy classes (Theorems 2.1 and 2.3). Nonetheless, we emphasize that parameter sharing is a natural assumption in practice, and is satisfied whenever the expert policy is stationary with respect to time, which commonly occurs in application domains such as robotics and transformers. Indeed, if parameter sharing does not hold, it means we are training a completely different policy for each step $h$, which is rarely, if ever, done in practice.
>
> We mention in passing that beyond its practical relevance, parameter sharing is a useful special case to isolate the effect that horizon has on offline imitation learning, in the sense that it illustrates the possibility of horizon-free rates.
>
>
> > The experiments are not solid. For example, the Atari figure seems like the Expected regret is random w.r.t. horizon.
>
> Regarding the Atari experiments, we emphasize that Figure 1b and 3b show a a clear trend where the regret with 200 trajectories decreases uniformly as the horizon increases, a demonstration that is amplified by the lack of overlap in the confidence intervals (which is consistent with our theory as we prove that the regret should not increase with horizon). Note that this empirical setup is consistent with our theory, as we have a stationary expert and sparse rewards, and thus expect the regret to be non-increasing with respect to horizon. If the reviewer can clarify why they believe the expected regret is "random" w.r.t. horizon for these experiments, we would greatly appreciate it.
>
>
> Perhaps the only surprising feature of the Atari experiments is that the regret is actually *decreasing* with horizon (as opposed to being non-increasing). To understand this, note that our theory provides horizon-agnostic upper bounds independent of the environment. Our lower bounds are constructed for specific worst-case environments, and do not rule out the possibility of improved performance with longer horizons environments with favorable structure. We conjecture that this phenomenon is related to the fact that longer horizons yield fundamentally more data, as the total number of state-action pairs in the expert dataset is equal to nH. In the final version of the paper, we will expand the discussion around this phenomenon, but we would like to emphasize that in no way do our experimental results conflict with the theory established in the rest of the paper.
>
> > It is natural to question whether the MuJoCo results really fit the theoretical prediction or simply are due to the task properties.
>
> We are a little confused by this point, but would be more than happy to respond if the reviewer would be willing to further elaborate and clarify why they feel that the MuJoCo results do not fit the theoretical prediction, or explain which task properties they are concerned with.

---

> > ### Author Response · Authors · 2024-08-12
> > **Following up**
> >
> > Dear reviewer,
> >
> > Thank you for your time! We are following up to check whether our rebuttal has addressed your concerns. Please let us know if you have any further questions.
> >
> > Thank you,
> > Authors

---

### Official Review · Reviewer_meEq · 2024-07-09

**Soundness:** 3
**Presentation:** 3
**Contribution:** 3
**Rating:** 6
**Confidence:** 4

**Summary:**

This paper studies the horizon dependence in imitation learning (IL). In particular, the authors analyze the sample complexity of Behavioral Cloning (BC) with logarithmic loss and general policy class $\Pi$.

For deterministic experts, they first present a sharp horizon-independent regret decomposition and then provide a regret bound for BC. They show that BC achieves a linear horizon dependence under dense rewards when $\log (|\Pi|) = O (1)$, including stationary policies and policies with parameter sharing. Furthermore, they prove a lower bound, indicating that online IL cannot improve over offline IL with $|\Pi|=2$.

For stochastic experts, they present a horizon-independent and variance-dependent regret decomposition and then give a regret bound for BC. This result shows that when $\log (|\Pi|) = O (1)$, BC can achieve a fully horizon-independent sample complexity in the sparse reward case. As for the dense reward case, BC requires a quadratic sample complexity, which is proven to be necessary in both offline and online settings.

**Strengths:**

1. This paper provides a new understanding of the horizon dependence in IL. Classical IL works show that the offline IL method BC suffers a quadratic horizon dependence, motivating various online IL approaches that attain improved linear horizon dependence. However, this paper proves that in certain cases, it is possible for BC to achieve linear horizon dependence and online IL cannot improve over offline IL, suggesting that there is no fundamental gap between offline IL and online IL.

2. This paper contributes new techniques for analyzing IL methods, which could be of independent interest. To attain a linear horizon dependence for BC, this paper presents a sharp horizon-independent regret decomposition. For deterministic experts, this regret decomposition is based on a novel trajectory-level analysis. For stochastic experts,  they provide an interesting information-theoretic analysis. I believe that these new techniques could inspire further advancements in IL theory.

**Weaknesses:**

1. Some formulas and claims in this paper are misleading. In the logarithmic loss of Eq. (5), the policy variable $\pi (a^i_h|s^i_h)$ is written as a stationary policy, which does not include the case of non-stationary policies $\Pi = \Pi_1 \times \Pi_2 \times \cdots \times \Pi_H$ discussed in lines 242-248. Moreover, in Eq. (39) and Eq. (40) in Appendix E.1, the policy becomes non-stationary again, which is inconsistent with Eq. (5). As such, I suggest the authors to write logarithmic losses for stationary policies and non-stationary policies separately. Furthermore, in line 224, they claim that “This bound improves upon the guarantee for indicator loss behavior cloning in Eq.(3) by an O(H) factor”. This statement is a little problematic since the bound in Corollary 2.1 does **not** improve upon previous bounds for the case of non-stationary policies.
2. This paper focuses on the horizon dependence in imitation learning. However, this paper misses a closely related work [1] that proves that a kind of IL method called adversarial imitation learning can achieve a horizon-independent sample complexity bound in a certain class of IL problems.

References:

[1] Xu et al., Understanding Adversarial Imitation Learning in Small Sample Regime: A Stage-coupled Analysis, 2023.

[2] Rajamaran et al., Toward the fundamental limits of imitation learning, 2020.

**Questions:**

1. Theorem 2.4 implies that the slow $H/\sqrt{n}$ rate for $\sigma^2_{\pi^*}=H^2$ is necessary in both offline and online IL. How can we reconcile this result with the claim in [2] that BC can achieve the $1/n$ rate for stochastic experts in the tabular setting?

**Limitations:**

The authors have adequately discussed and addressed the limitations of this paper.

---

> ### Author Rebuttal · Authors · 2024-08-06
>
> We thank the reviewer for their positive review and their helpful comments! Please see responses to individual questions below.
>
> > Some formulas and claims in this paper are misleading. In the logarithmic loss of Eq. (5), the policy variable is written as a stationary policy, which does not include the case of non-stationary policies discussed in lines 242-248. Moreover, in Eq. (39) and Eq. (40) in Appendix E.1, the policy becomes non-stationary again, which is inconsistent with Eq. (5). As such, I suggest the authors to write logarithmic losses for stationary policies and non-stationary policies separately.
>
> Our results apply as-is to non-stationary policies. We believe this confusion is caused by a small typo, which is that Eq. (5) (as well as the first equation in App E.1) is intended to be stated with $\log(1/\pi_{h}(a_h | x_h))$, not $\log(1/\pi(a_h | x_h))$. We will be sure to include the `_h` subscript here and elsewhere in the final version of the paper.
>
> > Furthermore, in line 224, they claim that “This bound improves upon the guarantee for indicator loss behavior cloning in Eq.(3) by an O(H) factor”. This statement is a little problematic since the bound in Corollary 2.1 does not improve upon previous bounds for the case of non-stationary policies.
>
> The statement “This bound improves upon the guarantee for indicator loss behavior cloning in Eq.(3) by an O(H) factor” is intended to convey that the bound improves upon Eq. (3) for *general* policy classes (which is true), not to claim that the bound offers a strict improvement on a per-policy class basis. While the paper already includes extensive discussion around this point (e.g., Page 6), we are happy to change the wording of the statement to make this as clear as possible.
>
> > This paper focuses on the horizon dependence in imitation learning. However, this paper misses a closely related work [1] that proves that a kind of IL method called adversarial imitation learning can achieve a horizon-independent sample complexity bound in a certain class of IL problems.
>
> Thank you for pointing us to this work, which we are happy to cite and discuss in the final version of our paper. While the paper [1] does indeed contain some results concerning horizon-independence, their findings are quite restrictive compared to the results in our paper (in particular, unlike our work, their notion of horizon-dependence is not consistent with the standard notion consider in prior work on horizon-independent RL, e.g., Jiang & Agarwal ‘18). In more detail, their work:
> * Is restricted to tabular MDPs and policies (while our work considers general MDPs and function approximation).
> * Requires knowledge of the dynamics of the MDP (while our work considers the purely offline imitation learning setting, with no knowledge beyond expert trajectories).
> * Only achieves horizon-independence for a restricted class of MDPs called RBAS-MDPs (while our work achieves horizon-indepedence for *any MDP*, as long as the rewards are appropriately normalized). In particular, our notion of horizon-independence is consistent with that considered in Jiang & Agarwal ‘18 and subsequent work.
>
> Thus, while the work of [1] is certainly related and we are happy to cite it, we believe that all claims of novelty and significance of our work stand.
>
> > Theorem 2.4 implies that the slow rate for is necessary in both offline and online IL. How can we reconcile this result with the claim in [2] that BC can achieve the rate for stochastic experts in the tabular setting?
>
> This is a great question, which we discuss in Footnote 17. To restate here: Rajaraman et al. [2] show that for the tabular setting, it is possible to achieve a fast $1/n$-type rate in-expectation for stochastic policies. Their result critically exploits the assumption that |X| and |A| are small and finite to argue that it is possible to build an unbiased estimator for the expert policy $\pi^\ast$. Theorem 2.4 shows that such a result cannot hold with even constant probability for the same setting, thereby revealing a separation between the best rate that can be achieved in expectation versus the best rate that can be achieved in probability (note that since the expert is not assumed to be optimal, regret can be negative, which precludes the use of Markov’s inequality to convert an in-expectation bound into a bound in probability).
>
> More broadly, we believe the fact that a 1/n-type rate is even possible in expectation for the tabular setting to be an artifact of the unique structure of this setting, and unlikely to hold for general policy classes or MDPs (e.g., for general policy classes, there is no hope of achieving unbiased estimation).

---

> > ### Comment · Reviewer_meEq · 2024-08-12
> >
> > I thank the authors for the detailed response and I would suggest the authors incorporate the above responses in the revised version. Overall, I am maintaining my original score and remain in favor of acceptance.

---

### Official Review · Reviewer_2nRQ · 2024-07-12

**Soundness:** 4
**Presentation:** 4
**Contribution:** 4
**Rating:** 9
**Confidence:** 3

**Summary:**

The paper proposes theoretical results on whether and when can offline imitation learning (IL) match online imitation learning in sample efficiency. The paper connects the results of existing works and demonstrates that offline IL can indeed match online IL under certain conditions. The paper further provides results on stochastic experts rather than deterministic experts.

**Strengths:**

- As far as I know this is the first approach that leverages Hellinger distance on analyzing imitation learning methods.
- The paper extends the BC results to stochastic policies under other function classes.
	- Interesting that online IL cannot really improve upon offline IL without further assumptions.
- In addition to the theory the appendix provides experimentation that supports the theory.
- The paper is very well written---I enjoyed reading the paper.

**Weaknesses:**

- The paper only considers log loss, which I understand is very commonly used, but I am curious about how much of these analyses transfer to other losses.

**Questions:**

- Does the MDP assume finiteness?
	- If so, how would this result change under continuous action space with the most common parameterization (i.e. Gaussian policies)?
- Does online IL algorithm include IRL?

**Limitations:**

- See above.

---

> ### Author Rebuttal · Authors · 2024-08-06
>
> We thank the reviewer for their positive review and their helpful comments! Please see responses to individual questions below.
>
> > The paper only considers log loss, which I understand is very commonly used, but I am curious about how much of these analyses transfer to other losses.
>
> This is a great question! Our analysis takes advantage of unique statistical properties of the log-loss (notably, that it provides *trajectory-level* control over deviations from pi*; see discussion in Sec 2.1.2). In general, other losses do not enjoy similar horizon-independent guarantees (for example, Ross & Bagnell ‘10 show that the indicator loss can have suboptimal horizon dependence, and it is straightforward to see that their counterexample also applies to the square loss and absolute loss). Indeed, one of the key findings of our paper is that using the log-loss is uniquely beneficial.
>
>
> > Does the MDP assume finiteness? If so, how would this result change under continuous action space with the most common parameterization (i.e. Gaussian policies)?
>
> No, our results do not assume finiteness of the action space. Our analysis of Log-loss BC can be applied as-is to Gaussian policy parameterizations, as long as the realizability assumption in Assumption 1.1 holds.
>
> > Does online IL algorithm include IRL?
>
> The formulation of online IL that we consider assumes that the learner has online/interactive access to both the MDP $M$ and expert $\pi^{\star}$. Typical IRL approaches use a less powerful access model in which we have online access to the MDP $M$ in the same fashion, but do not assume online access to $\pi^{\star}$ itself. Hence, all of the lower bounds for online imitation learning in our paper also apply to IRL.

---

> > ### Comment · Reviewer_2nRQ · 2024-08-10
> >
> > Thank you for the response.
> >
> > Regarding IRL, I understand that the lower bound applies to IRL as well, I was more curious about the upper bound. In particular whether the authors have considered using this line of technique to analyze IRL.

---

> > > ### Author Response · Authors · 2024-08-10
> > >
> > > Thank you for the clarification!
> > >
> > > Our main results (Theorem 2.1 and Theorem 2.3) are not specialized to log-loss behavior cloning; rather they are reductions that apply to any algorithm that achieves a bound on the trajectory-level Hellinger distance $D_{H}^2(\mathbb{P}^{\hat{\pi}}, \mathbb{P}^{\pi^{\star}})$. For future work, it would be interesting to see if we can design IRL algorithms based on minimizing this quantity. In particular, many existing IRL algorithms can be viewed as performing distribution matching at the occupancy level, but our analysis suggests that performing distribution matching at the trajectory-level might lead to tighter guarantees with respect to horizon.

---

### Decision · Program_Chairs · 2024-09-25

**Decision:**

Accept (spotlight)

**Comment:**

This paper challenges the existing orthodoxy about imitation learning (IL), which says that the performance of behavioural cloning scales unfavourably in the problem horizon. The paper shows that BC is in fact a competitive IL algorithm when trained using the log loss. This has large practical significance since, in many practical settings, BC is the first (and sometimes only) algorithm that can be shipped.

In the AC-reviewer discussion, one reviewer pointed out concerns about clarity of the writing - please try to have the paper proofread again before submitting the camera-ready version.